# Revisiting Matrix Sketching in Linear Bandits: Achieving Sublinear Regret via Dyadic Block Sketching

**Dongxie Wen**[1]   **Hanyan Yin**[1]   **Xiao Zhang**[1]   **Peng Zhao**[2]   **Lijun Zhang**[2]   **Zhewei Wei**[1]*

[1]Gaoling School of Artificial Intelligence, Renmin University of China, Beijing, China
[2]National Key Laboratory for Novel Software Technology, Nanjing University, Nanjing, China

```
{dongxie, yinhanyan, zhangx89, zhewei}@ruc.edu.cn
{zhaop, zhanglj}@lamda.nju.edu.cn
```

## Abstract

Linear bandits have become a cornerstone of online learning and sequential decision-making, providing solid theoretical foundations for balancing exploration and exploitation. Within this domain, matrix sketching serves as a critical component for achieving computational efficiency, especially when confronting high-dimensional problem instances. The sketch-based approaches reduce per-round complexity from $\Omega(d^2)$ to $O(dl)$, where $d$ is the dimension and $l < d$ is the sketch size. However, this computational efficiency comes with a fundamental pitfall: when the streaming matrix exhibits heavy spectral tails, such algorithms can incur vacuous *linear regret*. In this paper, we revisit the regret bounds and algorithmic design for sketch-based linear bandits. Our analysis reveals that inappropriate sketch sizes can lead to substantial spectral error, severely undermining regret guarantees. To overcome this issue, we propose Dyadic Block Sketching, a novel multi-scale matrix sketching approach that dynamically adjusts the sketch size during the learning process. We apply this technique to linear bandits and demonstrate that the new algorithm achieves *sublinear regret* bounds without requiring prior knowledge of the streaming matrix properties. It establishes a general framework for efficient sketch-based linear bandits, which can be integrated with any matrix sketching method that provides covariance guarantees. Comprehensive experimental evaluation demonstrates the superior utility-efficiency trade-off achieved by our approach.

## 1 Introduction

Multi-Armed Bandits (MAB) is a general framework for modeling sequential decision-making under partial information (Herbert, 1952), which has been widely adopted in various applications, including recommendation systems (Zhang et al., 2022), public health surveillance (Bastani et al., 2021), and green security (Xu et al., 2021). We consider the Stochastic Linear Bandit (SLB), a variant of the MAB under the linear assumption (Auer, 2002; Dani et al., 2007; Abbasi-Yadkori et al., 2011; Chu et al., 2011). In SLB, at round $t$, the player selects an arm $\boldsymbol{x}_t$ from a decision set $\mathcal{X}_t \subseteq \mathbb{R}^d$, and then observes the reward $r_t \in \mathbb{R}$. The expected reward $\mathbb{E}[r_t|\boldsymbol{x}_t] = \boldsymbol{x}_t^\top \boldsymbol{\theta}_\star$, where $\boldsymbol{\theta}_\star$ represents unknown coefficients. Utilizing the regularized least squares estimator and upper confidence bounds, the seminal work Abbasi-Yadkori et al. (2011) propose OFUL algorithm and achieve $\widetilde{O}(d\sqrt{T})$ regret bound, where $d$ is the dimension and $T$ denotes the number of rounds, and the $\widetilde{O}(\cdot)$-notation hides logarithmic factors. Notably, OFUL exhibits a complexity of $\Omega(d^2)$ per step.

In real-world decision-making problems, $d$ can be very large such that traditional linear bandits become computationally prohibitive. Consequently, various studies apply *matrix sketching* techniques to eliminate the quadratic dependence on $d$ and enhance efficiency. Yu et al. (2017) use random projection to map high-dimensional arms to a low $m$-dimensional subspace, reducing the update

---

*Zhewei Wei is the corresponding author.

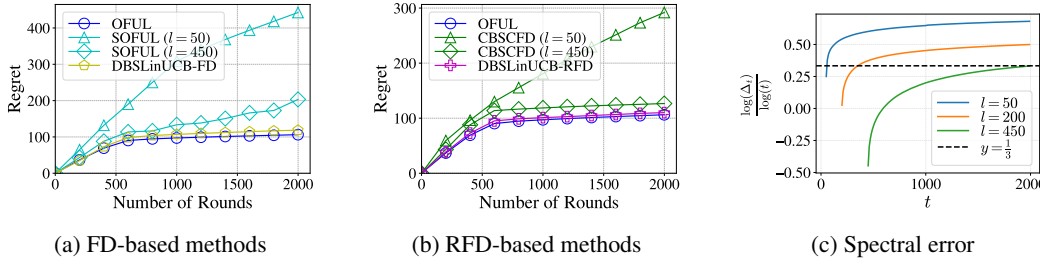

(a) FD-based methods     (b) RFD-based methods     (c) Spectral error

Figure 1: (a), (b): Cumulative regret of the compared algorithms, the proposed methods on synthetic dataset; (c): Scaling of spectral error with rounds on synthetic dataset w.r.t. sketch size $l$

time from $\Omega(d^2)$ to $O(md + m^3)$. Another line of these works is based on a well-known deterministic sketching method – Frequent Directions (FD), which has been proved to offer better theoretical guarantees than random projection under the streaming setting (Liberty, 2013; Woodruff, 2014; Ghashami et al., 2016). Kuzborskij et al. (2019) are the first to employ FD to sketch the covariance matrix in linear bandits, reducing time complexity to $O(dl + l^2)$ while achieving an $\widetilde{O}((1 + \Delta_T)^{3/2}(l + d \log(1 + \Delta_T))\sqrt{T})$ regret, where $l < d$ is the sketch size and $\Delta_T$ represents the spectral error introduced by matrix sketching. Subsequently, Chen et al. (2020) extended this work by substituting FD with a robust variant, which reduces the order of the spectral error $\Delta_T$ and decouples it from $d$, yielding a regret bound of $\widetilde{O}((\sqrt{l + d \log(1 + \Delta_T)} + \sqrt{\Delta_T})\sqrt{lT})$.

**Motivation.** However, sketching-based methods suffer from the *linear regret pitfall*—catastrophic regret when matrices exhibit heavy spectral tails. Figure 1a, 1b illustrate this phenomenon through the regret of SOFUL and CBSCFD across different sketch size $l$. When $l = 450$, both sketch-based methods achieve regret comparable to that of the non-sketched OFUL. In stark contrast, when $l = 50$, they exhibit near-linear regret growth, demonstrating severe performance degradation. This discrepancy arises because insufficient size fails to preserve essential spectral information, resulting in substantial spectral error. The possibility of linear regret contradicts the objective of online learning, emphasizing the critical need to manage worst-case regret when employing matrix sketching.

Intuitively, avoiding the linear regret pitfall requires calibrating sketch size to the spectral properties of the matrix. However, existing methods employ single-scale sketching with *fixed* sketch size throughout learning. This rigid design creates a dilemma: optimal sketch size depends on spectral properties that remain unknown until data arrives, yet must be specified before learning begins. Too small risks catastrophic regret; too large sacrifices computational efficiency—the very motivation for sketching. This inherent tension raises a critical question: *Can we adaptively adjust sketch size during learning to guarantee sublinear regret without prior knowledge of the streaming matrix?*

**Contributions.** We answer the question affirmatively by developing a novel framework for sketch-based linear bandits. Our main contributions are summarized as follows.

- Uncovering the impact of spectral error on regret. We revisit the regret bound of sketch-based linear bandits, focusing on the spectral error induced by matrix sketching. Our analysis reveals that existing methods are susceptible to linear regret, primarily caused by an insufficient sketch size.

- Controlling approximation error via multi-scale sketching. We propose Dyadic Block Sketching, a novel matrix sketching method that adaptively adjusts sketch sizes across multiple scales to control error. We prove that the global error is bounded by a predetermined error $\epsilon$. Additionally, our method provably tracks the optimal rank-$k$ approximation in the streaming setting, ensuring efficiency in scenarios with low-rank matrices or light-tailed spectra.

- Achieving sublinear regret. By applying the proposed sketching framework to linear bandits, we effectively address the issue of linear regret observed in prior works. Our method ensures a sublinear regret, even when the streaming matrix is heavy-tailed. Furthermore, it is robust, scalable, and flexible, achieving diverse regret bounds through various matrix sketching approaches.

**Organization.** The rest is structured as follows. Section 2 revisits sketching in bandits and highlights current pitfalls. Section 3 and Section 4 present our novel multi-scale sketching method and its application to linear bandits. Section 5 reports the experiments. Finally, Section 6 concludes the paper. Due to page limits, the notations and all proofs are provided in the appendices.

## 2 PRELIMINARIES

**Notations.** Let $[n] = \{1, 2, \ldots, n\}$, upper-case bold letters (e.g., $\boldsymbol{A}$) represent matrix and lower-case bold letters (e.g., $\boldsymbol{a}$) represent vectors. We denote by $\|\boldsymbol{A}\|_2$ and $\|\boldsymbol{A}\|_F$ the spectral and Frobenius norms of $\boldsymbol{A}$. We define $|\boldsymbol{A}|$ and $\mathrm{Tr}(\boldsymbol{A})$ as the determinant and trace of matrix $\boldsymbol{A}$. For a positive semi-definite matrix $\boldsymbol{A}$, the matrix norm of vector $\boldsymbol{x}$ is defined by $\|\boldsymbol{x}\|_{\boldsymbol{A}} = \sqrt{\boldsymbol{x}^\top \boldsymbol{A} \boldsymbol{x}}$. For two positive semi-definite matrices $\boldsymbol{A}$ and $\boldsymbol{B}$, we use $\boldsymbol{A} \succeq \boldsymbol{B}$ to represent that $\boldsymbol{A} - \boldsymbol{B}$ is positive semi-definite. We use $\boldsymbol{A} = \boldsymbol{U}\boldsymbol{\Sigma}\boldsymbol{V}^\top$ to represent the SVD of $\boldsymbol{A}$, where $\boldsymbol{U}, \boldsymbol{V}$ denote the left and right matrices of singular vectors and $\boldsymbol{\Sigma} = \mathrm{diag}(\sigma_1, ..., \sigma_n)$ is the diagonal matrix of singular values in the descending order. We define $\boldsymbol{A}_{[k]} = \boldsymbol{U}_k \boldsymbol{\Sigma}_k \boldsymbol{V}_k^\top$ for $k \leq \mathrm{rank}(\boldsymbol{A})$ as the best rank-$k$ approximation to $\boldsymbol{A}$, where $\boldsymbol{U}_k \in \mathbb{R}^{n \times k}$ and $\boldsymbol{V}_k \in \mathbb{R}^{d \times k}$ are the first $k$ columns of $\boldsymbol{U}$ and $\boldsymbol{V}$.

### 2.1 FREQUENT DIRECTIONS

Frequent Directions (FD) (Liberty, 2013; Ghashami et al., 2016) is a deterministic matrix sketching technique. Given a streaming matrix $\boldsymbol{X}^{(t)} = [\boldsymbol{x}_1^\top, \ldots, \boldsymbol{x}_t^\top]^\top \in \mathbb{R}^{t \times d}, t \in [T]$, FD maintains a smaller sketch matrix $\boldsymbol{S}^{(t)} \in \mathbb{R}^{l \times d}$ to approximate $\boldsymbol{X}^{(t)}$, where $l$ denotes the sketch size. To process row $\boldsymbol{x}_t$, we first replace the last row of $\boldsymbol{S}^{(t)}$ with $\boldsymbol{x}_t$. Then, we perform SVD on $\boldsymbol{S}^{(t)}$, i.e., $\boldsymbol{S}^{(t)} = \boldsymbol{U}^{(t)}\boldsymbol{\Sigma}^{(t)}\boldsymbol{V}^{(t)}$. Let $\boldsymbol{\Sigma}^{(t)} = \mathrm{diag}(\sigma_1, \ldots, \sigma_d)$ and $\sigma = \sigma_l^2$, where $\sigma_l$ is the $l$-th largest singular value. Subsequently, we set $\boldsymbol{\Sigma}^{(t+1)} = \mathrm{diag}(\sqrt{\sigma_1^2 - \sigma}, \ldots, \sqrt{\sigma_l^2 - \sigma})$ and $\boldsymbol{S}^{(t+1)} = \boldsymbol{\Sigma}^{(t+1)}\boldsymbol{V}^{(t)}$.

We provide the pseudo-code of FD in Appendix B.1. FD uses $O(dl)$ space and has an amortized update time of $O(dl)$. The fundamental property of FD is to bound the covariance error in terms of the tail eigenvalues of $\boldsymbol{X}^{(T)}$. This property is formally expressed in the following lemma:

**Lemma 1** (Claim 1 of Liberty (2013) ). *Let $\boldsymbol{X}^{(T)}$ be the streaming matrix at round $T$ and $\boldsymbol{X}_{[k]}^{(T)}$ denote the matrix consisting of the first $k$ singular vectors of $\boldsymbol{X}^{(T)}$. Then, it holds that*

$$\left\| (\boldsymbol{X}^{(T)})^\top \boldsymbol{X}^{(T)} - (\boldsymbol{S}^{(T)})^\top \boldsymbol{S}^{(T)} \right\|_2 \leq \Delta_T, \ \text{ where } \ \Delta_T := \min_{0 \leq k < \ell} \frac{\left\| \boldsymbol{X}^{(T)} - \boldsymbol{X}_{[k]}^{(T)} \right\|_F^2}{\ell - k}. \quad (1)$$

### 2.2 LINEAR BANDITS

We first introduce the basic assumptions for the linear bandits setting. At any round $t$, the decision set $\mathcal{X}_t \subset \mathbb{R}^d$ is finite and for all $\boldsymbol{x} \in \mathcal{X}_t$, we have $\|\boldsymbol{x}\|_2 \leq L$. The reward for choosing arm $\boldsymbol{x}_t$ is defined as $r_t = \boldsymbol{x}_t^\top \boldsymbol{\theta}_\star + \eta_t$, where $\boldsymbol{\theta}_\star$ is a fixed, unknown vector of real coefficients, and $\eta_t$ denotes the conditionally $R$-subgaussian noise variable. Moreover, the norm $\|\boldsymbol{\theta}_\star\|_2$ is upper bounded by $H$.

OFUL (Abbasi-Yadkori et al., 2011) utilizes regularized least squares (RLS) to estimate $\boldsymbol{\theta}_\star$ as

$$\boldsymbol{A}^{(t)} = \lambda \boldsymbol{I}_d + \left( \boldsymbol{X}^{(t)} \right)^\top \boldsymbol{X}^{(t)}, \quad \widehat{\boldsymbol{\theta}}_t = \left( \boldsymbol{A}^{(t)} \right)^{-1} \sum_{s=1}^{t} r_s \boldsymbol{x}_s, \quad (2)$$

where $\boldsymbol{X}^{(t)} = [\boldsymbol{x}_1^\top, \ldots, \boldsymbol{x}_t^\top]^\top$ is the matrix containing all the arms selected up to round $t$ and $\lambda$ is the regularization. After computing the confidence ellipsoid $\beta_t(\delta)$, the arm selection is based on the upper confidence bound as $\boldsymbol{x}_{t+1} = \arg\max_{\boldsymbol{x} \in \mathcal{X}_t} \{\boldsymbol{x}^\top \widehat{\boldsymbol{\theta}}_t + \beta_t(\delta) \cdot \|\boldsymbol{x}\|_{(\boldsymbol{A}^{(t)})^{-1}}\}$.

The objective of the learner is to minimize the cumulative (pseudo) regret (Lattimore & Szepesvári, 2020) over the total $T$ rounds, defined as $\mathrm{Regret}_T = \sum_{t=1}^{T} \max_{\boldsymbol{x} \in \mathcal{X}_t} \boldsymbol{x}^\top \boldsymbol{\theta}_\star - \sum_{t=1}^{T} \boldsymbol{x}_t^\top \boldsymbol{\theta}_\star$.

### 2.3 SKETCH-BASED LINEAR BANDITS

Note that both the RLS estimator and arm selection require maintaining the inverse of $\boldsymbol{A}^{(t)}$, which necessitates an update time of $\Omega(d^2)$. To address this issue, Kuzborskij et al. (2019) proposes a sketch-based method, SOFUL, which reduces the time-consuming step via matrix sketching.

SOFUL produces a FD sketch $\boldsymbol{S}^{(t)} \in \mathbb{R}^{l \times d}$ of the streaming matrix $\boldsymbol{X}^{(t)}$. By applying Woodbury's identity, the inverse of the sketched covariance matrix can be written as $\left( \widehat{\boldsymbol{A}}^{(t)} \right)^{-1} =$

$\frac{1}{\lambda}\big(\boldsymbol{I}_d - (\boldsymbol{S}^{(t)})^\top \boldsymbol{M}^{(t)} \boldsymbol{S}^{(t)}\big)$, where $\boldsymbol{M}^{(t)} = \big(\boldsymbol{S}^{(t)}(\boldsymbol{S}^{(t)})^\top + \lambda \boldsymbol{I}_l\big)^{-1} \in \mathbb{R}^{l \times l}$ is a diagonal matrix that can be stored efficiently. Notably, $(\widehat{\boldsymbol{A}}^{(t)})^{-1}$ can be updated implicitly using the sketch matrix $\boldsymbol{S}^{(t)}$ and $\boldsymbol{M}^{(t)}$. Since matrix-vector multiplications with $\boldsymbol{S}^{(t)}$ require $O(dl)$ time and matrix-matrix multiplications with $\boldsymbol{M}^{(t)}$ take $O(l^2)$ time, the update cost is reduced from $\Omega(d^2)$ to $O(dl + l^2)$.

**Current Pitfalls.** Despite its improved efficiency, the sketch-based methods introduce errors in matrix approximation, which may lead to a vacuous linear regret bound. We begin by presenting the regret bound of SOFUL, which is characterized in terms of spectral error.

**Lemma 2** (Theorem 3 of Kuzborskij et al. (2019)). *Let* $\mathrm{Regret}_T^{\texttt{SOFUL}}$ *denote the regret of SOFUL, where the sketch size is $l$ and $\Delta_T$ is defined in equation 1. With high probability, the regret satisfies*

$$\mathrm{Regret}_T^{\texttt{SOFUL}} = \widetilde{O}\left(\min\left\{(1 + \Delta_T)^{\frac{3}{2}}(l + d\log(1 + \Delta_T))\sqrt{T}, T\right\}\right).$$

The regret bound of SOFUL is tightly linked to spectral error $\Delta_T$, which depends on both the spectral tail of $\boldsymbol{X}^{(T)}$ and the fixed sketch size $l$. This bound is meaningful only when $\Delta_T = o(T^{1/3})$. However, as shown in Figure 1c, if the sketch size is insufficient (e.g., the blue and orange lines), $\Delta_T$ grows rapidly with the number of rounds, violating this condition and leading to linear regret.

The underlying reason for this phenomenon is that low-regret algorithms must ensure sufficient exploration by estimating all relevant directions in the parameter space, a concept extensively studied by Banerjee et al. (2023). Specifically, they showed that when the arm space has a locally convex surface, the minimum eigenvalue of the covariance matrix satisfies $\sigma_d^2 = \Omega(T^q)$ in expectation, where $q \in (0, 1/2]$ depends on the geometry of the arm space. For the convenience of readers, we restate this result in Theorem 5 in Appendix C.1. Based on this result, we obtain the following observation:

**Observation 1.** *Assume the decision set is drawn from a locally convex arm space $\mathcal{X}$. If the sketch size of SOFUL satisfies $l < d - T^{\frac{1}{3} - q}$, then SOFUL incurs vacuous linear regret. Consequently, when the geometry constant $q \geq 1/3$, SOFUL suffers linear regret for any sketch size $l < d$.*

The proof is provided in Appendix C.1. Observation 1 indicates that it is difficult to constrain the spectral error by presetting a fixed sketch size, since the spectral properties of the streaming matrix are unknown in advance. In some cases, even allocating SOFUL the maximum sketch size fails to prevent linear regret. Similarly, other sketch-based methods, such as CBSCFD (Chen et al., 2020), suffer from the same limitation, as discussed in Appendix C.2. This underscores the necessity of dynamically adjusting the sketch size to guarantee worst-case sublinear regret.

## 3 Dyadic Block Sketching for Constrained Global Error Bound

In this section, we propose Dyadic Block Sketching, a novel multi-scale sketching paradigm that fundamentally departs from single-scale sketching. Inspired by dyadic frameworks in streaming algorithms (Wang et al., 2013; Wei et al., 2016), our method maintains multiple sketches with varying sizes. A key property is that the global error is governed by $\epsilon$, which is fixed before sketching.

### 3.1 Algorithm Descriptions

**High-Level Ideas.** As illustrated in Figure 2, we partition the streaming data into blocks, with each block approximated by a matrix sketch. For the initial block, we use a relatively small sketch, and for each subsequent block, the sketch size is doubled compared to the previous one. The following lemma shows that any sketch satisfying a covariance error guarantee is decomposable, allowing us to concatenate the individual sketches to construct an approximation of the entire streaming matrix:

**Lemma 3** (Decomposability). *Let* $\boldsymbol{X} = [\boldsymbol{X}_1^\top, \dots, \boldsymbol{X}_p^\top]^\top$ *with* $\boldsymbol{X}_i \in \mathbb{R}^{n_i \times d}$ *and* $\sum_i n_i = n$. *If each* $\boldsymbol{X}_i$ *admits a sketch* $\boldsymbol{S}_i$ *satisfying* $\|\boldsymbol{X}_i^\top \boldsymbol{X}_i - \boldsymbol{S}_i^\top \boldsymbol{S}_i\|_2 \leq \epsilon_i \|\boldsymbol{X}_i\|_F^2$, *then with* $\boldsymbol{S} = [\boldsymbol{S}_1^\top, \dots, \boldsymbol{S}_p^\top]^\top$ *we have* $\|\boldsymbol{X}^\top \boldsymbol{X} - \boldsymbol{S}^\top \boldsymbol{S}\|_2 \leq \sum_{i=1}^p \epsilon_i \|\boldsymbol{X}_i\|_F^2$.

**Data Structure.** The matrix rows are partitioned into blocks, with each block represented as the struct variable $\mathcal{B}$. Each block is associated with a matrix sketching instance, denoted as $\mathcal{B}.sketch$,

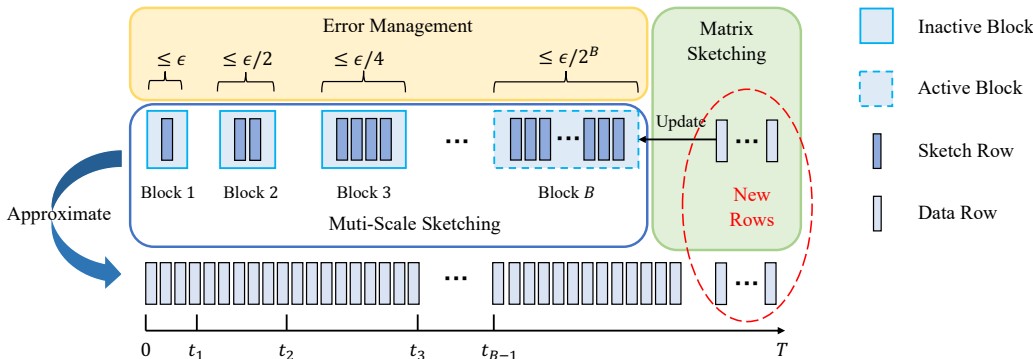

Figure 2: Illustration for Dyadic Block Sketching. For each inactive block $i \in [B - 1]$, the sketch covers the data from $t_{i-1}$ to $t_i$. For active block $B$, sketching updates are performed on the new rows. In Algorithm 1, $\mathcal{B}^\star$ represents the active block $B$, and $\mathcal{L}$ denotes the list of inactive blocks.

which covers a segment of consecutive, non-overlapping rows. Each block is characterized by two properties: block size and sketch size. The block size is defined as the sum of the squared norms of the rows contained within the block, i.e., $\mathcal{B}.\text{BlockSize} = \sum_{x \in \mathcal{B}} \|x\|_2^2$. The sketch size, $\mathcal{B}.\text{SketchSize}$, represents the constant sketch size of the sketch matrix associated with the block.

We categorize the blocks into two states: *active* and *inactive*. An active block is updated with new rows, while an inactive block remains unchanged. As shown in Figure 2, there is exactly one active block at any time. The active block is represented as $\mathcal{B}^\star$, and the list of inactive blocks is denoted by $\mathcal{L}$. Moreover, we maintain two invariants during the update process for error management:

**Invariant 1** (Inactive-Block Condition). *Each inactive block either has sketch size no smaller than its rank, or block size less than $\epsilon l_0$, where $l_0$ is the initial sketch size and $\epsilon$ the error parameter.*

**Invariant 2** (Maximum Number of Blocks). *The number of blocks is at most $\lfloor \log(d/l_0 + 1) \rfloor$.*

Algorithm 1 presents the pseudo-code of Dyadic Block Sketching. Upon receiving a new row $x_t$, we first verify that Invariant 2 holds. If the number of blocks reaches its upper limit, any error introduced by a matrix sketch becomes intolerable, requiring the full preservation of the information. Concretely, we employ complete rank-1 modifications to update the sketch.

In Lines 6–13, we update the active block and, when necessary, create a new block. The sketch $\mathcal{B}^\star.\text{sketch}$ is updated using a chosen matrix sketching method (e.g., FD or RFD; see Appendix B.1). We introduce a Boolean flag `willExcessSketch` to check the rank condition in Invariant 1: it is set to *True* if inserting the incoming row would cause the block rank to exceed the current sketch size, and *False* otherwise. This flag is easily computed by testing the update on a temporary copy of the sketch $\mathcal{B}^\star.\text{sketch}$; for FD, we tentatively add the new row and inspect the shrinking value $\sigma$: if $\sigma = 0$ (unchanged from before insertion) we set `willExcessSketch` to *False*, otherwise to *True*. Line 8 updates the information of the active block when either the block size remains below the threshold $\epsilon l_0$ or `willExcessSketch` is *False*. Otherwise, we mark the current active block as inactive, append it to $\mathcal{L}$, initialize a new active block with twice the previous sketch size, and then insert the incoming row into this new block.

In Lines 14–15, we return a matrix approximation for the current streaming matrix. We query the sketch matrices $S^\star$ and $M^\star$ from the active block $\mathcal{B}^\star$. Since the inactive blocks remain fixed, we can query the combined results of the sketch matrices from $\mathcal{L}$, denoted as $\widetilde{S}$ and $\widetilde{M}$, which are updated (similar to equation 3) once when a block is marked as inactive. Leveraging the decomposability property (Lemma 3), we combine the sketches from both the active and inactive blocks as follows:

$$S^{(t)} = \begin{bmatrix} \widetilde{S} \\ S^\star \end{bmatrix}, \qquad M^{(t)} = \left( \begin{bmatrix} \widetilde{M} & \widetilde{S}\, S^{\star\top} \\ S^\star \widetilde{S}^\top & M^\star \end{bmatrix} + \lambda I \right)^{-1}. \tag{3}$$

---

**Algorithm 1** Dyadic Block Sketching

---

**Input:** Data stream $\{\boldsymbol{x}_t\}_{t=1}^T$, initial sketch size $l_0$, error parameter $\epsilon$, regularization $\lambda$
**Output:** Sketch matrix $\boldsymbol{S}^{(t)}$, $\boldsymbol{M}^{(t)}$
 1: Initialize an empty list $\mathcal{L}$ and $\mathcal{B}^\star.sketch$, set $\mathcal{B}^\star$.BlockSize = 0 and $\mathcal{B}^\star$.SketchSize = $l_0$
 2: **for** $t = 1$ **to** $T$ **given** $\boldsymbol{x}_t$ **do**
 3:  **if** length($\mathcal{L}$) $\geq \lfloor \log(d/l_0 + 1) \rfloor - 1$ **then**
 4:   Update $\left(\boldsymbol{S}^{(t+1)}\right)^\top \boldsymbol{S}^{(t+1)} = \left(\boldsymbol{S}^{(t)}\right)^\top \boldsymbol{S}^{(t)} + \boldsymbol{x}_t^\top \boldsymbol{x}_t$ using rank-1 modifications
 5:  **else**
 6:   Query $\mathtt{willExcessSketch}$ from $\mathcal{B}^\star.sketch$ and $\boldsymbol{x}_t$
 7:   **if** $\mathcal{B}^\star$.BlockSize $+ \|\boldsymbol{x}_t\|_2^2 < \epsilon \cdot l_0$ **or** $\mathtt{willExcessSketch}$ is *False* **then**
 8:    Update $\mathcal{B}^\star.sketch$ with $\boldsymbol{x}_t$ and set $\mathcal{B}^\star$.BlockSize $+= \|\boldsymbol{x}_t\|_2^2$
 9:   **else**
10:    Set $l = \mathcal{B}^\star$.SketchSize and mark $\mathcal{B}^\star$ as inactive (appending it to $\mathcal{L}$)
11:    Initialize a new empty $\mathcal{B}^\star.sketch$ and set $\mathcal{B}^\star$.BlockSize = 0, $\mathcal{B}^\star$.SketchSize = $2l$
12:    Update $\mathcal{B}^\star.sketch$ with $\boldsymbol{x}_t$ and set $\mathcal{B}^\star$.BlockSize $+= \|\boldsymbol{x}_t\|_2^2$
13:   **end if**
14:   Query $\boldsymbol{S}^\star$, $\boldsymbol{M}^\star$ from $\mathcal{B}^\star.sketch$ and $\widetilde{\boldsymbol{S}}, \widetilde{\boldsymbol{M}}$ from the list $\mathcal{L}$ of inactive blocks
15:   Compute $\boldsymbol{S}^{(t)}$, $\boldsymbol{M}^{(t)}$ by equation 3
16:  **end if**
17: **end for**

---

## 3.2 Analysis

We now analyze the error guarantee and the space-time complexities of our approach. Let $\boldsymbol{X}$ denote the streaming matrix and $\widetilde{\boldsymbol{X}}$ the subset of rows approximated by inactive blocks. Consider a matrix sketching algorithm ALG that achieves covariance error $\|\boldsymbol{X}^\top \boldsymbol{X} - \boldsymbol{S}^\top \boldsymbol{S}\|_2 \leq \xi \cdot \|\boldsymbol{X}\|_F^2$, where $\boldsymbol{S}$ is the sketch matrix and $\xi$ is a constant. Assume ALG requires $\ell_\xi$ rows and $\mu_\xi$ update time. The following theorem applies to *any* matrix sketching method satisfying this error guarantee.

**Theorem 1** (Dyadic Block Sketching Guarantee). *Given initial sketch size $l_0$, error parameter $\epsilon$, and a single-scale matrix sketching algorithm* ALG, *our method produces a sketch $\boldsymbol{S}$ satisfying*

$$\|\boldsymbol{X}^\top \boldsymbol{X} - \boldsymbol{S}^\top \boldsymbol{S}\|_2 \leq 2\epsilon. \tag{4}$$

*The space complexity is $O\left(d \cdot \sum_{i=0}^B \ell_{\frac{1}{2^i l_0}}\right)$ and the per-round update complexity is $O\left(\mu_{\frac{1}{2^B l_0}}\right)$, where $B = \left\lceil \min\left\{\log \frac{k}{l_0}, \frac{\|\widetilde{\boldsymbol{X}}\|_F^2}{\epsilon l_0}\right\} \right\rceil$ with $k = \mathrm{rank}(\boldsymbol{X})$.*

The detailed proof is provided in Appendix D.2. Theorem 1 establishes that the global error is constrained by parameter $\epsilon$, while the complexity depends on the choice of ALG and the number of blocks $B$. The value of $B$ grows adaptively during sketching and depends not only on the parameters $l_0$ and $\epsilon$, but also on the spectral properties of the streaming matrix (e.g., $k$ and $\|\widetilde{\boldsymbol{X}}\|_F^2$). To illustrate how different ALG yield different complexities, we present the following corollary for FD:

**Dyadic Block Sketching for FD.** We employ FD (Liberty, 2013) as ALG for each block in our method. With a given covariance error $\xi$, FD requires $\ell_\xi = O(1/\xi)$ rows and processes updates at an amortized cost of $\mu_\xi = O(d/\xi)$. The following corollary specializes Theorem 1 to FD:

**Corollary 1.** *Dyadic Block Sketching with FD guarantees the error bound in equation 4, with both space complexity and amortized update cost $O\left(dl_0 \cdot \min\{k/l_0, 2^{\|\widetilde{\boldsymbol{X}}\|_F^2/(\epsilon l_0)}\}\right)$.*

Our method provides a framework for constraining the global error of matrix approximation by integrating sketches across multiple scales. This mechanism is particularly critical for learning and optimization algorithms that impose strict accuracy requirements, which the single-scale matrix sketching method cannot always satisfy. More precisely, as shown in Theorem 1, the error of Dyadic Block Sketching is governed by a pre-specified parameter $\epsilon$, which can, in principle, be tuned to arbitrarily small values. When a stringent error tolerance is required, such as $\epsilon < \sigma_d^2/2$ where $\sigma_d$ is the smallest singular value of $\boldsymbol{X}$, even the largest FD with $l = d - 1$ cannot meet this constraint.

Furthermore, as the active block's sketch size grows dyadically, our method closely tracks the optimal rank-$k$ approximation. Once it exceeds $k$, the spectral error in the active block vanishes. Note that the full-rank case $k = d$ corresponds to the edge case that triggers rank-1 modifications. This behavior matches the first term in the complexity bound of Corollary 1 and shows that, in the low-rank regime, our method attains the optimal FD complexity of $O(dk)$.

**Remark 1** (Efficient Implementation). The update costs include calculating SVD to obtain $\boldsymbol{S}^{(t)}$ and performing matrix multiplication to compute $\boldsymbol{M}^{(t)}$, both of which cost $O(dl^2)$, where $l$ is the current sketch size. In implementation, the amortized cost can be reduced to $O(dl)$ either by doubling space (detailed in Appendix B.2), or by employing the Gu-Eisenstat procedure (Gu & Eisenstat, 1993).

## 4 APPLICATION TO LINEAR BANDITS

In this section, we incorporate Dyadic Block Sketching into linear bandits and propose a novel framework, termed DBSLinUCB. DBSLinUCB guarantees worst-case sublinear regret, independent of the streaming matrix, and readily extends to other sketch-based approaches.

### 4.1 ALGORITHM AND REGRET GUARANTEE

We use FD as the base algorithm in Dyadic Block Sketching. First, we present the estimator utilized in our approach, followed by the derivation of the confidence ellipsoid, which is essential for both the algorithm design and the regret analysis. Using the upper confidence bound, we then propose a selection criterion. Finally, we provide the theoretical guarantee on the regret of our method.

**Estimator.** We adopt a sketch-based RLS estimator, similar to previous work (Kuzborskij et al., 2019; Chen et al., 2020), with the key difference that we use Algorithm 1 to generate the sketch. Let $\boldsymbol{X}^{(t)} = [\boldsymbol{x}_1^\top, \ldots, \boldsymbol{x}_t^\top]^\top \in \mathbb{R}^{t \times d}$ denote the matrix containing all the arms selected up to round $t$. We utilize the sketch matrix $\boldsymbol{S}^{(t)} \in \mathbb{R}^{l_{B_t} \times d}$ and $\boldsymbol{M}^{(t)}$ to approximate $\boldsymbol{X}^{(t)}$, where $l_{B_t}$ is the current sketch size. The sketched RLS estimator is given by

$$\widehat{\boldsymbol{\theta}}_t = \left(\widehat{\boldsymbol{A}}^{(t)}\right)^{-1} \sum_{s=1}^t r_s \boldsymbol{x}_s, \quad \left(\widehat{\boldsymbol{A}}^{(t)}\right)^{-1} = \frac{1}{\lambda}\left(\boldsymbol{I}_d - (\boldsymbol{S}^{(t)})^\top \boldsymbol{M}^{(t)} \boldsymbol{S}^{(t)}\right). \tag{5}$$

**Confidence Ellipsoid.** For the estimator equation 5, we derive the corresponding confidence ellipsoid, which is a key component in achieving sublinear regret in the worst case.

**Theorem 2** (Multi-scale sketched confidence ellipsoid). *Following the assumption of linear bandits in section 2.2. Let $B_t$ be the number of blocks at round $t$. For any $\delta \in (0, 1)$, the optimal weight $\boldsymbol{\theta}_\star$ belongs to the set $\Theta_t \equiv \{\boldsymbol{\theta} \in \mathbb{R}^d : \|\boldsymbol{\theta} - \widehat{\boldsymbol{\theta}}_t\|_{\widehat{\boldsymbol{A}}^{(t)}} \leq \widehat{\beta}_t(\delta)\}$ with probability at least $1 - \delta$, where*

$$\widehat{\beta}_t(\delta) \lesssim R\sqrt{d \ln\left(1 + \frac{\epsilon}{\lambda}\right) + 2l_{B_t}} \cdot \sqrt{1 + \frac{\epsilon}{\lambda}} + \frac{H(\lambda + \epsilon)}{\sqrt{\lambda}}.$$

The proof is provided in Appendix E.1. Importantly, this result departs from the previous single-scale sketched one ((Kuzborskij et al., 2019), Theorem 2). Here, the ellipsoid is constructed by leveraging multiple sketches at different scales. We can then define the selection criterion as

$$\boldsymbol{x}_t = \underset{\boldsymbol{x} \in \mathcal{X}_t}{\arg\max}\left\{\boldsymbol{x}^\top \widehat{\boldsymbol{\theta}}_{t-1} + \widehat{\beta}_{t-1}(\delta) \cdot \|\boldsymbol{x}\|_{(\widehat{\boldsymbol{A}}^{(t-1)})^{-1}}\right\}. \tag{6}$$

**Complexity.** The overall algorithm is summarized in Algorithm 5. The computational cost arises from three components: updating the sketch via Algorithm 1, computing the sketched RLS estimator in equation 5, and performing arm selection in equation 6. Let $l_{B_t}$ be the sketch size of the active block at round $t$. Since both equation 5 and equation 6 require computing $(\widehat{\boldsymbol{A}}^{(t)})^{-1}$, we can employ the sketch-based acceleration discussed in Section 2.3, which costs $O(dl_{B_T} + l_{B_T}^2)$. Combined with the sketch maintenance cost from Corollary 1, we obtain a total space complexity of $O(dl_{B_T})$ and an amortized update complexity of $O(dl_{B_T} + l_{B_T}^2)$, where $l_{B_T} = \min\left\{k, l_0 \cdot 2^{\|\widetilde{\boldsymbol{X}}\|_F^2/(\epsilon l_0)}\right\}$.

**Regret Bound.** We demonstrate that the regret bound of DBSLinUCB using FD is as follows:

**Theorem 3** (Regret bound of DBSLinUCB-FD). *Consider the basic assumptions of linear bandits outlined in Section 2.2, and assume that $L \geq \sqrt{\lambda}$. Let the sketch size of the active block at round $t$ be denoted by $l_{B_t} \leq d$. Given the error parameter $\epsilon$, the regret of DBSLinUCB-FD satisfies the following bound with probability at least $1 - 1/T$:*

$$\text{Regret}_T = \widetilde{O}\left(\left(1 + \frac{\epsilon}{\lambda}\right)^{\frac{3}{2}} \cdot (d + l_{B_T}) \cdot \sqrt{T}\right),$$

where the constants and logarithmic terms are omitted for brevity. The detailed proof and a concrete upper bound are provided in Appendix E.2. We now provide a detailed comparison of our method with the single-scale sketch-based method SOFUL and the non-sketched method OFUL.

SOFUL utilizes FD with a fixed sketch size, achieving $\widetilde{O}((1 + \Delta_T)^{3/2}\sqrt{T})$ regret bound. As discussed in Section 2.3, this dependence on the spectral error $\Delta_T$ introduces a fundamental vulnerability that can lead to linear regret. In contrast, DBSLinUCB achieves a $\widetilde{O}(\epsilon^{3/2}\sqrt{T})$ regret bound, which reduces to $\widetilde{O}(\sqrt{T})$ when setting $\epsilon = O(1)$, matching that of the slower, non-sketched counterpart. Crucially, DBSLinUCB dynamically adjusts its sketch size $l_{B_t}$ based on the observed data, with the final size $l_{B_T} = \min\left\{k, l_0 \cdot 2^{\|\widetilde{X}\|_F^2/(\epsilon l_0)}\right\}$ determined by the streaming matrix's properties.

Given a target order of regret bound, such as $\widetilde{O}(\sqrt{T})$, our method can recover the complexity of SOFUL or OFUL across different streaming data environments. For simplicity, we set $l_0 = 1$. When the streaming matrix exhibits favorable spectral properties (e.g., low rank $k$), the optimal complexity for SOFUL is $O(dk)$ since this yields $\Delta_T = 0$. In this case, by setting $\epsilon < \|\widetilde{X}\|_F^2 / \log k$, DBSLinUCB achieves the same regret bound and $O(dk)$ complexity as SOFUL, differing only by constant factors. Conversely, when the streaming matrix exhibits a heavy spectral tail with full rank $k = d$, DBSLinUCB adaptively performs rank-1 modifications after a certain number of rounds, effectively degenerating to the non-sketched method OFUL with complexity $O(d^2)$. In this scenario, our regret bound differs from OFUL's regret bound only by a constant factor of $\epsilon^{3/2}$.

We therefore view our work as analyzing the trade-off between regret utility and sketching efficiency under streaming matrices with unknown spectral properties. SOFUL and OFUL represent the two extremes of this trade-off: SOFUL is tailored for matrices with favorable characteristics, while OFUL is more effective in scenarios with unsatisfactory properties. DBSLinUCB, positioned between these extremes, provides a flexible solution that can swing to both ends of the spectrum and generalizes to a wide range of scenarios. In practical applications, SOFUL is better suited for environments with strict cost constraints, such as microcontrollers (Lin et al., 2023), whereas DBSLinUCB excels in settings where maximizing efficiency while maintaining accuracy is essential, such as in large-scale online recommendation systems (Zhang et al., 2022).

**Remark 2** (Practical Guidance of Parameters). The parameters of DBSLinUCB include the error parameter $\epsilon$ and the initial sketch size $l_0$. The regret bound in Theorem 3 is controlled by $\epsilon$, which allows us to obtain regret bounds of different orders by adjusting its value. In particular, setting $\epsilon$ as a small constant yields the non-sketched $\widetilde{O}(\sqrt{T})$ regret bound, as discussed above. More generally, by choosing $\epsilon = O\left(T^{\frac{2\gamma-1}{3}}\right)$, one can achieve an arbitrary sublinear regret bound $O(T^\gamma)$ for any $\gamma \in [0.5, 1)$. For $l_0$, we recommend selecting a value substantially smaller than $d$ in the absence of prior knowledge about the streaming matrix, thereby ensuring sufficient exploration. In practice, if prior information about the effective dimensionality is available (e.g., an estimate $\widehat{l}$), $l_0$ can be chosen as a fraction of $\widehat{l}$ scaled by a constant. The parameter tuning results are provided in Section 5.

## 4.2 EXPAND TO OTHER MATRIX SKETCHING METHODS

DBSLinUCB provides a scalable framework for efficient sketch-based linear bandits, capable of incorporating various matrix sketching methods. Robust Frequent Directions (RFD) is another matrix sketching technique developed to address the rank deficiency issue inherent in FD. It has been proven to be an ideal sketching method for sequential decision-making problems (Luo et al., 2019; Chen et al., 2020; Feinberg et al., 2023). We use the RFD (see Algorithm 3) as the sketching method in Algorithm 1 and derive the following regret bound. The proof is provided in Appendix E.3.

**Theorem 4** (Regret Bound of DBSLinUCB-RFD). *Consider the basic assumptions of linear bandits outlined in Section 2.2, and assume that $L \geq \sqrt{\lambda}$. Let the sketch size of the active block at round $t$ be denoted by $l_{B_t} \leq d$. Given the error parameter $\epsilon$, the regret of DBSLinUCB-RFD satisfies the following bound with probability at least $1 - 1/T$:*

$$\text{Regret}_T = \widetilde{O}\left(\left(1 + \frac{\epsilon}{\lambda}\right)^{\frac{1}{2}} \cdot \sqrt{l_{B_T}T} + \sqrt{dl_{B_T}T}\right).$$

Compared to Theorem 3, Theorem 4 reduces the order of $\epsilon$ from $3/2$ to $1/2$. Apart from logarithmic terms, decoupling $d$ and $\epsilon$ further mitigates the impact of $\epsilon$. Since RFD and FD yield identical error bounds with the same sketch size, DBSLinUCB-FD and DBSLinUCB-RFD share the same complexity expression, though their hidden constants may differ due to algorithmic details. Replacing FD with RFD is straightforward, but its theoretical analysis is non-trivial. The improved regret bound stems from two key properties: positive definite monotonicity and well-conditioning, both of which are demonstrated under decomposability in Appendix E.4. This is, to our knowledge, the first result to establish these properties within the context of multi-scale sketching.

## 5 EXPERIMENTS

In this section, we evaluate the performance of DBSLinUCB on the synthetic dataset and several real-world datasets. The baselines include the non-sketched method OFUL (Abbasi-Yadkori et al., 2011) and the sketch-based methods SOFUL (Kuzborskij et al., 2019), CBSCFD (Chen et al., 2020). All sketch-based methods employed the efficient implementations described in Remark 1. The experimental setting, additional experiments, and configurations are available in Appendix F.

**Online Regression in Synthetic Data.** Inspired by the experimental settings in (Chen et al., 2020), we build synthetic datasets using multivariate Gaussian distributions $\mathcal{N}(\mathbf{0}, \mathbf{I}_d)$ with 100 arms and $d = 500$ features per context. The true parameter $\theta_\star$ is drawn from $\mathcal{N}(\mathbf{0}, \mathbf{I}_d)$ and is normalized. We set the sketch size $l \in \{50, 450\}$ for SOFUL and CBSCFD and the initial sketch size $l_0 = 50$ for DBSLinUCB. We set the error parameter $\epsilon = 8$ for DBSLinUCB.

The experimental results presented in Figures 1a and 1b (Section 1) demonstrate that DBSLinUCB, using both FD and RFD, consistently outperforms corresponding single-scale sketch-based methods. Notably, when $l = 50$, both SOFUL and CBSCFD show significantly worse performance compared to DBSLinUCB, exhibiting nearly linear regret. In Figure 1c, we report the trajectory of the spectral error term $\log(\Delta_T)/\log t$ over round $t$. We observe that for insufficient sketch sizes ($l = 50, 200$), this term crosses the benchmark line of $y = 1/3$, indicating that excessive spectral error leads to linear regret, which aligns with our theoretical analysis. We also evaluate the performance of our method in terms of matrix approximation in Appendix F.2, showing its ability to limit spectral error.

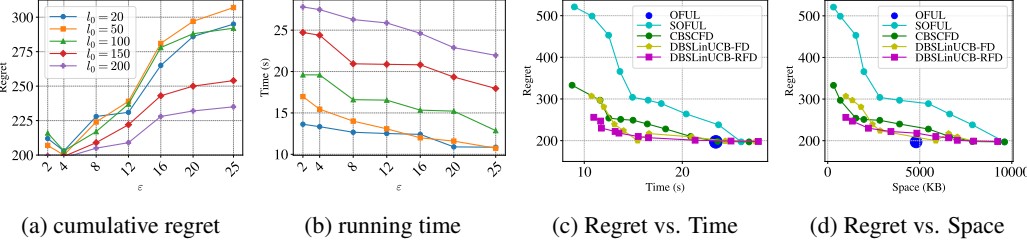

| (a) cumulative regret | (b) running time | (c) Regret vs. Time | (d) Regret vs. Space |

Figure 3: (a), (b): Cumulative regret and total running time of DBSLinUCB w.r.t. error parameter $\epsilon$ on `MNIST`; (c), (d): Pareto frontiers for regret vs. time and regret vs. space on `MNIST`, illustrating the utility-efficiency trade-off between the proposed DBSLinUCB and the compared methods.

**Online Classification in Real-world Data.** We perform online classification on the real-world dataset `MNIST`. The dataset contains $60,000$ samples, each with $d = 784$ features, and there are $M = 10$ possible labels. We follow the setup in (Kuzborskij et al., 2019), details in Appendix F.1. We first investigate the impact of $\epsilon$ and the initial sketch size $l_0$ on the performance of DBSLinUCB. Figures 3a and 3b present the cumulative regret and total running time after 2000 rounds. Our results

indicate that larger values of $\epsilon$ lead to increased regret but improved computational efficiency. With respect to $l_0$, we observe that larger values yield better regret at the cost of increased runtime. An interesting observation is that when $\epsilon$ is very small, the regret across different sketch sizes becomes nearly identical. This phenomenon can be attributed to Invariant 2, which constrains the number of sketching blocks. Under a small $\epsilon$, the algorithm tends to sketch fewer rows and relies more heavily on non-sketched updates, thereby diminishing the influence of $l_0$ on overall performance. Furthermore, Figure 3a shows that for relatively small values of $\epsilon$ (e.g., $\epsilon = 2, 4$), the actual regret is not necessarily monotonically increasing. This is because $\epsilon$ constrains the upper bound of the matrix approximation error, while the tightness of this bound may vary with different parameter choices.

We then compare DBSLinUCB variants against OFUL, SOFUL, and CBSCFD. For baseline methods, we vary sketch size $l \in [10, 600]$ across 10 equally-spaced points; for DBSLinUCB, we evaluate 10 configurations with $\epsilon \in [2, 25]$ and $l_0 \in [50, 200]$. Figures 3c and 3d present the Pareto frontiers for regret-efficiency trade-offs. DBSLinUCB demonstrates superior performance across both dimensions: it consistently dominates SOFUL with up to $40\%$ regret reduction at comparable resource usage, while DBSLinUCB-RFD outperforms CBSCFD across nearly the entire Pareto frontier. Notably, our method approaches OFUL's optimal regret ($\approx 200$) while achieving $60\%$ time and $80\%$ space reduction in certain configurations. A key advantage of DBSLinUCB is its regret-robustness, which consistently maintains regret below 300, whereas single-scale methods like SOFUL exceed 500 under an insufficient sketch size. Additional experimental results on MNIST and other real-world datasets are provided in Appendices F.3 and F.4.

## 6 CONCLUSION

This paper addresses the current pitfall of linear regret in sketch-based linear bandits for the first time. We propose Dyadic Block Sketching with a constrained global error bound and provide formal theoretical guarantees. By leveraging Dyadic Block Sketching, we present a framework for efficient sketch-based linear bandits. Even in the worst-case scenario, our method can achieve sublinear regret without prior knowledge of the streaming matrix. The experimental evaluations conducted on both real and synthetic datasets underscore the superior performance of our method.

Our multi-scale sketching scheduling can also be applied to more challenging one-pass bandit settings, such as linear bandits with heavy-tailed noise (Shao et al., 2018; Wang et al., 2025) and generalized linear bandits (Filippi et al., 2010; Zhang et al., 2025), where the interplay between sketching error and model misspecification introduces additional technical challenges. Moreover, it would be promising to extend the Dyadic Block Sketching paradigm to broader online optimization problems (Luo et al., 2016; Zhang et al., 2023; Feinberg et al., 2023), to control worst-case regret under streaming constraints. There is one important open question left on how to leverage the adaptive spectral error bound of FD to achieve a better allocation strategy for multi-scale sketches (Liberty, 2013). Our current analysis employs the Frobenius-based additive error bound, as the Frobenius norm can be tracked incrementally in the streaming setting. Exploiting the tighter, data-dependent spectral tail bound could enable more aggressive sketch size scheduling and improved complexity-regret trade-offs. Finally, it would be greatly important to further understand the minimal complexity required to achieve a given regret guarantee without prior knowledge.

## ACKNOWLEDGMENTS

The work was partially done at Gaoling School of Artificial Intelligence, Beijing Key Laboratory of Research on Large Models and Intelligent Governance, Engineering Research Center of Next-Generation Intelligent Search and Recommendation, MOE, and Pazhou Laboratory (Huangpu), Guangzhou, Guangdong 510555, China. This research was supported in part by National Science and Technology Major Project (2022ZD0114802), by National Natural Science Foundation of China (No. U2241212, No. 92470128, No. 62376275). We also wish to acknowledge the support provided by the fund for building world-class universities (disciplines) of Renmin University of China, by Engineering Research Center of Next-Generation Intelligent Search and Recommendation, Ministry of Education, by Intelligent Social Governance Interdisciplinary Platform, Major Innovation & Planning Interdisciplinary Platform for the "Double-First Class" Initiative, Public Policy and Decision-making Research Lab, and Public Computing Cloud, Renmin University of China.

## ETHICS STATEMENT

This work studies algorithmic methods for efficient linear bandits and evaluates them on synthetic data and the public MNIST dataset. It does not involve human subjects, personally identifiable information, or sensitive attributes; experiments use non-identifiable benchmarks and simulated data only. We therefore do not foresee direct risks to privacy or safety. Potential downstream uses (e.g., recommendation or allocation) could amplify biases present in third-party data; to mitigate this, we encourage practitioners to pair our sketching framework with careful dataset curation, bias monitoring, and domain-appropriate safeguards. Computational demands are modest (we report running times alongside regret to support resource transparency), limiting environmental impact. We affirm compliance with the ICLR Code of Ethics. Evidence of our experimental setup and datasets appears in the paper's experiments section (synthetic and MNIST) and running-time reporting.

## REPRODUCIBILITY STATEMENT

We facilitate reproducibility through: (i) full algorithmic descriptions and pseudocode for Dyadic Block Sketching and the underlying FD/RFD sketches; (ii) precise statements of assumptions and theorems with complete proofs placed in the appendices; and (iii) detailed experimental settings (data preprocessing, hyperparameters, and evaluation protocol) for both synthetic data and MNIST. Specifically, high-level algorithms and composition formulas are given in the main method section; FD/RFD pseudocode is provided in Appendix B.1; and the paper states that all proofs are included in the appendices, with experimental details collected in Appendix F. In our submission, we include an anonymized artifact implementing DBSLinUCB with FD/RFD, configuration files, and seed control to reproduce all figures and tables. See method/algorithm details and composition, FD/RFD algorithms, statements regarding proofs in appendices, and experimental details.

## USE OF LARGE LANGUAGE MODELS (LLMS)

LLMs were used only as writing assistants for minor language polishing. They were not used for research ideation, mathematical derivations, proof development, or code generation. All technical contents were created and verified by the authors. The authors accept full responsibility for all content and have checked that no generated text constitutes plagiarism or factual fabrication.

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

# A    MORE RELATED WORKS

**Sketch-based Online Learning.**    Sketch-based online learning leverages probabilistic (e.g., random projections or sampling) and deterministic (e.g., Frequent Directions) matrix sketches to reduce per-update time and memory while preserving essential learning information. In Table 1, we summarize the theoretical results of state-of-the-art sketch-based linear bandit methods and compare them with our approach. Beyond the linear bandit setting, sketching has been employed to accelerate second-order online gradient methods (Luo et al., 2016; Feinberg et al., 2023), online kernel learning (Calandriello et al., 2017; Luo et al., 2019), stochastic optimization (Gonen et al., 2016), and contextual batched bandits (Zhang et al., 2023). Another closely related direction is CoRE-Learning (Zhou, 2024; Wang et al., 2024), which studies the interplay between resource allocation and learnability. In this sense, a sketch-size scheduling policy can be viewed as a special case of computational resource allocation during learning. To the best of our knowledge, existing work has almost exclusively relied on single-scale matrix sketching schemes, while systematic investigations of multi-scale matrix sketching scheduling in online learning remain largely absent.

Table 1: Comparison of regret bounds, complexities, and sketching methods for sketch-based linear bandit algorithms. Here $d$ denotes the dimension and $T$ the horizon. For single-scale methods, the sketch size $l$ is fixed. For multi-scale methods, the sketch size $l_{B_T} = \min\{k, 2^{\|\widetilde{X}\|_F^2/(\epsilon l_0)} \cdot l_0\}$ is dynamically adjusted, where $l_0$ is the initial sketch size and $k = \mathrm{rank}(X)$.

| Algorithms | Regret Bounds | Time | Space | Sketching Method |
|---|---|---|---|---|
| OFUL (Abbasi-Yadkori et al. (2011)) | $\widetilde{O}(d\sqrt{T})$ | $O(d^2)$ | $O(d^2)$ | - |
| CBRAP (Yu et al. (2017)) | $\widetilde{O}(\sqrt{lT} + T/\sqrt{l})$ | $O(dl + l^3)$ | $O(dl)$ | Random Projection |
| SOFUL (Kuzborskij et al. (2019)) | $\widetilde{O}((1+\Delta_T)^{3/2}(l + d\log(1+\Delta_T))\sqrt{T})$ | $O(dl)$ | $O(dl)$ | Single-scale FD |
| CBSCFD (Chen et al. (2020)) | $\widetilde{O}((\sqrt{l + d\log(1+\Delta_T)} + \sqrt{\Delta_T})\sqrt{lT})$ | $O(dl)$ | $O(dl)$ | Single-scale RFD |
| **DBSLinUCB-FD** (This work) | $\widetilde{O}((1+\epsilon)^{3/2}(d + l_{B_T})\sqrt{T})$ | $O(dl_{B_T})$ | $O(dl_{B_T})$ | Multi-scale FD |
| **DBSLinUCB-RFD** (This work) | $\widetilde{O}(\sqrt{(1+\epsilon)l_{B_T}T} + \sqrt{dl_{B_T}T})$ | $O(dl_{B_T})$ | $O(dl_{B_T})$ | Multi-scale RFD |

**Matrix Sketching.**    Matrix sketching algorithms are typically designed for the unbounded streaming model. In this framework, the algorithm receives rows of a matrix $A \in \mathbb{R}^{n \times d}$ sequentially over time. The objective is to maintain a matrix sketch structure that produces an approximation matrix $B \in \mathbb{R}^{l \times d}$ with only $l$ rows. The goal is to ensure that the covariance matrix approximation satisfies $B^\top B \approx A^\top A$, meaning that $B$ approximates $A$ well.

Streaming matrix sketching methods can be broadly categorized into three groups: The first approach is sampling a small subset of matrix rows or columns that approximates the entire matrix (Frieze et al., 2004; Deshpande & Rademacher, 2010). The second approach is randomly combining matrix rows via random projection. Several results are available in the literature, including random projections and hashing (Achlioptas, 2001; Sarlós, 2006). The third approach employs a deterministic matrix sketching technique proposed by Liberty (2013), which adapts the well-known MG algorithm from Misra & Gries (1982) (originally used for approximating item frequencies) to sketch a streaming matrix by tracking its frequent directions. For further details, we refer readers to the survey (Woodruff, 2014).

**Multi-scale Sketching.**    Maintaining multiple streaming sketches at different scales is beneficial for a variety of streaming problems and has been well-studied in the literature. For instance, Wang et al. (2013) employ a dyadic aggregation structure, expressing a range as a sum of a bounded number of estimated counts. Additionally, multi-scale sketching has been applied to problems such as the heavy-hitter problem (Larsen et al., 2019), the sliding-window problem (Wei et al., 2016; Yin et al., 2024), and persistent sketching (Wei et al., 2015; Zeng et al., 2022). We emphasize that Dyadic Block Sketching is fundamentally distinct from multi-scale sketching methods studied in streaming algorithms. Multi-scale sketches in classical streaming settings typically focus on capturing statistics over a restricted portion of the stream (e.g., a sliding window), and therefore provide relative, data-dependent error guarantees, much like their single-scale counterparts. In contrast, our method must accommodate the entire sequence of actions generated by the linear bandit process and aims to

guarantee worst-case regret. This yields a data-independent, absolute error bound at the global level, rather than a range-specific guarantee.

# B    OMITTED ALGORITHMS

In this section, we present the pseudo-code for deterministic matrix sketching (Appendix B.1), an efficient implementation of Dyadic Block Sketching (Appendix B.2), and the multi-scale sketched linear bandit framework DBSLinUCB (Appendix B.3).

## B.1    PSEUDO-CODE OF DETERMINISTIC MATRIX SKETCHING

Frequent Directions (FD) is a deterministic sketching method (Liberty, 2013; Ghashami et al., 2016). FD uniquely maintains the invariant that the last row of the sketch matrix, $S$, is always zero. In each round, a new row $x_t$ is inserted into this last row of $S$, and the matrix undergoes singular value decomposition into $U \Sigma V^\top$. Subsequently, $S$ is updated to $\sqrt{\Sigma_l^2 - \sigma I} \cdot V_l^\top$, where $\sigma$ represents the square of the $l$-th singular value. Given that the rows of $S$ are orthogonal, $M = (SS^\top + \lambda I)^{-1}$ remains a diagonal matrix, facilitating efficient maintenance.

The Robust Frequent Directions (RFD) [1] sketching technique is designed to tackle the problem of rank deficiency (Luo et al., 2019; Chen et al., 2020). RFD enhances the Frequent Directions (FD) method by maintaining a counter $\alpha$, which captures the spectral error. More precisely, RFD approximates $X^\top X$ by $S^\top S + \alpha I$. The error bound for RFD is equivalent to that of FD, as follows:

**Lemma 4** (Theorem 1 of Chen et al. (2020)). *Let $X^{(t)}$ be the streaming matrix at round $t$, and $S^{(t)}$ be the sketch matrix and $\alpha_t$ be the counter produced by RFD. Define the spectral error as*

$$\Delta_t := \min_{k < l} \frac{\left\| X^{(t)} - X^{(t)}_{[k]} \right\|_F^2}{l - k},$$

*where $X^{(t)}_{[k]}$ denotes the matrix consisting of the first $k$ singular vectors of $X^{(T)}$. Then, it holds that*

$$\left\| \left( S^{(t)} \right)^\top S^{(t)} + \alpha_t I - \left( X^{(t)} \right)^\top X^{(t)} \right\| \le \Delta_T.$$

---

**Algorithm 2** FD sketch

**Input:** Data $X \in \mathbb{R}^{T \times d}$, sketch size $l$, regularization $\lambda$
**Output:** Sketch $S$, $M$
Initialize $S \leftarrow \mathbf{0}^{l \times d}$, $M \leftarrow \frac{1}{\lambda} I_l$
**for** $t = 1$ **to** $T$ **do**
   Append $x_t$ to the last row of $S$
   Compute $[U, \Sigma, V] \leftarrow \text{svd}(S)$
   Set $\sigma \leftarrow \sigma_l^2$
   Update $S \leftarrow \sqrt{\Sigma_l^2 - \sigma I} \cdot V_l^\top$
   Update $M \leftarrow \text{diag}\left\{ \frac{1}{\lambda + \sigma_1^2 - \sigma}, ..., \frac{1}{\lambda} \right\}$
**end for**

---

**Algorithm 3** RFD sketch

**Input:** Data $X \in \mathbb{R}^{T \times d}$, sketch size $l$, regularization $\lambda$
**Output:** Sketch $S$, $M$ and counter $\alpha$
Initialize $S \leftarrow \mathbf{0}^{l \times d}$, $M \leftarrow \frac{1}{\lambda} I_l$, $\alpha \leftarrow 0$
**for** $t = 1$ **to** $T$ **do**
   Append $x_t$ to the last row of $S$
   Compute $[U, \Sigma, V] \leftarrow \text{svd}(S)$
   Set $\sigma \leftarrow \sigma_l^2$, $\alpha \leftarrow \alpha + \sigma$
   Update $S \leftarrow \sqrt{\Sigma_l^2 - \sigma I} \cdot V_l^\top$
   Set $M \leftarrow \text{diag}\left\{ \frac{1}{\lambda + \sigma_1^2 - \sigma + \alpha}, ..., \frac{1}{\lambda + \alpha} \right\}$
**end for**

---

The pseudocode for FD and RFD is given in Algorithms 2 and 3. Within our algorithm, we use a Boolean flag `willExcessSketch` to test the rank condition in Invariant 1; it is obtained by applying the update to a temporary copy of the sketch. However, when the number of rows currently stored in a block is smaller than its sketch size $l$ (or dimension $d$), this test is uninformative:

---

[1] In the linear bandit literature, this variant is typically referred to as SCFD (Algorithm 1 of Chen et al. (2020)), which corresponds to Algorithm 3 in our paper. For consistency, we refer to it as RFD throughout the paper.

---

**Algorithm 4** Fast Dyadic Block Sketching

---

**Input:** Data stream $\{\boldsymbol{x}_t\}_{t=1}^T$, sketch size $l_0$, error parameter $\epsilon$, regularization $\lambda$
**Output:** Sketch matrix $\boldsymbol{S}^{(t)}$, $\boldsymbol{M}^{(t)}$
 1: Initialize an empty list $\mathcal{L}$ and $\mathcal{B}^\star.sketch$
 2: Initialize $\mathcal{B}^\star.\text{BlockSize} = 0$, $\mathcal{B}^\star.\text{SketchSize} = l_0$
 3: **for** $t = 1$ **to** $T$ **do**
 4:     Receive $\boldsymbol{x}_t$
 5:     **if** $\text{length}(\mathcal{L}) \geq \lfloor \log{(d/l_0 + 1)} \rfloor - 1$ **then**
 6:         Update $\boldsymbol{S}^{(t)}$ with rank-1 modifications.
 7:     **else**
 8:         Query `willExcessSketch` from $\mathcal{B}^\star.sketch$ and $\boldsymbol{x}_t$
 9:         **if** $\mathcal{B}^\star.\text{BlockSize} + \|\boldsymbol{x}_t\|^2 < \epsilon \cdot l_0$ or `willExcessSketch` is *False* **then**
10:            Update $\mathcal{B}^\star.\text{BlockSize} += \|\boldsymbol{x}_t\|^2$
11:            Append $\boldsymbol{x}_t$ below $\mathcal{B}^\star.sketch$
12:            Query $\widetilde{\boldsymbol{S}}$, $\widetilde{\boldsymbol{M}}$ from the inactive blocks $\mathcal{L}$
13:            **if** $\mathcal{B}^\star.sketch$ have $2 \cdot \mathcal{B}^\star.\text{SketchSize}$ rows **then**
14:                Update $\mathcal{B}^\star.sketch$
15:                Query $\boldsymbol{S}^\star$, $\boldsymbol{M}^\star$ from $\mathcal{B}^\star.sketch$
16:                Compute $\boldsymbol{S}^{(t)}$ and $\boldsymbol{M}^{(t)}$ by equation 3
17:            **else**
18:                Combine $\boldsymbol{S}^{(t)} = \begin{pmatrix} \boldsymbol{S} \\ \boldsymbol{S}^\star \end{pmatrix}$
19:                Combine $\boldsymbol{M}^{(t)}$ by equation 7
20:            **end if**
21:         **else**
22:            Set $l = \mathcal{B}^\star.\text{SketchSize}$
23:            Mark $\mathcal{B}^\star$ as inactive and append it to $\mathcal{L}$
24:            Initialize a new empty $\mathcal{B}^\star.sketch$
25:            Set $\mathcal{B}^\star.\text{BlockSize} = 0$, $\mathcal{B}^\star.\text{SketchSize} = 2l$
26:            Append $\boldsymbol{x}_t$ below $\mathcal{B}^\star.sketch$ and update $\mathcal{B}^\star.\text{BlockSize} += \|\boldsymbol{x}_t\|^2$
27:            Query $\widetilde{\boldsymbol{S}}$, $\widetilde{\boldsymbol{M}}$ from the inactive blocks $\mathcal{L}$
28:            Combine $\boldsymbol{S}^{(t)} = \begin{pmatrix} \boldsymbol{S} \\ \boldsymbol{S}^\star \end{pmatrix}$
29:            Combine $\boldsymbol{M}^{(t)}$ by equation 7
30:         **end if**
31:     **end if**
32: **end for**

---

FD/RFD necessarily yields a shrinkage value $\sigma = 0$ but no compression would occur. In practice, we therefore skip the test in this regime and set `willExcessSketch` to *True* as a sentinel, so as to bypass a rank-check; the block is still updated via simple accumulation, and the block size threshold governs new block generation. Finally, these deterministic matrix sketching methods can be accelerated by doubling the sketch size. More details can be found in Appendix B.2.

### B.2   Fast Algorithm of Dyadic Block Sketching

The computational cost of both FD and RFD, as outlined in Algorithms 2 and 3, is primarily determined by the singular value decomposition operations. Specifically, SVD must be performed at every update, which results in an update complexity of $O(dl^2)$. However, this amortized update cost can be reduced to $O(dl)$ by doubling the sketch size, as discussed in several works (Liberty, 2013; Luo et al., 2016; Kuzborskij et al., 2019).

In Algorithm 1, at round $t$, with $B_t + 1$ blocks, let $l_i$ denote the sketch size of the $i$-th block. This results in an amortized time complexity of $O(dl_{B_t}^2)$, due to the standard SVD process in the active block. Additionally, the computation of $\boldsymbol{M}^{(t)}$ via matrix multiplication and inversion requires

---

**Algorithm 5** DBSLinUCB

---

**Input:** Data stream $\{\boldsymbol{x}_t\}_{t=1}^T$, sketch size $l_0$, error parameter $\epsilon$, regularization parameter $\lambda$, confidence parameter $\delta$

1: Initialize a Dyadic Block Sketching instance $\text{Sketch}(\boldsymbol{S}^{(0)}, \boldsymbol{M}^{(0)})$ with parameters $l_0, \lambda, \epsilon$
2: **for** $t = 1$ **to** $T$ **do**
3:   Get arm set $\mathcal{X}_t$
4:   Compute the confidence ellipsoid $\widehat{\beta}_{t-1}(\delta)$
5:   Select $\boldsymbol{x}_t = \underset{\boldsymbol{x} \in \mathcal{X}_t}{\arg\max} \left\{ \boldsymbol{x}^\top \widehat{\boldsymbol{\theta}}_{t-1} + \widehat{\beta}_{t-1}(\delta) \cdot \|\boldsymbol{x}\|_{(\widehat{\boldsymbol{A}}^{(t-1)})^{-1}} \right\}$
6:   Receive the reward $r_t$
7:   Update $\text{Sketch}(\boldsymbol{S}^{(t)}, \boldsymbol{M}^{(t)})$ with $\boldsymbol{x}_t$ by Algorithm 1
8:   Compute $(\widehat{\boldsymbol{A}}^{(t)})^{-1} = \frac{1}{\lambda}\left( \boldsymbol{I}_d - (\boldsymbol{S}^{(t)})^\top \boldsymbol{M}^{(t)} \boldsymbol{S}^{(t)} \right)$
9:   Compute $\widehat{\boldsymbol{\theta}}_t = (\widehat{\boldsymbol{A}}^{(t)})^{-1} \sum_{s=1}^t r_s \boldsymbol{x}_s$
10: **end for**

---

$O\left( \sum_{i=0}^{B_t-1} l_i \cdot l_{B_t} \cdot d + \left( \sum_{i=0}^{B_t} l_i \right)^3 \right) = O\left( dl_{B_t}^2 \right)$. Similarly, we can improve the efficiency of our Dyadic Block Sketching method by doubling the sketch size, as detailed in Algorithm 4.

We perform the SVD step only after adding $\mathcal{B}^\star$.SketchSize rows. Note that within each epoch where no updates occur, the construction of $\boldsymbol{M}^{(t)}$ can be formulated as

$$\boldsymbol{M}^{(t)} = \begin{pmatrix} \boldsymbol{M}^{(t-1)} + \frac{\phi\phi^\top}{\xi} & \frac{-\phi}{\xi} \\ \frac{-\phi^\top}{\xi} & \frac{1}{\xi} \end{pmatrix}, \tag{7}$$

where $\phi = \boldsymbol{M}^{(t-1)} \boldsymbol{S}^{(t-1)} \boldsymbol{x}_t^\top$ and $\xi = \boldsymbol{x}_t \boldsymbol{x}_t^\top - \boldsymbol{x}_t (\boldsymbol{S}^{(t-1)})^\top \phi + \alpha + \lambda$. When the sketching method is FD, $\alpha$ is set to 0; conversely, when the sketching method is RFD, $\alpha$ serves as the counter maintained in the RFD sketch.

Given that the size of $\boldsymbol{M}^{(t)}$ is at most twice of the $\mathcal{B}^\star$.SketchSize, the amortized computation time required for $\boldsymbol{M}^{(t)}$ is limited to $O\left( dl_{B_t} \right)$. Additionally, we perform the SVD only after every addition of $\mathcal{B}^\star$.SketchSize rows, reducing the amortized update time complexity to $O\left( dl_{B_t} \right)$.

### B.3 PSEUDO-CODE OF DBSLINUCB

We present the pseudo-code for DBSLinUCB in Algorithm 5. DBSLinUCB introduces an innovative framework for sketch-based linear bandits, leveraging the multi-scale sketching technique to compute the sketched covariance matrix. As demonstrated in Theorems 3 and 4, the regret bound of DBSLinUCB is parametrized by $\epsilon$, offering a regret guarantee that is both controllable and adjustable.

## C OMITTED DETAILS FOR SECTION 2

In this section, we provide the omitted details from Section 2. In Section C.1, we present the proof of Observation 1, which demonstrates that an insufficient sketch size will lead to linear regret. In Section C.2, we discuss how RFD-based linear bandits are also susceptible to linear regret.

### C.1 PROOF OF OBSERVATION 1

Observation 1 follows from Theorem 3.3 of Banerjee et al. (2023), which shows that in a locally convex arm space (defined in Definition 3.1 of Banerjee et al. (2023)) the design matrix generated by any linear bandit algorithm with expected $O(\sqrt{T})$ regret has a heavy spectral tail. For convenience, we restate:

**Theorem 5** (Theorem 3.3 of Banerjee et al. (2023)). *Let $\mathcal{X}$ be a locally convex arm space and let $\overline{\boldsymbol{G}}_T = \mathbb{E}\left[ \sum_{t=1}^T \boldsymbol{x}_t \boldsymbol{x}_t^\top \right]$ denote the expected design matrix. For any bandit algorithm with expected*

*regret at most $O(\sqrt{T})$, there exists $q \in (0, 1/2]$ such that*

$$\lambda_d(\overline{\boldsymbol{G}}_T) = \Omega(T^q).$$

The exponent $q$ depends on the geometry of $\mathcal{X}$ (e.g., how well the surface approximates a locally constant Hessian). Since SOFUL essentially sketches the OFUL design-matrix sequence, the spectral error term $\Delta_T$ in its regret bound inherits this growth.

Fix a sketch size $l$. By Lemma 1, choosing $k = l - 1$ gives

$$\Delta_T = \left\| \boldsymbol{X}^{(T)} - \boldsymbol{X}^{(T)}_{[l-1]} \right\|_F^2 = \sum_{i=l}^d \sigma_i^2(\boldsymbol{X}^{(T)}) \geq (d-l)\,\sigma_d^2(\boldsymbol{X}^{(T)}),$$

where $\sigma_i(\boldsymbol{X}^{(T)})$ are the singular values of $\boldsymbol{X}^{(T)}$ and $\sigma_d^2(\boldsymbol{X}^{(T)}) = \lambda_{\min}((\boldsymbol{X}^{(T)})^\top \boldsymbol{X}^{(T)})$. Taking expectation and using $\lambda_d(\overline{\boldsymbol{G}}_T) = \lambda_{\min}(\mathbb{E}[(\boldsymbol{X}^{(T)})^\top \boldsymbol{X}^{(T)}])$, we obtain

$$\mathbb{E}[\Delta_T] \geq (d-l)\,\lambda_d(\overline{\boldsymbol{G}}_T) = \Omega\big((d-l)\,T^q\big).$$

Lemma 2 gives

$$\mathrm{Regret}_T^{\texttt{SOFUL}} = \widetilde{O}\Big( \min\big\{ \Delta_T^{3/2}\sqrt{T}, T \big\} \Big).$$

Thus if $l < d - T^{\frac{1}{3}-q}$, then $\mathbb{E}[\Delta_T] \gtrsim T^{1/3}$ and the bound collapses to the trivial $O(T)$. In particular, when the unknown geometry constant $q \geq 1/3$, we have $\mathbb{E}[\Delta_T] \gtrsim T^{1/3}$ for any $l < d$, so SOFUL suffers linear regret even at the maximal sketch size $l = d - 1$.

## C.2 Linear Regret Pitfalls in RFD-based Methods

Beyond FD-based approaches such as SOFUL, algorithms based on Robust Frequent Directions (RFD) are also vulnerable to linear regret when the sketch size is insufficient. Similar to the discussion in Section 2.3, we recall the regret bound of CBSCFD:

**Lemma 5** (Theorem 2 of Chen et al. (2020)). *Let* $\mathrm{Regret}_T^{\texttt{CBSCFD}}$ *denote the regret of CBSCFD with sketch size $l$ and spectral error $\Delta_T$ defined in equation 1. With high probability,*

$$\mathrm{Regret}_T^{\texttt{CBSCFD}} = \widetilde{O}\Big( \min\big\{ \big(\sqrt{l + d\log(1+\Delta_T)} + \sqrt{\Delta_T}\big)\sqrt{lT}, T \big\} \Big).$$

Although CBSCFD reduces the dependence on $\Delta_T$ compared to FD-based methods, it still degenerates to linear regret whenever $\Delta_T = \Omega(T)$. In particular, if the sketch size is chosen without knowledge of the spectral properties of the streaming matrix, this risk cannot be avoided:

**Observation 2.** *Let $\mathcal{X}$ be a locally convex arm space. If the sketch size of CBSCFD satisfies $l < d - T^{1-q}$, then CBSCFD incurs vacuous linear regret.*

This highlights a fundamental limitation of single-scale sketching: any fixed sketch size inevitably ties the spectral error $\Delta_T$ to the horizon $T$, making linear regret unavoidable when the spectral structure of the data is unknown in advance.

# D Omitted Proofs for Section 3

In this section, we provide the omitted proofs for Section 3. In Appendix D.1, we prove Lemma 3 of decomposability, which abstracts the key idea of multi-scale matrix sketching. Later, the proof of Theorem 1 is provided in Appendix D.2.

## D.1 Proof of Lemma 3

Since we have $\boldsymbol{X}^\top \boldsymbol{X} = \sum_{i=1}^p \boldsymbol{X}_i^\top \boldsymbol{X}_i$ and $\boldsymbol{S}^\top \boldsymbol{S} = \sum_{i=1}^p \boldsymbol{S}_i^\top \boldsymbol{S}_i$. Therefore

$$\left\| \boldsymbol{X}^\top \boldsymbol{X} - \boldsymbol{S}^\top \boldsymbol{S} \right\|_2 \leq \sum_{i=1}^p \left\| \boldsymbol{X}_i^\top \boldsymbol{X}_i - \boldsymbol{S}_i^\top \boldsymbol{S}_i \right\|_2 \leq \sum_{i=1}^p \epsilon_i \cdot \left\| \boldsymbol{X}_i \right\|_F^2,$$

and the Lemma follows.

### D.2 Proof of Theorem 1

We begin by analyzing the number of blocks of Dyadic Block Sketching. Let the stream of rows form a matrix $\boldsymbol{X} \in \mathbb{R}^{n \times d}$. Dyadic Block Sketching partitions the stream into contiguous blocks and maintains a dyadically growing sketch size. Concretely, let

$$\boldsymbol{X}^\top = \big[\, \boldsymbol{X}_0^\top, \boldsymbol{X}_1^\top, \ldots, \boldsymbol{X}_B^\top \,\big],$$

where block $i$ contains $n_i$ rows ($\boldsymbol{X}_i \in \mathbb{R}^{n_i \times d}$) and stores a sketch $\boldsymbol{S}_i$. We call blocks $0, 1, \ldots, B-1$ inactive and the last block $B$ active. In the setting of matrix sketching, low–rank detection is only evaluated once a block has accumulated at least $d$ rows; in particular, whenever detection is triggered, the corresponding block satisfies $n_i \geq d$. We analyze the following two cases:

When the streaming matrix is low-rank, i.e., $\mathrm{rank}(\boldsymbol{X}) = k$. By Invariant 1, once the sketch size exceeds the rank, shrinkage vanishes and the block remains low–rank (i.e., the sketch $\boldsymbol{S}_B$ tracks rank $k$). Equivalently, the last active level $B$ is the unique integer such that

$$2^{B-1} l_0 \leq k < 2^B l_0,$$

which yields the tight bounds

$$\log\Big(\frac{k}{l_0}\Big) \leq B < \log\Big(\frac{k}{l_0}\Big) + 1 \quad \Longrightarrow \quad B = \big\lceil \log(k/l_0) \big\rceil.$$

When the streaming matrix is full-rank, the above derivation shows that the rank-1 modification is triggered at block index $B = \lceil \log(d/l_0) \rceil$, which represents the maximum number of blocks possible in any case. However, when the row norms of the streaming matrix are small or the error parameter $\epsilon$ is relatively large, the actual number of blocks can be strictly smaller.

Assume that the maximum row norm is bounded by $\|\boldsymbol{x}\|_2^2 \leq L \ll \epsilon l_0$. By Invariant 1, we have

$$\epsilon l_0 - L \leq \|\boldsymbol{X}_i\|_F^2 \leq \epsilon l_0, \qquad i = 0, 1, \ldots, B-1.$$

Let $\widetilde{\boldsymbol{X}} = \big[\boldsymbol{X}_0^\top, \boldsymbol{X}_1^\top, \ldots, \boldsymbol{X}_{B-1}^\top\big]^\top$ collect all rows summarized by inactive blocks. Summing the above bounds yields

$$B\,(\epsilon l_0 - L) \leq \|\widetilde{\boldsymbol{X}}\|_F^2 \leq B\,\epsilon l_0 \quad \Longrightarrow \quad \frac{\|\widetilde{\boldsymbol{X}}\|_F^2}{\epsilon l_0} \leq B \leq \frac{\|\widetilde{\boldsymbol{X}}\|_F^2}{\epsilon l_0 - L},$$

and therefore we can approximate this case by $B = \big\lceil \|\widetilde{\boldsymbol{X}}\|_F^2 / (\epsilon l_0) \big\rceil$. Combining the low-rank and full-rank scenarios, the number of blocks is therefore given by

$$B = \left\lceil \min\left\{ \log\frac{k}{l_0}, \frac{\|\widetilde{\boldsymbol{X}}\|_F^2}{\epsilon l_0} \right\} \right\rceil.$$

For the error guarantee, we bound the global error by exploiting the decomposability of blockwise matrix sketches, as shown in Lemma 3. Without loss of generality, assume the streaming matrix has rank $k$. Partition the block indices into

$$\mathcal{I} = \big\{\, i \in \{0, \ldots, B-1\} : \|\boldsymbol{X}_i\|_F^2 \leq \epsilon l_0 \,\big\} \quad \text{(inactive/approximate blocks)}$$

and

$$\mathcal{E} = \big\{\, i \in \{0, \ldots, B\} : \text{sketch size} \geq \mathrm{rank}(\boldsymbol{X}_i) \,\big\} \quad \text{(exact-capture blocks)}.$$

In particular, by design of the dyadic growth, the sketch size of last active block $B$ alway larger than $k$, hence $B \in \mathcal{E}$. For every $i \in \mathcal{E}$ the sketch is exact in the sense that

$$\big\| \boldsymbol{X}_i^\top \boldsymbol{X}_i - \boldsymbol{S}_i^\top \boldsymbol{S}_i \big\|_2 = 0,$$

since once the sketch size exceeds the rank of block, the best rank-$k$ (or local rank) approximation is captured exactly. For $i \in \mathcal{I}$, Invariant 1 guarantees $\|\boldsymbol{X}_i\|_F^2 \leq \epsilon l_0$, and the $i$-th block employs a streaming sketch with error parameter $1/(2^i l_0)$, implying the per-block spectral error bound

$$\big\| \boldsymbol{X}_i^\top \boldsymbol{X}_i - \boldsymbol{S}_i^\top \boldsymbol{S}_i \big\|_2 \leq \frac{1}{2^i l_0} \|\boldsymbol{X}_i\|_F^2 \leq \frac{1}{2^i l_0}\,(\epsilon l_0) = \frac{\epsilon}{2^i}.$$

Let
$$\boldsymbol{S}^\top = \left[\boldsymbol{S}_0^\top, \boldsymbol{S}_1^\top, \ldots, \boldsymbol{S}_B^\top\right]$$
be the concatenated sketch used to approximate $\boldsymbol{X}$. By Lemma 3, we have

$$
\begin{aligned}
\left\|\boldsymbol{X}^\top \boldsymbol{X} - \boldsymbol{S}^\top \boldsymbol{S}\right\|_2 &\le \sum_{i=0}^{B} \left\|\boldsymbol{X}_i^\top \boldsymbol{X}_i - \boldsymbol{S}_i^\top \boldsymbol{S}_i\right\|_2 \\
&= \sum_{i \in \mathcal{I}} \left\|\boldsymbol{X}_i^\top \boldsymbol{X}_i - \boldsymbol{S}_i^\top \boldsymbol{S}_i\right\|_2 + \sum_{i \in \mathcal{E}} 0 \le \sum_{i \in \mathcal{I}} \frac{\epsilon}{2^i} \le 2\epsilon.
\end{aligned}
$$

Hence, the global spectral error is bounded by $2\epsilon$.

For space complexity, note that the $i$-th block employs a streaming matrix sketch with error parameter $1/(2^i l_0)$, which requires a sketch size of $\ell_{1/(2^i l_0)}$. Hence, the total number of rows stored across all sketches is

$$\sum_{i=0}^{B} \ell_{1/(2^i l_0)},$$

and the overall space requirement is

$$O\left(d \cdot \sum_{i=0}^{B} \ell_{1/(2^i l_0)}\right).$$

For update complexity, only the active block needs to be updated at each step. Consequently, the per-update cost is determined solely by the sketch size of the active block, leading to $O\left(\mu_{1/(2^B l_0)}\right)$.

## E    OMITTED PROOFS FOR SECTION 4

In this section, we first provide a proof of Theorem 2 in Appendix E.1, which is the key theorem leading to the regret bound of our method when using FD. The proofs for the regret bounds in Theorem 3 and Theorem 4 are provided in Appendix E.2 and Appendix E.3, respectively. Later, we illustrate and prove the properties of Dyadic Block Sketching for RFD in Appendix E.4, which are consistent with the properties of single-scale RFD.

Our key technical contribution is extending the theoretical framework (Kuzborskij et al., 2019; Chen et al., 2020) from single-scale to multi-scale matrix sketching. In particular, we show that the RLS estimator can effectively leverage multiple sketches at different scales, where the estimation error depends on their collective approximation quality. This analytical framework bridges the theoretical gap between multi-scale sketching and linear bandit algorithms, opening avenues for applying multi-scale techniques to other online learning problems.

### E.1    PROOF OF THEOREM 2

Denote $B_t$ as the number of blocks at round $t$, and $\overline{\sigma}_i$ as the sum of shrinking singular values in the sketch of block $i$. Let $l_{B_t}$ be the sketch size in the active block at round $t$. According to Algorithm 5, the approximate covariance matrix is

$$\widehat{\boldsymbol{A}}^{(t)} = \lambda \boldsymbol{I} + \sum_{i=1}^{B_t} \left(\boldsymbol{S}_i^{(t)}\right)^\top \boldsymbol{S}_i^{(t)},$$

where $\boldsymbol{S}_i^{(t)}$ is the sketch matrix in block $i$ at round $t$. Define $\boldsymbol{\eta}_1^\top, \ldots, \boldsymbol{\eta}_t^\top \in \mathbb{R}^d$ is the noise sequence conditionally R-subgaussian for a fixed constant $R$ and $\boldsymbol{r}_t^\top = (r_1, r_2, \ldots, r_t) \in \mathbb{R}^d$ is the reward vector. We begin by noticing that

$$\widehat{\boldsymbol{\theta}}_t = \left(\widehat{\boldsymbol{A}}^{(t)}\right)^{-1} \boldsymbol{X}_t^\top \boldsymbol{r}_t = \left(\widehat{\boldsymbol{A}}^{(t)}\right)^{-1} \boldsymbol{X}_t^\top \left(\boldsymbol{X}_t \boldsymbol{\theta}_\star + \boldsymbol{\eta}_t\right).$$

Therefore, we can decompose $\left\|\widehat{\boldsymbol{\theta}}_t - \boldsymbol{\theta}_\star\right\|^2_{\widehat{\boldsymbol{A}}^{(t)}}$ into two parts as follows

$$
\begin{aligned}
&\left\|\widehat{\boldsymbol{\theta}}_t - \boldsymbol{\theta}_\star\right\|^2_{\widehat{\boldsymbol{A}}^{(t)}} \\
&= \left(\widehat{\boldsymbol{\theta}}_t - \boldsymbol{\theta}_\star\right)^\top \widehat{\boldsymbol{A}}^{(t)} \left(\widehat{\boldsymbol{\theta}}_t - \boldsymbol{\theta}_\star\right) \\
&= \left(\widehat{\boldsymbol{\theta}}_t - \boldsymbol{\theta}_\star\right)^\top \widehat{\boldsymbol{A}}^{(t)} \left(\left(\widehat{\boldsymbol{A}}^{(t)}\right)^{-1} \boldsymbol{X}_t^\top \left(\boldsymbol{X}_t \boldsymbol{\theta}_\star + \boldsymbol{\eta}_t\right) - \boldsymbol{\theta}_\star\right) \\
&= \underbrace{\left(\widehat{\boldsymbol{\theta}}_t - \boldsymbol{\theta}_\star\right)^\top \widehat{\boldsymbol{A}}^{(t)} \left(\left(\widehat{\boldsymbol{A}}^{(t)}\right)^{-1} \boldsymbol{X}_t^\top \boldsymbol{X}_t \boldsymbol{\theta}_\star - \boldsymbol{\theta}_\star\right)}_{\text{Term 1: Bias Error}} + \underbrace{\left(\widehat{\boldsymbol{\theta}}_t - \boldsymbol{\theta}_\star\right)^\top \boldsymbol{X}_t^\top \boldsymbol{\eta}_t}_{\text{Term 2: Variance Error}}.
\end{aligned}
\tag{8}
$$

**Bounding the bias error.** We first focus on bounding the first term. We have that

$$
\begin{aligned}
&\left(\widehat{\boldsymbol{\theta}}_t - \boldsymbol{\theta}_\star\right)^\top \widehat{\boldsymbol{A}}^{(t)} \left(\left(\widehat{\boldsymbol{A}}^{(t)}\right)^{-1} \boldsymbol{X}_t^\top \boldsymbol{X}_t \boldsymbol{\theta}_\star - \boldsymbol{\theta}_\star\right) \\
&= \left(\widehat{\boldsymbol{\theta}}_t - \boldsymbol{\theta}_\star\right)^\top \left(\widehat{\boldsymbol{A}}^{(t)}\right)^{\frac{1}{2}} \left(\widehat{\boldsymbol{A}}^{(t)}\right)^{-\frac{1}{2}} \left(\boldsymbol{X}_t^\top \boldsymbol{X}_t \boldsymbol{\theta}_\star - \widehat{\boldsymbol{A}}^{(t)} \boldsymbol{\theta}_\star\right) \\
&= \left(\widehat{\boldsymbol{\theta}}_t - \boldsymbol{\theta}_\star\right)^\top \left(\widehat{\boldsymbol{A}}^{(t)}\right)^{\frac{1}{2}} \left(\widehat{\boldsymbol{A}}^{(t)}\right)^{-\frac{1}{2}} \left[\left(\boldsymbol{A}^{(t)} - \widehat{\boldsymbol{A}}^{(t)}\right) \boldsymbol{\theta}_\star - \lambda \boldsymbol{\theta}_\star\right].
\end{aligned}
\tag{9}
$$

In accordance with the decomposability of matrix sketches, as detailed in Lemma 3, we have

$$
\left\|\boldsymbol{X}_t^\top \boldsymbol{X}_t - \sum_{i=1}^{B_t} \left(\boldsymbol{S}_i^{(t)}\right)^\top \boldsymbol{S}_i^{(t)}\right\|_2 \leq \sum_{i=1}^{B_t} \overline{\sigma}_i
\tag{10}
$$

By Cauchy-Schwartz inequality and the triangle inequality, we have

$$
\begin{aligned}
&\left(\widehat{\boldsymbol{\theta}}_t - \boldsymbol{\theta}_\star\right)^\top \left(\widehat{\boldsymbol{A}}^{(t)}\right)^{\frac{1}{2}} \left(\widehat{\boldsymbol{A}}^{(t)}\right)^{-\frac{1}{2}} \left[\left(\boldsymbol{A}^{(t)} - \widehat{\boldsymbol{A}}^{(t)}\right) \boldsymbol{\theta}_\star - \lambda \boldsymbol{\theta}_\star\right] \\
&\leq \left|\lambda + \sum_{i=1}^{B_t} \overline{\sigma}_i\right| \cdot \left\|\widehat{\boldsymbol{\theta}}_t - \boldsymbol{\theta}_\star\right\|_{\widehat{\boldsymbol{A}}^{(t)}} \cdot \left\|\boldsymbol{\theta}_\star\right\|_{\left(\widehat{\boldsymbol{A}}^{(t)}\right)^{-1}} \\
&\leq H \cdot \frac{\lambda + \sum_{i=1}^{B_t} \overline{\sigma}_i}{\sqrt{\lambda}} \cdot \left\|\widehat{\boldsymbol{\theta}}_t - \boldsymbol{\theta}_\star\right\|_{\widehat{\boldsymbol{A}}^{(t)}},
\end{aligned}
\tag{11}
$$

where the last inequality holds beacause $\widehat{\boldsymbol{A}}^{(t)} \succeq \lambda \boldsymbol{I}$ and $\|\boldsymbol{\theta}_\star\|_2 \leq H$.

**Bounding the variance error.** Then, we aim to bound the second term. We use the following self-normalized martingale concentration inequality by (Abbasi-Yadkori et al., 2011).

**Proposition 1** (Lemma 9 of Abbasi-Yadkori et al. (2011)). *Assume that $\boldsymbol{\eta}_1, ..., \boldsymbol{\eta}_t$ is a conditionally $R$-subgaussian real-valued stochastic process and $\boldsymbol{X}_t^\top = [\boldsymbol{x}_1^\top, ..., \boldsymbol{x}_t^\top]$ is any stochastic process such that $\boldsymbol{x}_i$ is measurable with respect to the $\sigma$-algebra generated by $\boldsymbol{\eta}_1, ..., \boldsymbol{\eta}_t$. Then, for any $\delta > 0$, with probability at least $1 - \delta$, for all $t \geq 0$,*

$$
\left\|\boldsymbol{X}_t^\top \boldsymbol{\eta}_t\right\|^2_{\left(\boldsymbol{A}^{(t)}\right)^{-1}} \leq 2R^2 \ln\left(\frac{1}{\delta} \left|\boldsymbol{A}^{(t)}\right|^{\frac{1}{2}} |\lambda \boldsymbol{I}|^{-\frac{1}{2}}\right).
$$

Notice that the variance error can be reformulated as

$$
\begin{aligned}
\left(\widehat{\boldsymbol{\theta}}_t - \boldsymbol{\theta}_\star\right)^\top \boldsymbol{X}_t^\top \boldsymbol{\eta}_t &= \left(\widehat{\boldsymbol{\theta}}_t - \boldsymbol{\theta}_\star\right)^\top \left(\boldsymbol{A}^{(t)}\right)^{-\frac{1}{2}} \left(\boldsymbol{A}^{(t)}\right)^{\frac{1}{2}} \boldsymbol{X}_t^\top \boldsymbol{\eta}_t \\
&\leq \left\|\widehat{\boldsymbol{\theta}}_t - \boldsymbol{\theta}_\star\right\|_{\widehat{\boldsymbol{A}}^{(t)}} \cdot \frac{\left\|\widehat{\boldsymbol{\theta}}_t - \boldsymbol{\theta}_\star\right\|_{\boldsymbol{A}^{(t)}}}{\left\|\widehat{\boldsymbol{\theta}}_t - \boldsymbol{\theta}_\star\right\|_{\widehat{\boldsymbol{A}}^{(t)}}} \cdot \left\|\boldsymbol{X}_t^\top \boldsymbol{\eta}_t\right\|_{\left(\boldsymbol{A}^{(t)}\right)^{-1}},
\end{aligned}
\tag{12}
$$

where the last inequality uses Cauchy-Schwartz inequality.

For any vector $\boldsymbol{a}$, we have

$$
\begin{aligned}
\|\boldsymbol{a}\|_{\boldsymbol{A}^{(t)}}^2 - \|\boldsymbol{a}\|_{\widehat{\boldsymbol{A}}^{(t)}}^2 &= \boldsymbol{a}^\top \left( \boldsymbol{A}^{(t)} - \widehat{\boldsymbol{A}}^{(t)} \right) \boldsymbol{a} \\
&= \boldsymbol{a}^\top \left( \boldsymbol{X}^\top \boldsymbol{X} - \sum_{i=1}^{B_t} \left( \boldsymbol{S}_i^{(t)} \right)^\top \boldsymbol{S}_i^{(t)} \right) \boldsymbol{a} \leq \sum_{i=1}^{B_t} \overline{\sigma}_i \cdot \|\boldsymbol{a}\|_2^2.
\end{aligned} \tag{13}
$$

Therefore, the ratios of norms on the right-hand side of equation 12 can be bounded as

$$
\begin{aligned}
\frac{\left\| \widehat{\boldsymbol{\theta}}_t - \boldsymbol{\theta}_\star \right\|_{\boldsymbol{A}^{(t)}}}{\left\| \widehat{\boldsymbol{\theta}}_t - \boldsymbol{\theta}_\star \right\|_{\widehat{\boldsymbol{A}}^{(t)}}} &= \sqrt{\frac{\left\| \widehat{\boldsymbol{\theta}}_t - \boldsymbol{\theta}_\star \right\|_{\boldsymbol{A}^{(t)}}^2}{\left\| \widehat{\boldsymbol{\theta}}_t - \boldsymbol{\theta}_\star \right\|_{\widehat{\boldsymbol{A}}^{(t)}}^2}} \\
&\leq \sqrt{\frac{\left\| \widehat{\boldsymbol{\theta}}_t - \boldsymbol{\theta}_\star \right\|_{\widehat{\boldsymbol{A}}^{(t)}}^2 + \sum_{i=1}^{B_t} \overline{\sigma}_i \left\| \widehat{\boldsymbol{\theta}}_t - \boldsymbol{\theta}_\star \right\|^2}{\left\| \widehat{\boldsymbol{\theta}}_t - \boldsymbol{\theta}_\star \right\|_{\widehat{\boldsymbol{A}}^{(t)}}^2}} \\
&\leq \sqrt{1 + \frac{\sum_{i=1}^{B_t} \overline{\sigma}_i}{\lambda}}.
\end{aligned} \tag{14}
$$

Substituting equation 14 and Proposition 1 into equation 12 gives

$$
\begin{aligned}
&\left\| \widehat{\boldsymbol{\theta}}_t - \boldsymbol{\theta}_\star \right\|_{\widehat{\boldsymbol{A}}^{(t)}} \cdot \frac{\left\| \widehat{\boldsymbol{\theta}}_t - \boldsymbol{\theta}_\star \right\|_{\boldsymbol{A}^{(t)}}}{\left\| \widehat{\boldsymbol{\theta}}_t - \boldsymbol{\theta}_\star \right\|_{\widehat{\boldsymbol{A}}^{(t)}}} \cdot \left\| \boldsymbol{X}_t^\top \boldsymbol{\eta}_t \right\|_{\left( \boldsymbol{A}^{(t)} \right)^{-1}} \\
&\leq \sqrt{1 + \frac{\sum_{i=1}^{B_t} \overline{\sigma}_i}{\lambda}} \cdot \sqrt{2R^2 \ln \left( \frac{1}{\delta} \left| \boldsymbol{A}^{(t)} \right|^{\frac{1}{2}} |\lambda \boldsymbol{I}|^{-\frac{1}{2}} \right)} \cdot \left\| \widehat{\boldsymbol{\theta}}_t - \boldsymbol{\theta}_\star \right\|_{\widehat{\boldsymbol{A}}^{(t)}}.
\end{aligned} \tag{15}
$$

Motivated by Abbasi-Yadkori et al. (2011); Kuzborskij et al. (2019), we apply the multi-scale sketch-based determinant-trace inequality. Compared to the non-sketched version, this inequality depends on the approximate covariance matrix $\widehat{\boldsymbol{A}}$, reflecting the costs associated with the shrinkage due to multi-scale sketching.

**Lemma 6** (Multi-scale sketch-based determinant-trace inequality). *For any $t \geq 1$, define $\boldsymbol{A}^{(t)} = \lambda \boldsymbol{I} + \boldsymbol{X}_t^\top \boldsymbol{X}_t$, and assume $\|\boldsymbol{x}_t\|_2 \leq L$, we have*

$$
\ln \left( \frac{|\boldsymbol{A}^{(t)}|}{|\lambda \boldsymbol{I}|} \right) \leq d \ln \left( 1 + \frac{\sum_{i=1}^{B_t} \overline{\sigma}_i}{\lambda} \right) + 2l_{B_t} \cdot \ln \left( 1 + \frac{tL^2}{2l_{B_t} \lambda} \right).
$$

*Proof.* $\sum_{i=1}^{B_t} (\boldsymbol{S}_i^{(t)})^\top \boldsymbol{S}_i^{(t)}$ has rank at most $2l_{B_t}$ due to the Dyadic Block Sketching. Since $\widehat{\boldsymbol{A}}^{(t)} = \lambda \boldsymbol{I} + \sum_{i=1}^{B_t} (\boldsymbol{S}_i^{(t)})^\top \boldsymbol{S}_i^{(t)}$ and $\boldsymbol{A}^{(t)} \preceq \widehat{\boldsymbol{A}}^{(t)} + \sum_{i=1}^{B_t} \overline{\sigma}_i \cdot \boldsymbol{I}$, we have

$$
\begin{aligned}
\left| \boldsymbol{A}^{(t)} \right| &\leq \left| \widehat{\boldsymbol{A}}^{(t)} + \sum_{i=1}^{B_t} \overline{\sigma}_i \cdot \boldsymbol{I} \right| \\
&\leq \left( \lambda + \sum_{i=1}^{B_t} \overline{\sigma}_i \right)^{d - 2l_{B_t}} \cdot \left( \frac{\sum_{i=1}^{2l_{B_t}} \left( \lambda_i \left( \widehat{\boldsymbol{A}}^{(t)} \right) + \sum_{i=1}^{B_t} \overline{\sigma}_i \right)}{2l_{B_t}} \right)^{2l_{B_t}} \\
&\leq \left( \lambda + \sum_{i=1}^{B_t} \overline{\sigma}_i \right)^{d - 2l_{B_t}} \cdot \left( \lambda + \sum_{i=1}^{B_t} \overline{\sigma}_i + \frac{\mathrm{Tr} \left( \sum_{i=1}^{B_t} \left( \boldsymbol{S}_i^{(t)} \right)^\top \boldsymbol{S}_i^{(t)} \right)}{2l_{B_t}} \right)^{2l_{B_t}} \\
&\leq \left( \lambda + \sum_{i=1}^{B_t} \overline{\sigma}_i \right)^{d - 2l_{B_t}} \cdot \left( \lambda + \sum_{i=1}^{B_t} \overline{\sigma}_i + \frac{tL^2}{2l_{B_t}} \right)^{2l_{B_t}},
\end{aligned}
$$

where the last inequality holds because

$$\mathrm{Tr}\left(\sum_{i=1}^{B_t}\left(\boldsymbol{S}_i^{(t)}\right)^\top \boldsymbol{S}_i^{(t)}\right) \le \mathrm{Tr}\left(\boldsymbol{X}_t^\top \boldsymbol{X}_t\right) = \sum_{s=1}^{t}\boldsymbol{x}_s^\top \boldsymbol{x}_s \le tL^2$$

Therefore, we have

$$\ln\left(\frac{|\boldsymbol{A}^{(t)}|}{|\lambda\boldsymbol{I}|}\right) \le \ln\left\{\left(\frac{\lambda + \sum_{i=1}^{B_t}\overline{\sigma}_i}{\lambda}\right)^{d-2l_{B_t}}\cdot\left(\frac{\lambda + \sum_{i=1}^{B_t}\overline{\sigma}_i + \frac{tL^2}{2l_{B_t}}}{\lambda}\right)^{2l_{B_t}}\right\}$$

$$= (d - 2l_{B_t})\ln\left(1 + \frac{\sum_{i=1}^{B_t}\overline{\sigma}_i}{\lambda}\right) + 2l_{B_t}\ln\left(1 + \frac{\sum_{i=1}^{B_t}\overline{\sigma}_i}{\lambda} + \frac{tL^2}{2l_{B_t}\lambda}\right)$$

$$\le d\ln\left(1 + \frac{\sum_{i=1}^{B_t}\overline{\sigma}_i}{\lambda}\right) + 2l_{B_t}\cdot\ln\left(1 + \frac{tL^2}{2l_{B_t}\lambda}\right).$$

$\square$

According to Lemma 6, we finally bound the variance error term as follows

$$\left\|\widehat{\boldsymbol{\theta}}_t - \boldsymbol{\theta}_\star\right\|_{\widehat{\boldsymbol{A}}^{(t)}} \cdot \frac{\left\|\widehat{\boldsymbol{\theta}}_t - \boldsymbol{\theta}_\star\right\|_{\boldsymbol{A}^{(t)}}}{\left\|\widehat{\boldsymbol{\theta}}_t - \boldsymbol{\theta}_\star\right\|_{\widehat{\boldsymbol{A}}^{(t)}}} \cdot \left\|\boldsymbol{X}_t^\top \boldsymbol{\eta}_t\right\|_{\left(\boldsymbol{A}^{(t)}\right)^{-1}}$$

$$\le \sqrt{1 + \frac{\sum_{i=1}^{B_t}\overline{\sigma}_i}{\lambda}}\cdot\sqrt{2R^2\ln\left(\frac{1}{\delta}|\boldsymbol{A}^{(t)}|^{\frac{1}{2}}|\lambda\boldsymbol{I}|^{-\frac{1}{2}}\right)}\cdot\left\|\widehat{\boldsymbol{\theta}}_t - \boldsymbol{\theta}_\star\right\|_{\widehat{\boldsymbol{A}}^{(t)}}$$

$$\le R\cdot\sqrt{1 + \frac{\sum_{i=1}^{B_t}\overline{\sigma}_i}{\lambda}}\cdot\sqrt{2\ln\left(\frac{1}{\delta}\right) + d\ln\left(1 + \frac{\sum_{i=1}^{B_t}\overline{\sigma}_i}{\lambda}\right) + 2l_{B_t}\cdot\ln\left(1 + \frac{tL^2}{2l_{B_t}\lambda}\right)}\cdot\left\|\widehat{\boldsymbol{\theta}}_t - \boldsymbol{\theta}_\star\right\|_{\widehat{\boldsymbol{A}}^{(t)}}.$$

Sum up the bias error and the variance error and divide both sides of equation 8 by $\|\widehat{\boldsymbol{\theta}}_t - \boldsymbol{\theta}_\star\|_{\widehat{\boldsymbol{A}}^{(t)}}$ simultaneously, we have

$$\left\|\widehat{\boldsymbol{\theta}}_t - \boldsymbol{\theta}_\star\right\|_{\widehat{\boldsymbol{A}}^{(t)}} \le R\cdot\sqrt{1 + \frac{\sum_{i=1}^{B_t}\overline{\sigma}_i}{\lambda}}\cdot\sqrt{2\ln\left(\frac{1}{\delta}\right) + d\ln\left(1 + \frac{\sum_{i=1}^{B_t}\overline{\sigma}_i}{\lambda}\right) + 2l_{B_t}\cdot\ln\left(1 + \frac{tL^2}{2l_{B_t}\lambda}\right)}$$

$$+ H\cdot\frac{\lambda + \sum_{i=1}^{B_t}\overline{\sigma}_i}{\sqrt{\lambda}}$$

$$\le R\cdot\sqrt{1 + \frac{\epsilon}{\lambda}}\cdot\sqrt{2\ln\left(\frac{1}{\delta}\right) + d\ln\left(1 + \frac{\epsilon}{\lambda}\right) + 2l_{B_t}\cdot\ln\left(1 + \frac{tL^2}{2l_{B_t}\lambda}\right)} + H\cdot\frac{\lambda + \epsilon}{\sqrt{\lambda}}$$

$$\lesssim R\sqrt{d\ln\left(1 + \frac{\epsilon}{\lambda}\right) + 2l_{B_t}}\cdot\sqrt{1 + \frac{\epsilon}{\lambda}} + \frac{H(\lambda + \epsilon)}{\sqrt{\lambda}},$$

where the second inequality follows from the error bound of Dyadic Block Sketching as stated in Theorem 1.

### E.2 Proof of Theorem 3

Having established the confidence ellipsoid, we now focus on analyzing the regret. We begin with an analysis of the instantaneous regret. Recall that the optimal arm at round $t$ is defined as $\boldsymbol{x}_t^\star = \arg\max_{\boldsymbol{x}\in\mathcal{X}_t}(\boldsymbol{x}^\top\boldsymbol{\theta}_\star)$. On the other hand, the principle of optimism in the face of uncertainty ensures that $(\boldsymbol{x}_t, \widehat{\boldsymbol{\theta}}_{t-1}) = \arg\max_{(\boldsymbol{x},\boldsymbol{\theta})\in\mathcal{X}_t\times\Theta_{t-1}}\boldsymbol{x}^\top\boldsymbol{\theta}$. By denoting $\widetilde{\boldsymbol{\theta}}_t$ as the RLS estimator, we utilize these facts

to establish the bound on the instantaneous regret as follows

$$
\begin{aligned}
(\boldsymbol{x}_t^\star - \boldsymbol{x}_t)^\top \boldsymbol{\theta}_\star &\le \boldsymbol{x}_t^\top \widehat{\boldsymbol{\theta}}_{t-1} - \boldsymbol{x}_t^\top \boldsymbol{\theta}_\star \\
&= \boldsymbol{x}_t^\top \left( \widehat{\boldsymbol{\theta}}_{t-1} - \widetilde{\boldsymbol{\theta}}_{t-1} \right) + \boldsymbol{x}_t^\top \left( \widetilde{\boldsymbol{\theta}}_{t-1} - \boldsymbol{\theta}_\star \right) \\
&\le \|\boldsymbol{x}_t\|_{(\widehat{\boldsymbol{A}}^{(t-1)})^{-1}} \cdot \left( \left\| \widehat{\boldsymbol{\theta}}_{t-1} - \widetilde{\boldsymbol{\theta}}_{t-1} \right\|_{\widehat{\boldsymbol{A}}^{(t-1)}} + \left\| \widetilde{\boldsymbol{\theta}}_{t-1} - \boldsymbol{\theta}_\star \right\|_{\widehat{\boldsymbol{A}}^{(t-1)}} \right) \\
&\le 2 \widehat{\beta}_{t-1}(\delta) \cdot \|\boldsymbol{x}_t\|_{(\widehat{\boldsymbol{A}}^{(t-1)})^{-1}} .
\end{aligned}
\tag{16}
$$

Now, we are prepared to establish the upper bound of regret. Utilizing equation 16 and Cauchy-Schwartz inequality, we derive the following bound

$$
\begin{aligned}
\text{Regret}_T &= \sum_{t=1}^{T} \max_{\boldsymbol{x} \in \mathcal{X}} \boldsymbol{x}^\top \boldsymbol{\theta}_\star - \sum_{t=1}^{T} \boldsymbol{x}_t^\top \boldsymbol{\theta}_\star \\
&\le 2 \sum_{t=1}^{T} \min \left\{ HL, \widehat{\beta}_{t-1}(\delta) \cdot \|\boldsymbol{x}_t\|_{(\widehat{\boldsymbol{A}}^{(t-1)})^{-1}} \right\} \\
&\le 2 \sum_{t=1}^{T} \widehat{\beta}_{t-1}(\delta) \min \left\{ \frac{L}{\sqrt{\lambda}}, \|\boldsymbol{x}_t\|_{(\widehat{\boldsymbol{A}}^{(t-1)})^{-1}} \right\} \\
&\le 2 \cdot \max \left\{ 1, \frac{L}{\sqrt{\lambda}} \right\} \cdot \widehat{\beta}_T(\delta) \cdot \sum_{t=1}^{T} \min \left\{ 1, \|\boldsymbol{x}_t\|_{(\widehat{\boldsymbol{A}}^{(t-1)})^{-1}} \right\} \\
&\le 2 \cdot \max \left\{ 1, \frac{L}{\sqrt{\lambda}} \right\} \cdot \widehat{\beta}_T(\delta) \cdot \sqrt{T \sum_{t=1}^{T} \min \left\{ 1, \|\boldsymbol{x}_t\|_{(\widehat{\boldsymbol{A}}^{(t-1)})^{-1}}^2 \right\}} .
\end{aligned}
\tag{17}
$$

We further bound the terms in the above. In particular, we formulate $\widehat{\beta}_T(\delta)$ by Theorem 2 as follows

$$
\begin{aligned}
\widehat{\beta}_T(\delta) &= R \sqrt{1 + \frac{\sum_{i=1}^{B_T} \overline{\sigma}_i}{\lambda}} \cdot \sqrt{2 \ln \frac{1}{\delta} + d \ln \left( 1 + \frac{\sum_{i=1}^{B_T} \overline{\sigma}_i}{\lambda} \right) + 2 l_{B_T} \cdot \ln \left( 1 + \frac{TL^2}{2 l_{B_T} \lambda} \right)} \\
&\quad + H \sqrt{\lambda} \left( 1 + \frac{\sum_{i=1}^{B_T} \overline{\sigma}_i}{\lambda} \right) .
\end{aligned}
\tag{18}
$$

Besides, we adopt the Sketched leverage scores established by Kuzborskij et al. (2019) as follows

**Proposition 2** (Lemma 6 of Kuzborskij et al. (2019)). *The sketched leverage scores through sketching at round $T$ can be upper bounded as*

$$
\begin{aligned}
&\sum_{t=1}^{T} \min \left\{ 1, \|\boldsymbol{x}_t\|_{(\widehat{\boldsymbol{A}}^{(t)})^{-1}}^2 \right\} \\
&\le 2 \left( 1 + \frac{\sum_{i=1}^{B_T} \overline{\sigma}_i}{\lambda} \right) \cdot \ln \left( \frac{|\boldsymbol{A}^{(T)}|}{|\lambda \boldsymbol{I}|} \right) \\
&\le 2 \left( 1 + \frac{\sum_{i=1}^{B_T} \overline{\sigma}_i}{\lambda} \right) \cdot \left( d \ln \left( \frac{1 + \sum_{i=1}^{B_T} \overline{\sigma}_i}{\lambda} \right) + 2 l_{B_t} \cdot \ln \left( 1 + \frac{TL^2}{2 l_{B_T} \lambda} \right) \right) .
\end{aligned}
$$

Combining equation 18, equation 17 and Proposition 2, assuming $L \ge \sqrt{\lambda}$, we have

$$\text{Regret}_T \leq 2 \cdot \max\left\{1, \frac{L}{\sqrt{\lambda}}\right\} \cdot \widehat{\beta}_T(\delta) \cdot \sqrt{T \sum_{t=1}^{T} \min\left\{1, \|\boldsymbol{x}_t\|^2_{(\widehat{\boldsymbol{A}}^{(t-1)})^{-1}}\right\}}$$

$$\leq \frac{L}{\sqrt{\lambda}} \cdot \sqrt{T} \cdot \left(1 + \frac{\sum_{i=1}^{B_T} \overline{\sigma}_i}{\lambda}\right) \cdot \left(d \ln\left(\frac{1 + \sum_{i=1}^{B_T} \overline{\sigma}_i}{\lambda}\right) + 2l_{B_t} \cdot \ln\left(1 + \frac{TL^2}{2l_{B_T}\lambda}\right)\right)$$

$$\cdot \left(R\sqrt{1 + \frac{\sum_{i=1}^{B_T} \overline{\sigma}_i}{\lambda}} \cdot \sqrt{2\ln\frac{1}{\delta} + d\ln\left(1 + \frac{\sum_{i=1}^{B_T} \overline{\sigma}_i}{\lambda}\right) + 2l_{B_T} \cdot \ln\left(1 + \frac{TL^2}{2l_{B_T}\lambda}\right)}\right.$$

$$\left. + H\sqrt{\lambda}\left(1 + \frac{\sum_{i=1}^{B_T} \overline{\sigma}_i}{\lambda}\right)\right).$$

According to Theorem 1, we can bound the spectral error by error $\epsilon$, i.e., $\sum_{i=1}^{B_T} \overline{\sigma}_i \leq \epsilon$. Given that $L \geq \sqrt{\lambda}$, the complete regret bound of DBSLinUCB using FD is as follows

$$\text{Regret}_T \leq \frac{L}{\sqrt{\lambda}} \cdot \sqrt{T} \cdot \left(1 + \frac{\epsilon}{\lambda}\right) \cdot \left(d\ln\left(\frac{1 + \epsilon}{\lambda}\right) + 2l_{B_t} \cdot \ln\left(1 + \frac{TL^2}{2l_{B_T}\lambda}\right)\right)$$

$$\cdot \left(R\sqrt{1 + \frac{\epsilon}{\lambda}} \cdot \sqrt{2\ln\frac{1}{\delta} + d\ln\left(1 + \frac{\epsilon}{\lambda}\right) + 2l_{B_T} \cdot \ln\left(1 + \frac{TL^2}{2l_{B_T}\lambda}\right)} + H\sqrt{\lambda}\left(1 + \frac{\epsilon}{\lambda}\right)\right)$$

$$\lesssim \frac{L(R + H\sqrt{\lambda})}{\sqrt{\lambda}} \cdot \left(d\ln\left(1 + \frac{\epsilon}{\lambda}\right) + 2l_{B_T} \cdot \ln\left(1 + \frac{TL^2}{2l_{B_T}\lambda}\right)\right) \cdot \left(1 + \frac{\epsilon}{\lambda}\right)^{\frac{3}{2}} \sqrt{T}.$$

Ignoring the constants $L$, $R$, and $H$, as well as the logarithmic terms, we simplify the regret bound to

$$\text{Regret}_T \overset{\widetilde{O}}{=} \left(1 + \frac{\epsilon}{\lambda}\right)^{\frac{3}{2}} \cdot (d + l_{B_T}) \cdot \sqrt{T}.$$

### E.3 PROOF OF THEOREM 4

We denote $B_t$ as the number of blocks at round $t$, and $\overline{\sigma}_i$ as the cumulative shrinking singular values in the sketch of block $i$. Let $l_{B_t}$ be the sketch size in the active block at round $t$. Similarly, our analysis establishes an intermediate result regarding the confidence ellipsoid.

**Theorem 6** (Sketched confidence ellipsoid by RFD). *Let $\widehat{\boldsymbol{\theta}}_t$ be the RLS estimate constructed by an arbitrary policy for linear bandits after $t$ rounds of play. For any $\delta \in (0, 1)$, the optimal unknown weight $\boldsymbol{\theta}_\star$ belongs to the set $\Theta_t \equiv \left\{\boldsymbol{\theta} \in \mathbb{R}^d : \left\|\boldsymbol{\theta} - \widehat{\boldsymbol{\theta}}_t\right\|_{\widehat{\boldsymbol{A}}^{(t)}} \leq \widehat{\beta}_t(\delta)\right\}$ with probability at least $1 - \delta$, where*

$$\widehat{\beta}_t(\delta) = R \cdot \sqrt{2\ln\left(\frac{1}{\delta}\right) + d\ln\left(1 + \frac{\sum_{i=1}^{B_t} \overline{\sigma}_i}{\lambda}\right) + 2l_{B_t} \cdot \ln\left(1 + \frac{tL^2}{2l_{B_t}\lambda} + \frac{h_t}{\lambda}\right)}$$

$$+ H \cdot \sqrt{\lambda + \sum_{i=1}^{B_t} \overline{\sigma}_i}$$

*and*

$$h_t = \sum_{i=1}^{B_t} \overline{\sigma}_i - \frac{\sum_{i=1}^{B_t} l_i \cdot \overline{\sigma}_i}{2l_{B_t}}.$$

*Proof.* Notice that RFD uses the adaptive regularization term to approximate the covariance matrix, i.e., $\widehat{\boldsymbol{A}}^{(t)} = \lambda\boldsymbol{I} + \sum_{i=1}^{B_t} \alpha_i^{(t)}\boldsymbol{I} + \sum_{i=1}^{B_t}(\boldsymbol{S}_i^{(t)})^\top \boldsymbol{S}_i^{(t)}$, where $\boldsymbol{S}_i^{(t)}$ is the sketch matrix in block $i$ and $\alpha_i^{(t)}$ is the adaptive regularization term of RFD at round $t$.

Similarily, we decompose $\left\|\widehat{\boldsymbol{\theta}}_t - \boldsymbol{\theta}_\star\right\|_{\widehat{\boldsymbol{A}}^{(t)}}^2$ into two parts as follows

$$
\begin{aligned}
&\left\|\widehat{\boldsymbol{\theta}}_t - \boldsymbol{\theta}_\star\right\|_{\widehat{\boldsymbol{A}}^{(t)}}^2 \\
&= \left(\widehat{\boldsymbol{\theta}}_t - \boldsymbol{\theta}_\star\right)^\top \widehat{\boldsymbol{A}}^{(t)} \left(\widehat{\boldsymbol{\theta}}_t - \boldsymbol{\theta}_\star\right) \\
&= \left(\widehat{\boldsymbol{\theta}}_t - \boldsymbol{\theta}_\star\right)^\top \widehat{\boldsymbol{A}}^{(t)} \left(\left(\widehat{\boldsymbol{A}}^{(t)}\right)^{-1} \boldsymbol{X}_t^\top (\boldsymbol{X}_t \boldsymbol{\theta}_\star + \boldsymbol{\eta}_t) - \boldsymbol{\theta}_\star\right) \\
&= \underbrace{\left(\widehat{\boldsymbol{\theta}}_t - \boldsymbol{\theta}_\star\right)^\top \widehat{\boldsymbol{A}}^{(t)} \left(\left(\widehat{\boldsymbol{A}}^{(t)}\right)^{-1} \boldsymbol{X}_t^\top \boldsymbol{X}_t \boldsymbol{\theta}_\star - \boldsymbol{\theta}_\star\right)}_{\text{Term 1: Bias Error}} + \underbrace{\left(\widehat{\boldsymbol{\theta}}_t - \boldsymbol{\theta}_\star\right)^\top \boldsymbol{X}_t^\top \boldsymbol{\eta}_t}_{\text{Term 2: Variance Error}}.
\end{aligned}
$$

**Bounding the bias error.** For the bias error term, we have

$$
\begin{aligned}
&\left(\widehat{\boldsymbol{\theta}}_t - \boldsymbol{\theta}_\star\right)^\top \widehat{\boldsymbol{A}}^{(t)} \left(\left(\widehat{\boldsymbol{A}}^{(t)}\right)^{-1} \boldsymbol{X}_t^\top \boldsymbol{X}_t \boldsymbol{\theta}_\star - \boldsymbol{\theta}_\star\right) \\
&= \left(\widehat{\boldsymbol{\theta}}_t - \boldsymbol{\theta}_\star\right)^\top \left(\widehat{\boldsymbol{A}}^{(t)}\right)^{\frac{1}{2}} \left(\widehat{\boldsymbol{A}}^{(t)}\right)^{-\frac{1}{2}} \left(\boldsymbol{X}_t^\top \boldsymbol{X}_t \boldsymbol{\theta}_\star - \widehat{\boldsymbol{A}}^{(t)} \boldsymbol{\theta}_\star\right) \\
&= \left(\widehat{\boldsymbol{\theta}}_t - \boldsymbol{\theta}_\star\right)^\top \left(\widehat{\boldsymbol{A}}^{(t)}\right)^{\frac{1}{2}} \left(\widehat{\boldsymbol{A}}^{(t)}\right)^{-\frac{1}{2}} \left(\boldsymbol{X}_t^\top \boldsymbol{X}_t - \lambda \boldsymbol{I} - \sum_{i=1}^{B_t} \alpha_i^{(t)} \boldsymbol{I} - \sum_{i=1}^{B_t} \left(\boldsymbol{S}_i^{(t)}\right)^\top \boldsymbol{S}_i^{(t)}\right) \boldsymbol{\theta}_\star \\
&\triangleq \left(\widehat{\boldsymbol{\theta}}_t - \boldsymbol{\theta}_\star\right)^\top \left(\widehat{\boldsymbol{A}}^{(t)}\right)^{\frac{1}{2}} \left(\widehat{\boldsymbol{A}}^{(t)}\right)^{-\frac{1}{2}} \boldsymbol{D}_t \cdot \boldsymbol{\theta}_\star
\end{aligned}
\tag{19}
$$

Since $\boldsymbol{D}_t = \boldsymbol{X}_t^\top \boldsymbol{X}_t - \lambda \boldsymbol{I} - \sum_{i=1}^{B_t} \alpha_i^{(t)} \boldsymbol{I} - \sum_{i=1}^{B_t} \left(\boldsymbol{S}_i^{(t)}\right)^\top \boldsymbol{S}_i^{(t)}$, for any unit vector $\boldsymbol{a}$, we have

$$
\begin{aligned}
\left|\boldsymbol{a}^\top \boldsymbol{D}_t \boldsymbol{a}\right| &= \left|\boldsymbol{a}^\top \left(\boldsymbol{X}_t^\top \boldsymbol{X}_t - \lambda \boldsymbol{I} - \sum_{i=1}^{B_t} \alpha_i^{(t)} \boldsymbol{I} - \sum_{i=1}^{B_t} \left(\boldsymbol{S}_i^{(t)}\right)^\top \boldsymbol{S}_i^{(t)}\right) \boldsymbol{a}\right| \\
&= \left|\boldsymbol{a}^\top \left(\boldsymbol{X}_t^\top \boldsymbol{X}_t - \sum_{i=1}^{B_t} \left(\boldsymbol{S}_i^{(t)}\right)^\top \boldsymbol{S}_i^{(t)}\right) \boldsymbol{a} - \lambda \boldsymbol{I} - \sum_{i=1}^{B_t} \alpha_i^{(t)} \boldsymbol{I}\right|.
\end{aligned}
\tag{20}
$$

According to Theroem 1, we can get

$$
0 \leq \boldsymbol{a}^\top \left(\boldsymbol{X}_t^\top \boldsymbol{X}_t - \sum_{i=1}^{B_t} \left(\boldsymbol{S}_i^{(t)}\right)^\top \boldsymbol{S}_i^{(t)}\right) \boldsymbol{a} \leq \sum_{i=1}^{B_t} \overline{\sigma}_i.
$$

Bring the above equation into equation 20, since $\sum_{i=1}^{B_t} \alpha_i^{(t)} = \sum_{i=1}^{B_t} \overline{\sigma}_i$, we can bound the spectral norm of $\boldsymbol{D}_t$ as follows

$$
\|\boldsymbol{D}_t\|_2 \leq \lambda + \sum_{i=1}^{B_t} \overline{\sigma}_i.
\tag{21}
$$

By Cauchy-Schwartz inequality and the triangle inequality, we can bound equation 19 by

$$
\begin{aligned}
&\left(\widehat{\boldsymbol{\theta}}_t - \boldsymbol{\theta}_\star\right)^\top \left(\widehat{\boldsymbol{A}}^{(t)}\right)^{\frac{1}{2}} \left(\widehat{\boldsymbol{A}}^{(t)}\right)^{-\frac{1}{2}} \boldsymbol{D}_t \cdot \boldsymbol{\theta}_\star \\
&\leq \left\|\widehat{\boldsymbol{\theta}}_t - \boldsymbol{\theta}_\star\right\|_{\widehat{\boldsymbol{A}}^{(t)}} \cdot \|\boldsymbol{D}_t\|_2 \cdot \|\boldsymbol{\theta}_\star\|_{\left(\widehat{\boldsymbol{A}}^{(t)}\right)^{-1}} \\
&\leq H \cdot \sqrt{\lambda + \sum_{i=1}^{B_t} \overline{\sigma}_i} \cdot \left\|\widehat{\boldsymbol{\theta}}_t - \boldsymbol{\theta}_\star\right\|_{\widehat{\boldsymbol{A}}^{(t)}},
\end{aligned}
\tag{22}
$$

where the last inequality holds because

$$
\|\boldsymbol{\theta}_\star\|_{\left(\widehat{\boldsymbol{A}}^{(t)}\right)^{-1}}^2 \leq \frac{\|\boldsymbol{\theta}_\star\|_2^2}{\lambda_{\min}\left(\widehat{\boldsymbol{A}}^{(t)}\right)} \leq \frac{H^2}{\lambda + \sum_{i=1}^{B_t} \overline{\sigma}_i}.
$$

**Bounding the variance error.** For the variance error, we have

$$
\left(\widehat{\boldsymbol{\theta}}_t - \boldsymbol{\theta}_\star\right)^\top \boldsymbol{X}_t^\top \boldsymbol{\eta}_t = \left(\widehat{\boldsymbol{\theta}}_t - \boldsymbol{\theta}_\star\right)^\top \left(\boldsymbol{A}^{(t)}\right)^{-\frac{1}{2}} \left(\boldsymbol{A}^{(t)}\right)^{\frac{1}{2}} \boldsymbol{X}_t^\top \boldsymbol{\eta}_t
$$

$$
\leq \left\|\widehat{\boldsymbol{\theta}}_t - \boldsymbol{\theta}_\star\right\|_{\widehat{\boldsymbol{A}}^{(t)}} \cdot \frac{\left\|\widehat{\boldsymbol{\theta}}_t - \boldsymbol{\theta}_\star\right\|_{\boldsymbol{A}^{(t)}}}{\left\|\widehat{\boldsymbol{\theta}}_t - \boldsymbol{\theta}_\star\right\|_{\widehat{\boldsymbol{A}}^{(t)}}} \cdot \left\|\boldsymbol{X}_t^\top \boldsymbol{\eta}_t\right\|_{\left(\boldsymbol{A}^{(t)}\right)^{-1}} \tag{23}
$$

$$
\leq \left\|\widehat{\boldsymbol{\theta}}_t - \boldsymbol{\theta}_\star\right\|_{\widehat{\boldsymbol{A}}^{(t)}} \cdot \left\|\boldsymbol{X}_t^\top \boldsymbol{\eta}_t\right\|_{\left(\boldsymbol{A}^{(t)}\right)^{-1}}.
$$

where the last inequality holds because for any vector $\boldsymbol{a}$

$$
\|\boldsymbol{a}\|_{\boldsymbol{A}^{(t)}}^2 - \|\boldsymbol{a}\|_{\widehat{\boldsymbol{A}}^{(t)}}^2 = \boldsymbol{a}^\top \left( \boldsymbol{X}^\top \boldsymbol{X} - \sum_{i=1}^{B_t} \left(\boldsymbol{S}_i^{(t)}\right)^\top \boldsymbol{S}_i^{(t)} - \sum_{i=1}^{B_t} \overline{\sigma}_i \boldsymbol{I} \right) \boldsymbol{a}
$$

$$
= \boldsymbol{a}^\top \left( \boldsymbol{X}^\top \boldsymbol{X} - \sum_{i=1}^{B_t} \left(\boldsymbol{S}_i^{(t)}\right)^\top \boldsymbol{S}_i^{(t)} \right) \boldsymbol{a} - \sum_{i=1}^{B_t} \overline{\sigma}_i \|\boldsymbol{a}\|_2^2 \tag{24}
$$

$$
\leq \sum_{i=1}^{B_t} \overline{\sigma}_i \|\boldsymbol{a}\|_2^2 - \sum_{i=1}^{B_t} \overline{\sigma}_i \|\boldsymbol{a}\|_2^2 = 0
$$

By Proposition 1, we can bound the variance error term as follows

$$
\left(\widehat{\boldsymbol{\theta}}_t - \boldsymbol{\theta}_\star\right)^\top \boldsymbol{X}_t^\top \boldsymbol{\eta}_t
$$

$$
= \left(\widehat{\boldsymbol{\theta}}_t - \boldsymbol{\theta}_\star\right)^\top \left(\boldsymbol{A}^{(t)}\right)^{-\frac{1}{2}} \left(\boldsymbol{A}^{(t)}\right)^{\frac{1}{2}} \boldsymbol{X}_t^\top \boldsymbol{\eta}_t
$$

$$
\leq \left\|\widehat{\boldsymbol{\theta}}_t - \boldsymbol{\theta}_\star\right\|_{\widehat{\boldsymbol{A}}^{(t)}} \cdot \left\|\boldsymbol{X}_t^\top \boldsymbol{\eta}_t\right\|_{\left(\boldsymbol{A}^{(t)}\right)^{-1}} \tag{25}
$$

$$
\leq \sqrt{2R^2 \ln\left( \frac{1}{\delta} \left|\boldsymbol{A}^{(t)}\right|^{\frac{1}{2}} |\lambda \boldsymbol{I}|^{-\frac{1}{2}} \right)} \cdot \left\|\widehat{\boldsymbol{\theta}}_t - \boldsymbol{\theta}_\star\right\|_{\widehat{\boldsymbol{A}}^{(t)}}.
$$

According to equation 24, we have $|\widehat{\boldsymbol{A}}^{(t)}| \geq |\boldsymbol{A}^{(t)}|$. For any $t \in [T]$, since the rank of $\widehat{\boldsymbol{A}}^{(t)}$ is at most $2l_{B_t}$, we can bound the determinant of $\widehat{\boldsymbol{A}}^{(t)}$ as follows

$$
\left|\widehat{\boldsymbol{A}}^{(t)}\right| \leq \left( \sum_{i=1}^{B_t} \alpha_i^{(t)} + \lambda \right)^{d - 2l_{B_t}} \cdot \prod_{i=1}^{2l_{B_t}} \lambda_i\left(\widehat{\boldsymbol{A}}^{(t)}\right)
$$

$$
\leq \left( \sum_{i=1}^{B_t} \alpha_i^{(t)} + \lambda \right)^{d - 2l_{B_t}} \left( \frac{\sum_{i=1}^{2l_{B_t}} \lambda_i\left(\widehat{\boldsymbol{A}}^{(t)}\right)}{2l_{B_t}} \right)^{2l_{B_t}}
$$

$$
= \left( \sum_{i=1}^{B_t} \overline{\sigma}_i + \lambda \right)^{d - 2l_{B_t}} \left[ \sum_{i=1}^{B_t} \overline{\sigma}_i + \lambda + \frac{\text{Tr}\left( \sum_{i=1}^{B_t} \left(\boldsymbol{S}_i^{(t)}\right)^\top \boldsymbol{S}_i^{(t)} \right)}{2l_{B_t}} \right]^{2l_{B_t}} \tag{26}
$$

$$
\leq \left( \sum_{i=1}^{B_t} \overline{\sigma}_i + \lambda \right)^{d - 2l_{B_t}} \left( \left( \sum_{i=1}^{B_t} \overline{\sigma}_i - \frac{\sum_{i=1}^{B_t} l_i \cdot \overline{\sigma}_i}{2l_{B_t}} \right) + \lambda + \frac{TL^2}{2l_{B_t}} \right)^{2l_{B_t}},
$$

where the last inequality satisfies due to

$$
\text{Tr}\left( \sum_{i=1}^{B_t} \left(\boldsymbol{S}_i^{(t)}\right)^\top \boldsymbol{S}_i^{(t)} \right) = \sum_{i=1}^{B_t} \text{Tr}\left( \left(\boldsymbol{S}_i^{(t)}\right)^\top \boldsymbol{S}_i^{(t)} \right) = \sum_{s=1}^{t} \text{Tr}(\boldsymbol{x}_s^\top \boldsymbol{x}_s) - \sum_{i=1}^{B_t} l_i \cdot \overline{\sigma}_i
$$

$$
\leq TL^2 - \sum_{i=1}^{B_t} l_i \cdot \overline{\sigma}_i.
$$

Therefore, the variance error term can be bounded as

$$\left(\widehat{\boldsymbol{\theta}}_t - \boldsymbol{\theta}_\star\right)^\top \boldsymbol{X}_t^\top \boldsymbol{\eta}_t$$

$$\leq \sqrt{2R^2 \ln\left(\frac{1}{\delta} \left|\boldsymbol{A}^{(t)}\right|^{\frac{1}{2}} |\lambda \boldsymbol{I}|^{-\frac{1}{2}}\right)} \cdot \left\|\widehat{\boldsymbol{\theta}}_t - \boldsymbol{\theta}_\star\right\|_{\widehat{\boldsymbol{A}}^{(t)}}$$

$$\leq \sqrt{2R^2 \ln\left(\frac{1}{\delta} \left|\widehat{\boldsymbol{A}}^{(t)}\right|^{\frac{1}{2}} |\lambda \boldsymbol{I}|^{-\frac{1}{2}}\right)} \cdot \left\|\widehat{\boldsymbol{\theta}}_t - \boldsymbol{\theta}_\star\right\|_{\widehat{\boldsymbol{A}}^{(t)}}$$

$$\leq R \cdot \sqrt{2\ln\left(\frac{1}{\delta}\right) + (d - 2l_{B_t})\ln\left(1 + \frac{\sum_{i=1}^{B_t}\overline{\sigma}_i}{\lambda}\right) + 2l_{B_t} \cdot \ln\left(1 + \frac{tL^2}{2l_{B_t}\lambda} + \frac{h_t}{\lambda}\right)} \cdot \left\|\widehat{\boldsymbol{\theta}}_t - \boldsymbol{\theta}_\star\right\|_{\widehat{\boldsymbol{A}}^{(t)}}$$

$$\leq R \cdot \sqrt{2\ln\left(\frac{1}{\delta}\right) + d\ln\left(1 + \frac{\sum_{i=1}^{B_t}\overline{\sigma}_i}{\lambda}\right) + 2l_{B_t} \cdot \ln\left(1 + \frac{tL^2}{2l_{B_t}\lambda} + \frac{h_t}{\lambda}\right)} \cdot \left\|\widehat{\boldsymbol{\theta}}_t - \boldsymbol{\theta}_\star\right\|_{\widehat{\boldsymbol{A}}^{(t)}},$$

where $h_t = \sum_{i=1}^{B_t}\overline{\sigma}_i - \frac{\sum_{i=1}^{B_t} l_i \cdot \overline{\sigma}_i}{2l_{B_t}}$.

Sum up the bias error term and the variance error term and divide both sides by $\|\widehat{\boldsymbol{\theta}}_t - \boldsymbol{\theta}_\star\|_{\widehat{\boldsymbol{A}}^{(t)}}$ simultaneously, we have

$$\left\|\widehat{\boldsymbol{\theta}}_t - \boldsymbol{\theta}_\star\right\|_{\widehat{\boldsymbol{A}}^{(t)}} \leq R \cdot \sqrt{2\ln\left(\frac{1}{\delta}\right) + d\ln\left(1 + \frac{\sum_{i=1}^{B_t}\overline{\sigma}_i}{\lambda}\right) + 2l_{B_t} \cdot \ln\left(1 + \frac{tL^2}{2l_{B_t}\lambda} + \frac{h_t}{\lambda}\right)} + H \cdot \sqrt{\lambda + \sum_{i=1}^{B_t}\overline{\sigma}_i},$$

which concludes the proof. $\qquad\square$

Next, we start to prove the regret. Similar to the case using FD, since the algorithm uses the principle of optimism in the face of uncertainty to select the arm, we can bound instantaneous regret by equation 16. Utilizing equation 16 and Cauchy-Schwartz inequality, we derive the following bound

$$\begin{aligned}
\text{Regret}_T &= \sum_{t=1}^{T} \max_{\boldsymbol{x}\in\mathcal{X}} \boldsymbol{x}^\top \boldsymbol{\theta}_\star - \sum_{t=1}^{T} \boldsymbol{x}_t^\top \boldsymbol{\theta}_\star \\
&\leq 2\sum_{t=1}^{T} \min\left\{HL, \widehat{\beta}_{t-1}(\delta) \cdot \|\boldsymbol{x}_t\|_{(\widehat{\boldsymbol{A}}^{(t-1)})^{-1}}\right\} \\
&\leq 2\sum_{t=1}^{T} \widehat{\beta}_{t-1}(\delta) \min\left\{\frac{L}{\sqrt{\lambda}}, \|\boldsymbol{x}_t\|_{(\widehat{\boldsymbol{A}}^{(t-1)})^{-1}}\right\} \\
&\leq 2 \cdot \max\left\{1, \frac{L}{\sqrt{\lambda}}\right\} \cdot \widehat{\beta}_T(\delta) \cdot \sum_{t=1}^{T} \min\left\{1, \|\boldsymbol{x}_t\|_{(\widehat{\boldsymbol{A}}^{(t-1)})^{-1}}\right\} \\
&\leq 2 \cdot \max\left\{1, \frac{L}{\sqrt{\lambda}}\right\} \cdot \widehat{\beta}_T(\delta) \cdot \sqrt{T\sum_{t=1}^{T} \min\left\{1, \|\boldsymbol{x}_t\|^2_{(\widehat{\boldsymbol{A}}^{(t-1)})^{-1}}\right\}}.
\end{aligned} \tag{27}$$

We present a lemma of RFD-sketched leverage scores to conclude the proof.

**Lemma 7** (Sketch-based leverage scores by RFD).

$$\sum_{t=1}^{T} \min\left\{1, \|\boldsymbol{x}_t\|^2_{(\widehat{\boldsymbol{A}}^{(t-1)})^{-1}}\right\} \leq 2l_{B_T} \cdot \ln\left(1 + \frac{TL^2}{2l_{B_T}\lambda} + \frac{h_T}{\lambda}\right).$$

*Proof.* Denote $\boldsymbol{C}_t = \widehat{\boldsymbol{A}}^{(t-1)} + \boldsymbol{x}_t^\top \boldsymbol{x}_t$. Notice that the first $2l_{B_t}$ eigenvalues of $\boldsymbol{C}_t$ are the same as $\widehat{\boldsymbol{A}}^{(t)}$ while the other eigenvalues of $\boldsymbol{C}_t$ are $\sum_{i=1}^{B_t}\alpha_i^{(t-1)} + \lambda$. Thus we can obtain

$$\frac{\left|\widehat{\boldsymbol{A}}^{(t)}\right|}{|\boldsymbol{C}_t|} = \left(\frac{\sum_{i=1}^{B_t}\alpha_i^{(t)} + \lambda}{\sum_{i=1}^{B_{t-1}}\alpha_i^{(t-1)} + \lambda}\right)^{d - 2l_{B_t}}.$$

For the determinant of $\widehat{\boldsymbol{A}}^{(t)}$, we have

$$
\begin{aligned}
\left|\widehat{\boldsymbol{A}}^{(t)}\right| &= \left(\frac{\sum_{i=1}^{B_t} \alpha_i^{(t)} + \lambda}{\sum_{i=1}^{B_{t-1}} \alpha_i^{(t-1)} + \lambda}\right)^{d - 2l_{B_t}} \cdot |\boldsymbol{C}_t| \\
&= \left(\frac{\sum_{i=1}^{B_t} \alpha_i^{(t)} + \lambda}{\sum_{i=1}^{B_{t-1}} \alpha_i^{(t-1)} + \lambda}\right)^{d - 2l_{B_t}} \cdot \left|\widehat{\boldsymbol{A}}^{(t-1)}\right| \cdot \left|\boldsymbol{I} + \left(\widehat{\boldsymbol{A}}^{(t-1)}\right)^{-1} \boldsymbol{x}_t^\top \boldsymbol{x}_t\right| \\
&= \left(\frac{\sum_{i=1}^{B_t} \alpha_i^{(t)} + \lambda}{\sum_{i=1}^{B_{t-1}} \alpha_i^{(t-1)} + \lambda}\right)^{d - 2l_{B_t}} \cdot \left|\widehat{\boldsymbol{A}}^{(t-1)}\right| \cdot \left(1 + \|\boldsymbol{x}_t\|_{(\widehat{\boldsymbol{A}}^{(t-1)})^{-1}}^2\right) \\
&= \left(\frac{\sum_{i=1}^{B_t} \overline{\sigma}_i + \lambda}{\lambda}\right)^{d - 2l_{B_t}} \cdot |\lambda \boldsymbol{I}| \cdot \prod_{s=1}^{t} \left(1 + \|\boldsymbol{x}_s\|_{(\widehat{\boldsymbol{A}}^{(s-1)})^{-1}}^2\right).
\end{aligned} \tag{28}
$$

Since $\min(1, x) \le 2\ln(1 + x)$ for all $x \ge 0$, using equation 28, we can derive the following bound

$$
\begin{aligned}
\sum_{t=1}^{T} \min\left\{1, \|\boldsymbol{x}_t\|_{(\widehat{\boldsymbol{A}}^{(t-1)})^{-1}}^2\right\} &\le 2\sum_{t=1}^{T} \ln\left(1 + \|\boldsymbol{x}_t\|_{(\widehat{\boldsymbol{A}}^{(t-1)})^{-1}}^2\right) \\
&= 2 \cdot \ln\left(\left(\frac{\lambda}{\sum_{i=1}^{B_T} \overline{\sigma}_i + \lambda}\right)^{d - 2l_{B_T}} \cdot \frac{\left|\widehat{\boldsymbol{A}}^{(T)}\right|}{|\lambda \boldsymbol{I}|}\right) \\
&\le 2l_{B_T} \cdot \ln\left(1 + \frac{TL^2}{2l_{B_T}\lambda} + \frac{h_T}{\lambda}\right),
\end{aligned}
$$

where the last step holds by equation 26 and $h_T = \sum_{i=1}^{B_T} \overline{\sigma}_i - \frac{\sum_{i=1}^{B_T} l_i \cdot \overline{\sigma}_i}{2l_{B_T}}$.  $\qquad\square$

We combine equation 27, Theorem 6 and Lemma 7. Assume $L \ge \sqrt{\lambda}$, we have

$$
\begin{aligned}
\text{Regret}_T &= \sum_{t=1}^{T} \max_{\boldsymbol{x} \in \mathcal{X}} \boldsymbol{x}^\top \boldsymbol{\theta}_\star - \sum_{t=1}^{T} \boldsymbol{x}_t^\top \boldsymbol{\theta}_\star \\
&\le 2 \cdot \max\left\{1, \frac{L}{\sqrt{\lambda}}\right\} \cdot \widehat{\beta}_T(\delta) \cdot \sqrt{T \sum_{t=1}^{T} \min\left\{1, \|\boldsymbol{x}_t\|_{(\widehat{\boldsymbol{A}}^{(t-1)})^{-1}}^2\right\}} \\
&\le \frac{L}{\sqrt{\lambda}} \cdot \sqrt{T} \cdot \sqrt{2l_{B_T} \cdot \ln\left(1 + \frac{TL^2}{2l_{B_T}\lambda} + \frac{h_T}{\lambda}\right)} \cdot \left(H \cdot \sqrt{\lambda + \sum_{i=1}^{B_T} \overline{\sigma}_i +} \right. \\
&\qquad \left. R \cdot \sqrt{2\ln\left(\frac{1}{\delta}\right) + d\ln\left(1 + \frac{\sum_{i=1}^{B_T} \overline{\sigma}_i}{\lambda}\right) + 2l_{B_T} \cdot \ln\left(1 + \frac{TL^2}{2l_{B_T}\lambda} + \frac{h_T}{\lambda}\right)}\right).
\end{aligned}
$$

According to Theorem 1, the accumulated spectral error $\sum_{i=1}^{B_T} \overline{\sigma}_i$ is bounded by $\epsilon$, and we have

$$
\begin{aligned}
h_T = \sum_{i=1}^{B_T} \overline{\sigma}_i - \frac{\sum_{i=1}^{B_T} l_i \cdot \overline{\sigma}_i}{2l_{B_T}} &= \sum_{i=1}^{B_T} \left(1 - \frac{2^{i-1}}{2^{B_T}}\right) \cdot \overline{\sigma}_i \\
&\le \epsilon \cdot \sum_{i=1}^{B_T} \left(1 - \frac{2^{i-1}}{2^{B_T}}\right) \cdot \frac{1}{2^i} \\
&\le \epsilon.
\end{aligned}
$$

Therefore, we derive the complete regret bound as follows:

$$\text{Regret}_T \le \frac{L}{\sqrt{\lambda}} \cdot \sqrt{T} \cdot \sqrt{2l_{B_T} \cdot \ln\left(1 + \frac{TL^2}{2l_{B_T}\lambda} + \frac{\epsilon}{\lambda}\right)} \cdot \left(H \cdot \sqrt{\lambda + \epsilon} + \right.$$

$$\left. R \cdot \sqrt{2\ln\left(\frac{1}{\delta}\right) + d\ln\left(1 + \frac{\epsilon}{\lambda}\right) + 2l_{B_T} \cdot \ln\left(1 + \frac{TL^2}{2l_{B_T}\lambda} + \frac{\epsilon}{\lambda}\right)}\right)$$

$$\lesssim \frac{L}{\sqrt{\lambda}} \cdot \sqrt{l_{B_T}T} \cdot \sqrt{2\ln\left(1 + \frac{TL^2}{2l_{B_T}\lambda} + \frac{\epsilon}{\lambda}\right)} \cdot \left(H \cdot \sqrt{\lambda + \epsilon} + \right.$$

$$\left. R \cdot \sqrt{d\ln\left(1 + \frac{\epsilon}{\lambda}\right) + 2l_{B_T} \cdot \ln\left(1 + \frac{TL^2}{2l_{B_T}\lambda} + \frac{\epsilon}{\lambda}\right)}\right).$$

Ignoring the constants $L$, $R$, and $H$, as well as the logarithmic terms, we simplify the regret bound to

$$\text{Regret}_T \overset{\widetilde{O}}{=} \left(1 + \frac{\epsilon}{\lambda}\right)^{\frac{1}{2}} \cdot \sqrt{l_{B_T}T} + \sqrt{dl_{B_T}T}.$$

### E.4 PROPERTIES OF DYADIC BLOCK SKETCHING FOR RFD

In this section, we highlight two significant properties of Dyadic Block Sketching for RFD that elucidate why the regret bound of DBSLinUCB using RFD is improved. Although Robust Frequent Directions for ridge regression have been studied by Luo et al. (2019), their theory is limited to single-scale deterministic streaming sketches. We demonstrate that the decomposability of multi-scale sketching does not alter the properties of RFD.

We begin with the positive definite monotonicity of Dyadic Block Sketching for RFD, which ensures that the sequence of approximation matrices is per-step optimal.

**Theorem 7** (Positive Definite Monotonicity). *At round $t$, denote that the Dyadic Block Sketching for RFD provides a sketch $\boldsymbol{S}^{(t)}$, we have the following equation*

$$\left(\boldsymbol{S}^{(t)}\right)^\top \boldsymbol{S}^{(t)} + \alpha^{(t)}\boldsymbol{I} \succeq \left(\boldsymbol{S}^{(t-1)}\right)^\top \boldsymbol{S}^{(t-1)} + \alpha^{(t-1)}\boldsymbol{I}.$$

*Proof.* Notice that $\alpha^{(t)}\boldsymbol{I} + (\boldsymbol{S}^{(t)})^\top \boldsymbol{S}^{(t)} = \sum_{i=1}^{B_t} \alpha_i^{(t)}\boldsymbol{I} + \sum_{i=1}^{B_t}(\boldsymbol{S}_i^{(t)})^\top \boldsymbol{S}_i^{(t)}$, where $\boldsymbol{S}_i^{(t)}$ is the sketch matrix in block $i$ and $\alpha_i^{(t)}$ is the adaptive regularization term of RFD at round $t$.

Let $\boldsymbol{Q} = \left[(\boldsymbol{S}_{B_t}^{(t-1)})^\top, \boldsymbol{x}_t^\top\right]^\top$, $\sigma_t$ is the shrinking singular values of active block at round $t$, the shrinking step of RFD provides

$$\sum_{i=1}^{B_t}\left(\boldsymbol{S}_i^{(t)}\right)^\top \boldsymbol{S}_i^{(t)} + \sigma_t\boldsymbol{I} \succeq \sum_{i=1}^{B_t-1}\left(\boldsymbol{S}_i^{(t)}\right)^\top \boldsymbol{S}_i^{(t)} + \boldsymbol{Q}^\top\boldsymbol{Q} \succeq \sum_{i=1}^{B_t-1}\left(\boldsymbol{S}_i^{(t-1)}\right)^\top \boldsymbol{S}_i^{(t-1)}. \quad (29)$$

Therefore, for any unit vector $\boldsymbol{a}$, we have

$$\boldsymbol{a}^\top \left(\left(\boldsymbol{S}^{(t)}\right)^\top \boldsymbol{S}^{(t)} + \alpha^{(t)}\boldsymbol{I} - \left(\boldsymbol{S}^{(t-1)}\right)^\top \boldsymbol{S}^{(t-1)} + \alpha^{(t-1)}\boldsymbol{I}\right) \boldsymbol{a}$$

$$= \boldsymbol{a}^\top \left(\sum_{i=1}^{B_t} \alpha_i^{(t)}\boldsymbol{I} + \sum_{i=1}^{B_t}\left(\boldsymbol{S}_i^{(t)}\right)^\top \boldsymbol{S}_i^{(t)} - \sum_{i=1}^{B_t-1} \alpha_i^{(t-1)}\boldsymbol{I} - \sum_{i=1}^{B_t-1}\left(\boldsymbol{S}_i^{(t-1)}\right)^\top \boldsymbol{S}_i^{(t-1)}\right) \boldsymbol{a}$$

$$= \boldsymbol{a}^\top \left(\sum_{i=1}^{B_t}\left(\boldsymbol{S}_i^{(t)}\right)^\top \boldsymbol{S}_i^{(t)} + \sigma_t\boldsymbol{I} - \sum_{i=1}^{B_t-1}\left(\boldsymbol{S}_i^{(t-1)}\right)^\top \boldsymbol{S}_i^{(t-1)}\right) \boldsymbol{a}$$

$$\ge 0,$$

which concludes the proof. $\qquad \square$

Next, we prove that the sketch matrix produced by Dyadic Block Sketching for RFD is better conditioned than those produced by Dyadic Block Sketching for FD and the covariance matrix. In this context, the $\alpha$ selected by RFD is optimal, as choosing a smaller $\alpha$ would result in a worse condition number for the approximation matrices.

**Theorem 8** (Well-Conditioned Property). *Let* $\text{cond}(\boldsymbol{X}) = \frac{\sigma_{\max}(\boldsymbol{X})}{\sigma_{\min}(\boldsymbol{X})}$ *be the condition number of matrix* $\boldsymbol{X}$*. At round t, denote that the Dyadic Block Sketching for RFD provides a sketch* $\boldsymbol{S}^{(t)}$*, we have*

$$\text{cond}\left(\left(\boldsymbol{S}^{(t)}\right)^{\top}\boldsymbol{S}^{(t)} + \alpha^{(t)}\boldsymbol{I} + \lambda\boldsymbol{I}\right) \leq \text{cond}\left(\left(\boldsymbol{S}^{(t)}\right)^{\top}\boldsymbol{S}^{(t)} + \lambda\boldsymbol{I}\right),$$

$$\text{cond}\left(\left(\boldsymbol{S}^{(t)}\right)^{\top}\boldsymbol{S}^{(t)} + \alpha^{(t)}\boldsymbol{I} + \lambda\boldsymbol{I}\right) \leq \text{cond}\left(\boldsymbol{X}_t^{\top}\boldsymbol{X}_t + \lambda\boldsymbol{I}\right).$$

*Proof.* Notice that $\alpha^{(t)}\boldsymbol{I} + (\boldsymbol{S}^{(t)})^{\top}\boldsymbol{S}^{(t)} = \sum_{i=1}^{B_t}\alpha_i^{(t)}\boldsymbol{I} + \sum_{i=1}^{B_t}(\boldsymbol{S}_i^{(t)})^{\top}\boldsymbol{S}_i^{(t)}$, where $\boldsymbol{S}_i^{(t)}$ is the sketch matrix in block $i$ and $\alpha_i^{(t)}$ is the adaptive regularization term of RFD at round $t$. We have

$$\text{cond}\left(\left(\boldsymbol{S}^{(t)}\right)^{\top}\boldsymbol{S}^{(t)} + \alpha^{(t)}\boldsymbol{I} + \lambda\boldsymbol{I}\right) = \frac{\sigma_{\max}\left(\sum_{i=1}^{B_t}\left(\boldsymbol{S}_i^{(t)}\right)^{\top}\boldsymbol{S}_i^{(t)}\right) + \lambda + \sum_{i=1}^{B_t}\alpha_i^{(t)}}{\lambda + \sum_{i=1}^{B_t}\alpha_i^{(t)}}$$

$$\leq \frac{\sigma_{\max}\left(\sum_{i=1}^{B_t}\left(\boldsymbol{S}_i^{(t)}\right)^{\top}\boldsymbol{S}_i^{(t)}\right) + \lambda}{\lambda}$$

$$= \text{cond}\left(\left(\boldsymbol{S}^{(t)}\right)^{\top}\boldsymbol{S}^{(t)} + \lambda\boldsymbol{I}\right).$$

Similarly, we have

$$\text{cond}\left(\left(\boldsymbol{S}^{(t)}\right)^{\top}\boldsymbol{S}^{(t)} + \alpha^{(t)}\boldsymbol{I} + \lambda\boldsymbol{I}\right) = \frac{\sigma_{\max}\left(\sum_{i=1}^{B_t}\left(\boldsymbol{S}_i^{(t)}\right)^{\top}\boldsymbol{S}_i^{(t)}\right) + \lambda + \sum_{i=1}^{B_t}\alpha_i^{(t)}}{\lambda + \sum_{i=1}^{B_t}\alpha_i^{(t)}}$$

$$\leq \frac{\sigma_{\max}\left(\boldsymbol{X}_t^{\top}\boldsymbol{X}_t\right) + \lambda + \sum_{i=1}^{B_t}\alpha_i^{(t)}}{\lambda + \sum_{i=1}^{B_t}\alpha_i^{(t)}}$$

$$\leq \frac{\sigma_{\max}\left(\boldsymbol{X}_t^{\top}\boldsymbol{X}_t\right) + \lambda}{\lambda}$$

$$\leq \text{cond}\left(\boldsymbol{X}_t^{\top}\boldsymbol{X}_t + \lambda\boldsymbol{I}\right),$$

which concludes the proof. □

## F OMITTED DETAILS FOR SECTION 5

In this section, we provide the omitted details of experiments in section 5. We provide the experimental setups and configurations in Appendix F.1 and additional experiments in F.2, F.3, and F.4.

### F.1 EXPERIMENTAL SETUPS

All experiments are performed on a machine with 24-core Intel(R) Xeon(R) Gold 6240R 2.40GHz CPU and 256 GB memory. We compare our DBSLinUCB with the state-of-the-art linear bandit algorithms on the synthetic dataset and several well-known classification benchmarks. The training and testing data are merged into a single dataset, followed by vector normalization based on the $l_2$ norm. Each experiment is performed over 20 different random permutations of the datasets. The confidence ellipsoid $\beta$ of all algorithms is searched in $\{10^{-4}, 10^{-3}, \ldots, 1\}$ and $\lambda$ is searched in $\{2 \times 10^{-4}, 2 \times 10^{-3}, \ldots, 2 \times 10^4\}$.

In the experiments of online classification in real-world data, we follow the experimental setup in Kuzborskij et al. (2019). Specifically, we construct the online classification problem within the

contextual bandit setting as follows: given a dataset with data in $M$ labels, we first choose one cluster as the target label. In each round, we randomly draw one sample from each label and compose an arm set of $M$ samples in $M$ contexts. The algorithms choose one sample from the arm set and observe the reward based on whether the selected sample belongs to the target label. The reward is 1 if the selected sample comes from the target label and 0 otherwise.

## F.2 EXPERIMENTS OF MATRIX APPROXIMATION

We evaluate the performance of the proposed Dyadic Block Sketching in terms of matrix approximation. We compare it with FD (Liberty, 2013). We generated a synthetic dataset with $n = 1250$ rows and $d = 500$ columns. Specifically, each row $\boldsymbol{a}_t \in \mathbb{R}^{500}$ is independently drawn from a multivariate Gaussian distribution $\boldsymbol{a}_t \sim \mathcal{N}(\boldsymbol{0}, \boldsymbol{I}_d)$, followed by vector normalization based on the $l_2$ norm. We set the sketch size $l_0 = 50$ for FD and the initial sketch size $l_0 = 16$ for Dyadic Block Sketching. The spectral norm error is defined as $\|\boldsymbol{A}_t^\top \boldsymbol{A}_t - \boldsymbol{S}_t^\top \boldsymbol{S}_t\|_2$, where $\boldsymbol{A}_t$ is the streaming matrix at round $t$ and $\boldsymbol{S}_t$ is the sketch matrix at round $t$.

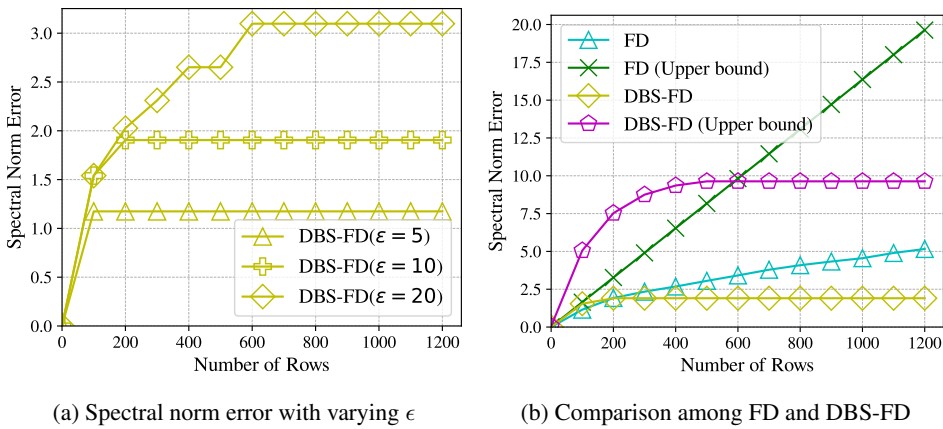

(a) Spectral norm error with varying $\epsilon$   (b) Comparison among FD and DBS-FD

Figure 4: (a): The spectral norm error w.r.t the error parameter $\epsilon$ on synthetic dataset; (b): Comparison among FD and our DBS-FD w.r.t. the error and its upper bound

We first vary the error parameter $\epsilon \in \{5, 10, 20\}$. As illustrated in Figures 4a, we observe that increasing the error parameter $\epsilon$ leads to a larger spectral norm error. Then we compare our method with FD. We set the error parameter $\epsilon = 10$ for Dyadic Block Sketching. Figure 4b presents the spectral norm error $\|\boldsymbol{A}_t^\top \boldsymbol{A}_t - \boldsymbol{S}_t^\top \boldsymbol{S}_t\|_2$ along with its upper bound for matrix sketching. We observe that Dyadic Block Sketching provides a constrained global error bound for matrix sketching. In comparison to FD, the rate of error growth in Dyadic Block Sketching decreases over time, effectively mitigating the linear growth of the spectral tail.

## F.3 MORE EXPERIMENTS ON MNIST

In this section, we present additional experimental results on the MNIST dataset, demonstrating our method's capability to adaptively adjust to optimal sketch sizes. The evaluation focuses primarily on FD-based methods for comparison. For the SOFUL algorithm, we evaluate performance across multiple sketch sizes with $l \in \{20, 100, 150\}$. For our proposed DBSLinUCB algorithm, we initialize the sketch size at $l_0 = 50$ and configure the error parameter to $\epsilon = 8$. We also report that the streaming matrix used by all methods is full rank.

**Regret Performance.** The cumulative regret results demonstrate that DBSLinUCB achieves competitive performance compared to the baseline algorithms. As shown in Figure 5a, DBSLinUCB maintains regret levels comparable to OFUL, the gold standard algorithm, throughout the 2000 rounds of evaluation on the MNIST dataset. In contrast, SOFUL exhibits significant performance degradation when configured with insufficient sketch sizes, particularly evident with $l = 20$ and $l = 100$, where the cumulative regret substantially exceeds that of both OFUL and DBSLinUCB. While SOFUL with $l = 150$ achieves regret performance similar to OFUL and DBSLinUCB, this

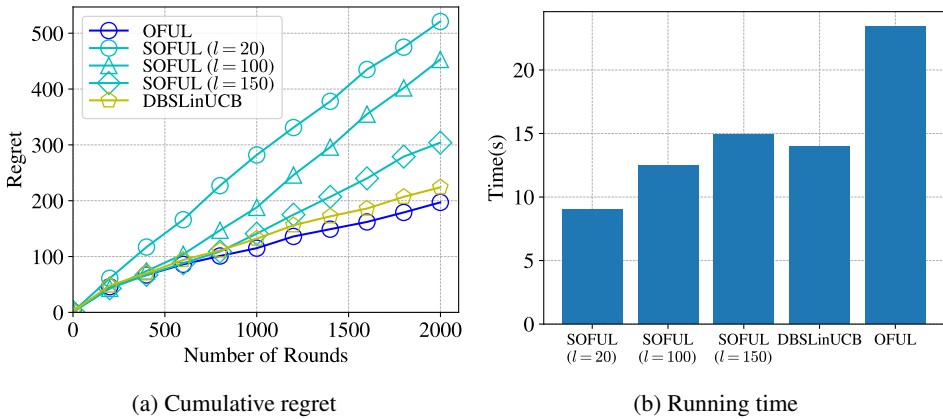

(a) Cumulative regret            (b) Running time

Figure 5: Cumulative regret and total running time of the compared algorithms, the proposed DB-SLinUCB on `MNIST`

configuration requires careful tuning of the sketch size parameter, which presents practical challenges in real-world applications where the optimal sketch size cannot be determined a priori.

**Running Time Efficiency.** The computational efficiency analysis presented in Figure 5b reveals the significant advantage of DBSLinUCB in terms of runtime performance. DBSLinUCB achieves approximately 14 seconds total running time, representing a substantial improvement over OFUL's 23 seconds execution time. Even when compared to SOFUL variants, DBSLinUCB maintains competitive efficiency while avoiding the performance trade-offs associated with fixed sketch sizes. SO-FUL with smaller sketch sizes ($l = 20$) achieves faster runtime at approximately 9 seconds, but this comes at the cost of severely degraded regret performance. The results indicate that DBSLinUCB successfully addresses the fundamental challenge of balancing computational efficiency with learning performance, eliminating the need for manual sketch size tuning while maintaining both competitive regret bounds and superior runtime characteristics.

**Space Complexity.** Table 2 reports the maximum memory footprint of each algorithm over 2000 rounds on `MNIST`. DBSLinUCB achieves a $40\%$ reduction in space usage compared to OFUL while maintaining competitive regret performance. Unlike SOFUL, which requires pre-specifying a fixed sketch size $l$, DBSLinUCB adaptively adjusts its sketch dimensions based on the observed data stream. This adaptive mechanism allows DBSLinUCB to match the space efficiency of SOFUL with $l = 150$ while providing stronger robustness guarantees—notably, SOFUL with smaller sketch sizes ($l \in \{20, 100\}$) achieves lower memory usage but suffers from degraded regret performance as shown in previous experiments. The results demonstrate that DBSLinUCB effectively navigates the space-regret trade-off without requiring prior knowledge of optimal hyperparameters.

Table 2: Comparison of Space Usage on `MNIST`

| Algorithm | Sketch Size ($l$) | Max Space (KB) |
|---|---|---|
| OFUL | N/A | 4802.0 |
| SOFUL | 20 | 682.9 |
| SOFUL | 100 | 1528.3 |
| SOFUL | 150 | 2529.3 |
| DBSLinUCB | Adaptive | 2842.0 |

### F.4 Experiments on Additional Real-World Data

In this section, we evaluate DBSLinUCB on online classification tasks across multiple real-world benchmarks beyond `MNIST`, validating its generalizability and practical effectiveness. Similarly, the baselines include the non-sketched method OFUL (Abbasi-Yadkori et al., 2011) and the sketch-based methods SOFUL (Kuzborskij et al., 2019), CBSCFD (Chen et al., 2020).

Table 3: Dataset Information

| Dataset | OpenML ID | Instances | Features | Classes |
|---------|-----------|-----------|----------|---------|
| cnae-9 | 1468 | 1080 | 856 | 9 |
| MFeat | 22 | 2000 | 48 | 10 |
| Spam | 44 | 4601 | 57 | 2 |

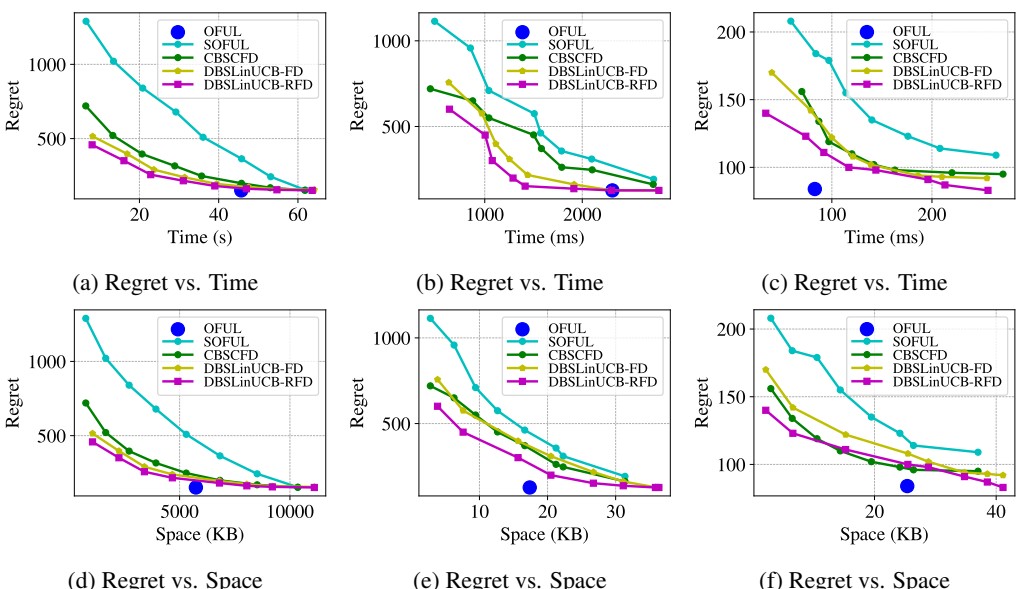

(a) Regret vs. Time  (b) Regret vs. Time  (c) Regret vs. Time

(d) Regret vs. Space  (e) Regret vs. Space  (f) Regret vs. Space

Figure 6: (a),(d): Pareto frontiers for regret vs. time and regret vs. space on cnae-9; (b),(e): Pareto frontiers for regret vs. time and regret vs. space on MFeat; (c),(f): Pareto frontiers for regret vs. time and regret vs. space on Spam

**Datasets.** We conduct experiments on three publicly available multiclass classification datasets from the OpenML repository (Vanschoren et al., 2013), as detailed in the table 3. The experimental setup is provided in Appendix F.1. The datasets include cnae-9, MFeat, and Spam, each with varying numbers of instances, features, and classes. These datasets are utilized to evaluate the performance of our proposed method across multiclass classification tasks.

**Pareto Frontier Analysis.** The Pareto frontier evaluation across three datasets (cnae-9, MFeat, and Spam) reveals the superior trade-off characteristics of our proposed DBSLinUCB methods. As illustrated in Figure 6, both DBSLinUCB-FD and DBSLinUCB-RFD consistently establish better positions on the Pareto frontiers for both regret vs. time and regret vs. space trade-offs. Notably, our methods form well-positioned curves that span a wide range of efficiency levels while maintaining competitive regret performance, demonstrating the flexibility and adaptability of the dyadic block sketching approach across different resource constraints.

**Adaptation to Low-Dimensional Data.** On relatively low-dimensional datasets, particularly on Spam (Figures 6c and 6f), OFUL serves as a strong baseline with an excellent trade-off between regret, time, and space. Notably, our DBSLinUCB-RFD variants closely approach the OFUL curve across multiple configurations. While this demonstrates a clear advantage over single-scale sketching methods, it also suggests that our method may incur some cost when degenerating to OFUL, a phenomenon more pronounced in certain low-dimensional settings.

**Variants Consistency and Robustness** Across multiple datasets (especially in cnae-9, Figures 6a and 6d), our method achieves similar trade-offs in terms of regret, space, and time for both

FD and RFD variants. In contrast, single-scale sketching methods, such as SOFUL (FD-based) and CBSCFD (RFD-based), exhibit significantly different trade-offs (e.g., Figures 6a, 6c, 6d, and 6f).

This performance consistency arises because our method effectively controls global error in matrix approximation through a unified parameter $\epsilon$, and replacing sketching methods does not result in a substantial loss of accuracy. This stability highlights the robustness of our dyadic block sketching framework, offering reliable performance regardless of the selected underlying sketching technique. Unlike traditional single-scale methods, where the choice between FD and RFD can dramatically affect performance, our approach ensures consistent outcomes.

