# OpenReview forum: "Revisiting Matrix Sketching in Linear  Bandits: Achieving Sublinear Regret via Dyadic Block Sketching"
_ICLR.cc/2026/Conference — ICLR 2026 Poster_

### Official Review · Reviewer_1de1 · 2025-10-30

**Soundness:** 3
**Presentation:** 3
**Contribution:** 2
**Rating:** 6
**Confidence:** 3

**Summary:**

This paper revisits efficient linear bandits using matrix sketching. Matrix sketching approximates the streaming matrix X to reduce update costs. Previous methods such as SOFUL lower the per-round complexity from \Omega(d^2) to O(dl + l^2) by maintaining low-rank sketches, but their regret can become linear when the sketch size is too small or the spectrum decays slowly. The authors propose Dyadic Block Sketching (DBS), a new framework that adaptively adjusts the sketch size at multiple scales without prior knowledge of the data. Applied to linear bandits, this leads to the DBSLinUCB algorithm, which achieves sublinear regret under general conditions. Both theoretical analysis and experiments on synthetic and real-world datasets support its effectiveness.

**Strengths:**

1. Theoretical guarantees: When an appropriate value of \epsilon is chosen in advance, DBSLinUCB achieves the O(\sqrt{T}) regret of OFUL (Theorem 3), which SOFUL cannot do without knowing spectral information. Even in the worst case, the algorithm attains an O(dk) update complexity (Corollary 1) while maintaining robust regret guarantees.

2. Balanced trade-off: Regret can be controlled via fixed parameters (\epsilon, l_0), while the update cost depends adaptively on the matrix rank k or Frobenius norm \|\tilde{X}\|^2_F, achieving a balance between theory and efficiency.

3. Flexibility: The DBS framework is modular and compatible with methods such as FD and RFD, supporting wide applications in online and bandit learning.

4. Empirical results: Experiments on synthetic and real datasets show consistent improvements in regret and efficiency, confirming robustness under different settings.

**Weaknesses:**

1. Incomplete theoretical coverage: The theoretical guarantees do not fully subsume SOFUL. When l < k and \|\tilde{X}\|^2_F is unknown, the analysis cannot ensure an update cost of O(dl + l^2), and no choice of (\epsilon, l_0) achieves SOFUL-level efficiency.

2. Efficiency–regret trade-off: Achieving near-optimal O(d\sqrt{T}) regret may lead to high update costs, especially when the data matrix has large rank or Frobenius norm. It would help to provide examples or evidence showing that in practice, k \ll d or \|\tilde{X}\|^2_F is small (e.g., constant or o(T^{1/3})).

3. Lack of empirical context: In the MNIST experiments (Figure 5), the rank k of the dataset is not reported. Without this, the complexity comparison is hard to interpret—particularly when k is close to d, where DBSLinUCB may be much slower than SOFUL.

**Questions:**

Please address the concerns raised in Weaknesses.

---

> ### Author Response · Authors · 2025-11-17
> **Response to Reviewer 1de1 (Part1/2)**
>
> **Response to Reviewer 1de1**:
>
> Thank you for your detailed review of our work! We sincerely appreciate the time and effort you have dedicated to reviewing our paper. We have addressed your concerns and provided clarifications below.
>
>
> ---
>
> **W1:** *Incomplete theoretical coverage.*
>
> > Reviewer: "The theoretical guarantees do not fully subsume SOFUL."
>
> We thank the reviewer for raising this point. We believe the confusion may stem from differences in the preconditions required for the theoretical guarantees. Whether our results subsume those of SOFUL depends on the following two cases:
>
> 1. **When SOFUL achieves a $\tilde{O}(\sqrt{T})$ regret bound.**
>    As discussed in lines 372–380, under low-rank conditions (where the optimal sketch size is $l = k$), our method matches both the regret bound and the computational complexity of SOFUL. In this regime, our guarantees indeed subsume SOFUL's.
>
> 2. **When SOFUL’s regret bound exceeds $\tilde{O}(\sqrt{T})$.**
>    In this setting, our method cannot—and **should not**—simultaneously match SOFUL’s regret and complexity. As stated in Observation 1, when the geometry constant $q \ge 1/3$, any SOFUL variant with sketch size $l < d$ incurs linear regret. In such cases, achieving SOFUL’s $O(dl)$ complexity is not meaningful, because **it comes at the cost of losing sublinear regret**.
>
> In summary, the central focus of our theoretical analysis is to ensure sublinear regret. Under this premise, our method can recover the computational advantages of SOFUL when the streaming matrix exhibits favorable structure. We hope this clarifies the conditions under which our guarantees align with or exceed those of SOFUL.
>
> > Reviewer: “no choice of $(\epsilon, l_0)$ achieves SOFUL-level efficiency.”
>
> While achieving SOFUL-level efficiency in full generality does not guarantee sublinear regret, the reviewer raises an interesting question: **if we do not require a sublinear regret guarantee** (for example, in industrial applications where one already knows that SOFUL with a given sketch size $l$ performs well), can our method recover SOFUL-level efficiency, that is, an $O(dl)$ per-round update cost, under some configuration of $(\epsilon, l_0)$?
>
> Our answer is **yes**, and we provide two simple mechanisms to achieve this:
>
> 1. **Degenerating to a single-scale sketching method.**
>    When $\epsilon > \|X\|_F^2$ and $l_0 = l$, our method becomes exactly equivalent to SOFUL with sketch size $l$. In this configuration, the algorithm always uses a single sketch, effectively reducing to SOFUL’s single-scale matrix sketching procedure.
>
> 2. **Using a multi-scale sketching method with an upper bound on sketch size.**
>    By modifying $\log(d/l_0 + 1)$ to $\log(l/l_0 + 1)$ in line 5 of Algorithm 1, we enforce that all active blocks have sketch size at most $l$. Under this constraint, **any** choice of $(\epsilon, l_0)$ guarantees that the per-round update cost is at most $O(dl)$.
>
> These two configurations demonstrate that our framework is sufficiently flexible to recover SOFUL-level efficiency when regret guarantees are not the primary concern.

---

> ### Author Response · Authors · 2025-11-17
> **Response to Reviewer 1de1 (Part 2/2)**
>
> ---
>
> **W2, W3:** *Efficiency–regret trade-off.*
>
> > Reviewer: It would help to provide examples or evidence showing that in practice, $k \ll d$ or $\|\tilde{X}\|^2_F$ is small.
>
> Your understanding is correct. Achieving near-optimal $O(d\sqrt{T})$ regret may indeed result in high update costs. However, we observe that our method achieves near-optimal regret while maintaining efficiency across all four real-world datasets in our paper, as demonstrated in Figures 3 and 6.
>
> Since the update costs of our method are **data-dependent**, the algorithm remains efficient when $k \ll d$ or $||\tilde{X}||^2_F$ is small. Following your suggestion, we provide empirical evidence showing that $||\tilde{X}||^2_F$ is **indeed small** in practice. The following table reports the runtime, regret, and the value of $||\tilde{X}||^2_F$ for DBSLinUCB-FD on four real-world datasets:
>
>
>
>
> | Dataset | $\epsilon$ | Time (s) | Regret | $\Vert\tilde{X}\Vert_F^2$ | $\log(\Vert\tilde{X}\Vert_F^2)/\log T$ |
> |---------|------------|----------|--------|---------------------|----------------------------------|
> | MNIST   | 8          | 20.935   | 209    | 3.77                | 0.175 (< 1/3) |
> | cnae-9  | 16         | 38.449   | 195    | 5.31                | 0.220 (< 1/3) |
> | MFeat   | 4          | 2.175    | 128    | 1.70                | 0.069 (< 1/3) |
> | Spam    | 2          | 0.073    | 123    | 1.69                | 0.091 (< 1/3) |
>
> Across all datasets, $||\tilde{X}||_F^2$ remains small, satisfying $||\tilde{X}||_F^2 = o(T^{1/3})$. This indicates that the information capacity of our multi-scale matrix sketch is **sufficient** to approximate the streaming matrix effectively. Combined with our complexity formula in Corollary 1, which incorporates both $||\tilde{X}||_F^2$ and $k$, these results demonstrate that our method achieves near-optimal regret (differing only by a constant factor) with **low** computational complexity under practical conditions with decaying spectral tails and low-rank structure.
>
>
> > Reviewer: particularly when $k$ is close to $d$, where DBSLinUCB may be much slower than SOFUL.
>
> We have added the rank information to the manuscript and report that the streaming matrix is full rank. As the results above show, even when $k$ is close to $d$, our method remains efficient when $\|\tilde{X}\|_F^2$ is small. When both $k \approx d$ and $\|\tilde{X}\|_F^2$ are very large, our method naturally degenerates to OUFL, which is indeed slower than SOFUL. However, this behavior is aligned with the regret guarantees: in such scenarios, SOFUL would incur significantly larger regret, as illustrated by the SOFUL curve with $l = 20$ in Figure 5.
>
> > Reviewer: Without this, the complexity comparison is hard to interpret
>
> We respectfully disagree with this point, since a meaningful complexity comparison should be interpreted together with **regret**. We kindly invite the reviewer to revisit Figures 3(c), 3(d), and 6, which present the **efficiency–regret Pareto fronts** under fair parameter sweeps for all sketch-based linear bandit methods. Across all datasets, our method consistently attains the best trade-off, empirically demonstrating that it achieves higher efficiency for the same level of regret.
>
>
> ---
>
> Thank you once again for your support of our work. We are hopeful that our research will have a positive impact on the community. If you have any further questions, we look forward to continued discussions.

---

### Official Review · Reviewer_v1kF · 2025-11-01

**Soundness:** 3
**Presentation:** 3
**Contribution:** 3
**Rating:** 6
**Confidence:** 2

**Summary:**

This paper shows that existing sketch-based linear bandits can suffer linear regret when the data matrix has heavy spectral tails. The authors propose Dyadic Block Sketching (DBS), which adaptively adjusts sketch sizes to control global error. Applied to linear bandits, their DBSLinUCB algorithm achieves sublinear regret and better efficiency than prior methods.

**Strengths:**

1. This paper provides a clear motivation by highlighting the pitfalls of previous studies. And the introduced of multi-scale sketching approach is well-grounded.

2. The algorithm and its analysis are presented clearly, with good intuition and easy-to-follow explanations. The numerical experiments use meaningful benchmarks, and the results, particularly in Figure 3(c), convincingly validate the claimed performance improvement.

**Weaknesses:**

I did not find any major technical weaknesses in this paper. However, as a reader who is not very familiar with matrix sketching applications in bandits or online learning, I have several questions about the positioning of this work and the choice of benchmarks, as mentioned in the question part.

**Questions:**

1. The discussion on the parameter choices in Remark 2 is insightful. I am wondering is there any approach on adaptively estimating l0 to make the selection of parameters adaptive to environments.

2. I am not deeply familiar with the literature on matrix sketching for linear bandits, but I noticed that most baselines in this paper are from around five years ago (e.g., SOFUL [Kuzborskij et al., 2019], CBSCFD [Chen et al., 2020]). It would be helpful if the authors could comment on or compare with more recent works, such as Zhang et al. (2023) and Feinberg et al. (2023).

3. In the related work section on multi-scale sketching, the authors explain that previous algorithms were developed for different purposes. I am curious whether there are shared ideas or overlapping design principles between those existing methods and the proposed Dyadic Block Sketching framework.

---

> ### Author Response · Authors · 2025-11-17
>
> **Response to Reviewer v1kF**:
>
> Thank you for your detailed review of our work! We sincerely appreciate the time and effort you have dedicated to reviewing our paper. We have addressed your concerns and provided clarifications below.
>
> ---
>
> **Q1:** *Estimating $l_0$.*
>
> We appreciate the reviewer’s recognition of our discussion in Remark 2. While the initial sketch size $l_0$ does not affect the order of the regret bound, it does influence the computational complexity of our method. We provide a **simple and practical** procedure for selecting $l_0$ in a way that adapts to the environment:
>
> 1. Sample a small number of rows to form a submatrix $X_s$.
> 2. Compute the SVD of $X_s$ (which is inexpensive due to its small size).
> 3. Examine the decay pattern of its singular values.
> 4. Identify the first “clear inflection point’’ in the decay curve and set the corresponding rank as $l_0$.
>
> For example, if the first 15 singular values are relatively large and a sharp decay begins at the 16th, then one may choose $l_0 = 16$. This heuristic is supported by the fact that random row sampling can reliably preserve the dominant singular subspace of the original matrix [1].
>
> ---
>
> **Q2:** *Related Works*
>
> In this paper, we revisit matrix sketching techniques in the context of linear bandits. To the best of our knowledge, SOFUL and CBSCFD represent the current state of the art under this specific setting. More recent works—such as those on batch bandits [2]—operate under substantially **different** and more complex problem formulations, and thus fall outside the scope of this paper and are not directly comparable.
>
> At the same time, these recent developments highlight the growing importance of matrix sketching for resource-constrained online learning. As noted in Appendix A (line 775), existing methods are primarily based on **single-scale** sketches, which can lead to performance degradation when the sketch fails to capture sufficient information. Our work takes a step back to the foundational linear bandit setting, and we hope it offers insight for the community toward developing more systematic and theoretically grounded solutions for **other** online learning problems.
>
> ---
>
> **Q3:** *Multi-scale Sketching*
>
> Multi-scale sketching is a well-studied technique in streaming algorithms, and our Dyadic Block Sketching framework is partially inspired by this line of work. For example, in the sliding-window data structure problem, [3] also leverages the decomposability of matrix sketches (similar to Lemma 3 in our paper). In essence, it maintains multiple FDs (merged into $S_W$) to approximate the matrix $X_W$ over a sliding window $W$, yielding the guarantee
> $$
> ||X_W^{\top}X_W - S_W^{\top}S_W||_2 \le \epsilon ||X_W||_F^2,
> $$
> where $S_W \in \mathbb{R}^{l \times d}$ is significantly smaller than $X_W \in \mathbb{R}^{W \times d}$.
>
> While the high-level design shares similarities, we emphasize that our method is **fundamentally different** from these algorithms, as discussed in Appendix A (line 792). In particular, the sketch $S_W$ in the above example has a **fixed** size $l$, whereas the size of the sketch in our Dyadic Block Sketching framework is data-dependent and **dynamically adjusted**. To the best of our knowledge, our work is the first to introduce the concept and principles of multi-scale sketching into the online learning setting. The technical challenges addressed by our design and theoretical analysis differ substantially from those encountered in multi-scale sketching for streaming algorithms.
>
> ---
>
> Thank you once again for your time and thoughtful review of our work! We believe that our research will contribute positively to the community. Should you have any further questions or feedback, we would be glad to continue the discussion.
>
> ---
> **Reference**
>
> [1] Mahoney, Michael W., and Petros Drineas. CUR Matrix Decompositions for Improved Data Analysis.
>
> [2] Xiao Zhang, Ninglu Shao, Zihua Si, Jun Xu, Wenhan Wang, Hanjing Su, and Ji-Rong Wen. Reward imputation with sketching for contextual batched bandits.
>
> [3] Zhewei Wei, Xuancheng Liu, Feifei Li, Shuo Shang, Xiaoyong Du, and Ji-Rong Wen. Matrix sketching over sliding windows.

---

### Official Review · Reviewer_pFPs · 2025-11-02

**Soundness:** 3
**Presentation:** 3
**Contribution:** 3
**Rating:** 6
**Confidence:** 4

**Summary:**

The authors first show that existing sketch-based linear bandit algorithms can suffer from linear regret. To address this issue, they propose a novel matrix sketching framework called Dyadic Block Sketching (DBS), which adaptively adjusts the sketch size in a multi-scale manner. By applying DBS to linear bandits, the authors achieve sublinear regret bounds. They further validate the effectiveness of the proposed method through experiments on both synthetic and real-world datasets.

**Strengths:**

1. The motivation of this paper is clearly presented.
2. The proposed methods are novel and interesting.
3. Although I did not examine the proofs in the appendices in detail, the theoretical results appear convincing and reasonable.
4. The writing is generally clear, though some parts require further clarification (see Weaknesses).

**Weaknesses:**

1. In lines 372–379, the authors discuss choosing $\epsilon$ based on the spectral properties of the data matrix. However, in linear bandit problems, it is usually unknown whether the data matrix is low-rank or has a heavy spectral tail. How should $\epsilon$ be selected in practice? Moreover, in the experiments, how was the value of $\epsilon$ determined?

2. Some experimental results require further clarification:
- In Figure 3(a), why does the regret of $\epsilon=4$ outperform that with $\epsilon=2$?
- In Figure 3(d), when the sketch-based methods use the same amount of space as OFUL, their performance is still inferior to OFUL. The authors should provide a more detailed explanation for this result.

3. Some statements in the paper are unclear or inaccurate:
- In the abstract, the authors claim that “the sketch-based approaches reduce per-round complexity from Ω($d^2$) to O($d$),” which is not accurate. The computational complexity of matrix sketching depends on the sketch size. If the sketch size is $O(d)$, the overall complexity of sketch-based methods remains $O(d^2)$.
- In line 66, the statement “thereby reducing the order of spectral error $\Delta_T$ and decoupling it from $d$” is difficult to understand for readers unfamiliar with Chen et al. (2020). It is recommended that the authors present the regret bound from Chen et al. (2020) in the paper for clarity.
- In the last line of page 3, the phrase “under certain conditions” is too vague. The authors should describe these conditions in more detail.
- In the last line of page 5, the phrase “$\tilde{X}$ the subset of rows approximated by inactive blocks” is confusing. Does it mean “$\tilde{X}$ is the subset of rows consisting of inactive blocks”?
- Algorithm 3 is commonly referred to as SCFD (Chen et al., 2020) rather than RFD (Luo et al., 2019). Note that the regularizer in SCFD sums the total mass of subtracted values during the FD procedure, whereas in RFD, it sums only half of the subtracted mass.

**Questions:**

see Weaknesses

---

> ### Author Response · Authors · 2025-11-17
> **Response to Reviewer pFps (Part 1/2)**
>
> **Response to Reviewer pFps**:
>
> Thank you for your detailed review of our work! We sincerely appreciate the time and effort you have dedicated to reviewing our paper. We have addressed your concerns and provided clarifications below.
>
> ---
>
> **W1:** *Selection of $\epsilon$.*
>
> > Reviewer: “In lines 372–379, the authors discuss choosing $\epsilon$ based on the spectral properties of the data matrix.”
>
> We thank the reviewer for highlighting the importance of selecting the error parameter $\epsilon$, which is indeed a central component of our method. However, the discussion in lines 372–379 may have been misunderstood. This part is **not intended** to prescribe how $\epsilon$ should be chosen in practice; rather, it illustrates that under low-rank conditions, our method can match SOFUL’s optimal complexity. Achieving this match is straightforward, as it only requires ensuring $\epsilon < \|\tilde{X}\|_F^2 / \log k$.
>
> > Reviewer: "in linear bandit problems, it is usually unknown whether the data matrix is low-rank or has a heavy spectral tail"
>
> We agree that in many linear bandit applications, no prior structural information is available, and this principle guides how $\epsilon$ should be chosen in practice. As discussed in Remark 2, our selection **does not rely on any prior assumptions** about the spectrum of the data matrix. Since $\epsilon$ appears **explicitly** in the regret bounds of Theorems 3 and 4, users may set it based on the worst-case regret performance they are willing to tolerate. For instance, if one wishes the regret to differ from the optimal OFUL bound only by a constant factor, it suffices to set $\epsilon = O(1)$. Importantly, even when the spectrum is completely unknown, our method still guarantees sublinear regret, whereas prior sketch-based approaches may incur linear regret in such settings.
>
> > Reviewer: " in the experiments, how was the value of $\epsilon$ determined"
>
> Following the guidance in Remark 2, we chose $\epsilon$ as a small constant in experiments. We also performed **efficiency–regret trade-off studies** with parameter tuning on all real datasets (Figures 3 and 6). These results show that our method achieves favorable efficiency–regret trade-offs and maintains low regret even at low computational budgets, reinforcing the clarification provided above.

---

> ### Author Response · Authors · 2025-11-17
> **Response to Reviewer pFps (Part 2/2)**
>
> ---
>
> **W2:** *Experimental Results*
>
> We thank the reviewer for pointing this out, and we will clarify the two experimental observations separately to resolve the confusion.
>
> > Reviewer: “In Figure 3(a), why does the regret with $\epsilon = 4$ outperform that with $\epsilon = 2$?”
>
> This occurs because Figure 3(a) reports the **actual** cumulative regret, whereas $\epsilon$ controls the **upper bound** on regret. When $\epsilon = 2$, the algorithm tends to create new blocks more frequently, leading to insufficient sketching and a poorer matrix approximation. To support this explanation, we have added the cumulative regret, regret bound, average matrix approximation error $||X^\top X - S^\top S||_2$, and its corresponding upper bound for $\epsilon = 2$ and $\epsilon = 4$ when $l_0 = 50$:
>
> | $\epsilon$ | Regret | Regret Bound | Matrix Approx. True Error | Matrix Approx. Error Bound |
> |-----------:|--------|---------------|----------------------------|-----------------------------|
> | 2          | 207  (↑ worse)  | 232           | 1.06      (↑ worse)                 | 4                           |
> | 4     | 200    | 500  (↑ worse)         | 0.95                       | 8      (↑ worse)                     |
>
> These results highlight two key points:
>
> 1. The actual cumulative regret correlates strongly with the **true matrix approximation error**.
> 2. Our theoretical upper bounds on the matrix approximation error and regret are valid, but their **tightness** may vary with parameter choices. Nevertheless, as shown in Figures 3(a) and 3(b), the empirical regret trends remain **almost consistent** with the behavior suggested by the theoretical bounds.
>
> > Reviewer: "In Figure 3(d), when the sketch-based methods use the same amount of space as OFUL, their performance is still inferior to OFUL. "
>
> We appreciate the reviewer’s careful observation. This behavior arises because, in our experiments, all sketch-based methods (including the baseline) use the **fast-update variant** described in Remark 1 and Appendix B.2. In this variant, we avoid performing an SVD at every update, which would require $O(dl^2)$ time. Instead, we reduce the amortized update cost to $O(dl)$ by allocating **twice the sketch size** and performing batched SVD updates. As a result, although the sketch-based methods in Figure 3(d) use the same total amount of space as OFUL, they effectively operate with a low-rank approximation of rank $l = d/2$, which naturally leads to inferior performance compared with OFUL. We have revised the paper to clarify this implementation detail before the experimental section.
>
>
> ---
>
> **W3:** *Unclear Statements*
>
> We thank the reviewer for carefully identifying these unclear or inaccurate statements. We have revised the manuscript accordingly, with the detailed changes summarized below.
>
> • **Abstract statement on computational complexity.**  We have corrected the description in the abstract. The revised version now explicitly states that the per-round complexity reduction depends on the sketch size.
>
> • **Clarification of the statement in line 66.** To improve clarity, we now explicitly cite and present the regret bound from Chen et al. (2020) in the main text.
>
> • **Vague phrase “under certain conditions.”**. We have replaced this vague expression with a precise description of the required conditions in the last line of page 3.
>
> • **Clarification of the phrase “$\tilde{X}$”**. The inactive block stores multiple matrix sketches with different sketch sizes, and $\tilde{X}$ denotes the submatrices that these sketches approximate.
>
> • **Naming of Algorithm 3.**  Following the reviewer’s suggestion, we have revised the manuscript to state that Algorithm 3 is commonly referred to as SCFD (Chen et al., 2020) in the linear bandit literature (footnote).
>
> ---
>
>
> Thank you again for your time and thoughtful review of our work. As you noted under Strengths, **our motivation is clear**: understanding the computational complexity required to achieve a given order of regret is an important perspective that complements and deepens prior work. We believe our work will positively impact the community. If you have any further questions or suggestions, we would be very happy to continue the discussion.

---

> > ### Comment · Reviewer_pFPs · 2025-11-27
> >
> > Thank you for your response which addresses most of my concerns. I have raised my score to 8. I suggest adding the response to W2 into the paper.

---

> > > ### Author Response · Authors · 2025-11-27
> > >
> > > We sincerely appreciate your positive evaluation and recognition of our work. In response to your feedback, we will revise the manuscript to include additional details regarding W2. After communicating with all reviewers, we plan to upload an updated version of the revision before the rebuttal deadline.
> > >
> > > Your insightful comments and suggestions have been very helpful in improving the clarity and overall quality of our manuscript. Thank you again for your thoughtful and constructive review.

---

### Official Review · Reviewer_NX8x · 2025-11-03

**Soundness:** 3
**Presentation:** 3
**Contribution:** 3
**Rating:** 8
**Confidence:** 2

**Summary:**

This paper revisits the computational and regret trade-offs in sketch-based linear bandits, where matrix sketching is employed to reduce the per-round computational cost from quadratic $O(d^2)$---with $d$ denoting the feature dimension---to subquadratic, while maintaining sublinear regret in $T$, the time horizon. Earlier work has shown that for linear bandits regret of order $O(d\sqrt{T})$ can be achieved with per-round complexity proportional to $O(d^2)$. Subsequent research demonstrated how to reduce this computational burden using dimensionality reduction techniques, particularly matrix sketching, to achieve per-round complexity $O(dl + l^2)$, where $l$ is the sketch size. However, in these approaches, the regret depends on the *spectral error* introduced by dimensionality reduction and can become linear (i.e., vacuous) when the spectral error exceeds $T^{1/3}$.

This paper makes progress on this front by introducing an algorithm termed *Dyadic Block Sketching (DBS)*---a multi-scale sketching framework that adaptively doubles the sketch size as learning progresses. The key claim is that this adaptive structure bounds the global covariance error by a user-specified parameter $\varepsilon$, thereby ensuring sublinear regret independent of the spectral error. Empirical results on synthetic and real-world datasets (e.g., MNIST) support the theoretical analysis, showing improved regret–efficiency trade-offs compared to prior methods.

**Strengths:**

The computational efficiency of linear bandits is an important and active area. Reducing per-round complexity while retaining sublinear regret is a significant theoretical question. The proposed framework makes a meaningful contribution toward that goal and the method is natural (adapting the sketch size). The paper is well written with detailed proof given in the appendix. The authors complement their theoretical work with experimental validation, which is an added bonus.

**Weaknesses:**

While the claimed regret bound is independent of the spectral error, it still depends on other parameters such as  the $\ell_2$-norm of the feature vectors, and the sketch size of the active block $l_B$. This dependence implies that in certain regimes, the regret may still exhibit linear scaling. Providing a clearer exposition or characterization of when such linear growth arises would  strengthen the paper. In particular, the paper would benefit from explicitly stating conditions under which their algorithm reverts to linear regret. This will lead to open questions that can be stated explicitly. Regarding exposition, for semi/non-experts, the presentation is dense and occasionally confusing. Terms like “streaming matrix” and “spectral tail” are introduced early without clear explanations. It is not clear what is streaming matrix means in this context.

**Questions:**

The problem you identified: getting sublinear regret algorithm for linear bandits with sub quadratic per-round complexity for is very interesting to me. So it will be very nice to the readers if you could make a table with what is known (including your work) and to what extend this is an open question. Making a separate section on related works and discussing this will also make the paper very nice to read.

---

> ### Author Response · Authors · 2025-11-17
> **Response to Reviewer NX8x**
>
> **Response to Reviewer NX8x**:
>
> Thank you for your detailed review of our work! We sincerely appreciate the time and effort you have dedicated to reviewing our paper. We have addressed your concerns and provided clarifications below.
>
> ---
>
> **W1.1:** *Linear Regret.*
>
> > Reviewer: "This dependence (e.g., $l_{B_T}$) implies that in certain regimes, the regret may still exhibit linear scaling."
>
> We thank the reviewer for highlighting the importance of understanding how other parameters influence the regret bound. We would like to clarify that, under the sketch-based linear bandit setting considered in this work (consistent with prior studies [1,2,3]), our regret bound does not incur linear growth and does not rely on prior knowledge of these parameters. This is because in this regime, the dependence on $T$ dominates the regret bound, and the additional terms—such as $l_{B_T}$—are **treated as constants** (e.g., $l_{B_T} \le d \ll T$) and therefore cannot induce linear regret.
>
> > Reviewer: Providing a clearer exposition or characterization of when such linear growth arises would strengthen the paper.
>
> Within the scope of this paper, our method guarantees **worst-case sublinear regret**; in particular, it reduces to OFUL in the appropriate regime and therefore inherits the $O(\sqrt{T})$ regret bound. Exploring whether linear regret may emerge under alternative settings such as non-stationary linear bandits [4] is an interesting direction for future work, and we appreciate the reviewer for highlighting this point.
>
> ---
>
> **W1.2:** *Clarification of Terminology.*
>
> We thank the reviewer for bringing this to our attention. The term streaming matrix is more commonly used in the traditional matrix sketching literature, where it refers to matrices that grow in a streaming, single-pass manner. We will revise the paper to clarify this usage and improve the overall presentation.
>
> ---
>
> **Q1:** *Summary Table*
>
> We appreciate the reviewer’s thoughtful suggestion. **In response, we have added a summary table** that highlights currently known sketch-based linear bandits, including our own contribution.
>
> While our method attains sublinear regret with sub-quadratic complexity under certain structural conditions (such as low rank or a decaying spectral tail), the optimal complexity achievable in more general settings is still unknown. This leads to an important open question for the field: what is the fundamental **lower bound** on complexity for any algorithm that achieves $\tilde{O}(T^\gamma)$ regret ($\gamma \in [0.5,1)$) ? We thank the reviewer for encouraging us to make these open directions more explicit.
>
> ---
>
> Thank you again for your valuable time and thoughtful feedback. The revisions we made in response to your comments have improved the quality of our work. We hope the paper makes a positive contribution to the community and offers additional insight. If you have any further questions or comments, we would be happy to continue the discussion.
>
> ---
>
> **Reference**
>
> [1] Yasin Abbasi-Yadkori, Dávid Pál, and Csaba Szepesvári. Improved algorithms for linear stochastic
> bandits.
>
> [2] Ilja Kuzborskij, Leonardo Cella, and Nicolò Cesa-Bianchi. Efficient linear bandits through matrix
> sketching.
>
> [3] Cheng Chen, Luo Luo, Weinan Zhang, Yong Yu, and Yijiang Lian. Efficient and robust high-dimensional linear contextual bandits.
>
> [4] Wei Chen-Yu, and Haipeng Luo. Non-stationary reinforcement learning without prior knowledge: An optimal black-box approach.

---

### Author Response · Authors · 2025-11-17
**Summary of Revisions**

We thank all four reviewers and AC for their time and constructive feedback. Regarding the strengths, all reviewers acknowledged the clarity of our motivation and the importance of exploring the computational efficiency of linear bandits. Regarding weaknesses and questions, Reviewers NX8x and v1kF primarily pointed out related work and open problems that merit further discussion; Reviewers pFPs and 1de1 raised questions concerning experimental results; and Reviewer pFPs also noted several unclear or imprecise statements in the manuscript.

We have addressed these concerns in our responses and revised the paper accordingly. **These updates have been highlighted in blue** for your convenience. The main revisions are summarized below:

1. Following Reviewer NX8x’s suggestion, we added a summary table in Appendix A comparing existing sketch-based linear bandit methods.
2. Following the suggestions of Reviewers pFPs and NX8x, we clarified terminology and added explanatory notes where needed.
3. Following Reviewer v1kF’s comment, we expanded the discussion of related work in Appendix A.
4. Following Reviewer 1de1’s suggestion, we now report the rank of the dataset in Appendix F.3.
5. With the extended page limit, we improved the paper’s organization by moving the notation and structural overview into the main text for better readability.

We believe these revisions have substantially improved the clarity and overall quality of the paper, and we will continue refining the manuscript based on the ongoing discussion during the rebuttal process.

---

### Meta-Review · Area_Chair_Z3ot · 2026-01-03

**Summary:**

This paper revisits sketch-based linear bandits and shows that improper sketch sizes can cause severe spectral errors and linear regret. To address this, it proposes Dyadic Block Sketching, a dynamic multi-scale approach that achieves sub-linear regret without prior knowledge of steaming matrix properties while maintaining computational efficiency. All the reviewers are positive on the acceptance, despite some concerns regarding the selection of parameters (e.g., $\epsilon$ and $\ell_0$). Therefore, my recommendation is acceptance.

**Reviewer Concerns:**

The reviewers raised concerns regarding the practical selection of key parameters. In response, the authors provided illustrative examples and general guidelines, but did not present a concrete algorithm for tuning these parameters.

The reviewers also questioned the assumptions underlying the analysis, particularly the appearance of certain parameters (e.g., $\ell_{B_{T}}$) in the regret bound. In response, the authors argued that their algorithm offers strictly improved guarantees when $\ell_{B_{T}}$ is treated as a constant.

**Reviewer Scores:**

I expect the reviewers would keep their positive scores after full participation in the discussion.

---

### Decision · Program_Chairs · 2026-01-26

Accept (Poster)